# SCALABLE VARIATIONAL BAYESIAN FINE-TUNING OF LLMS VIA ORTHOGONALIZED LOW-RANK ADAPTER

## ABSTRACT

When deploying large language models (LLMs) to safety-critical applications, uncertainty quantification (UQ) is of utmost importance to self-assess the reliability of the LLM-based decisions. However, such decisions typically suffer from overconfidence, particularly after parameter-efficient fine-tuning (PEFT) for downstream domain-specific tasks with limited data. Existing methods to alleviate this issue either rely on Laplace approximation based post-hoc framework, yielding uncalibrated uncertainty estimates, or variational Bayesian training that requires expensive Monte Carlo sampling with high computation and memory overheads. To address these limitations, we build on the Bayesian last layer (BLL) model, where the LLM-based *deterministic* feature extractor is followed by random LL parameters for uncertainty reasoning. Since existing low-rank adapters (LoRA) for PEFT have limited expressiveness due to rank collapse, we address this with Polar-decomposed Low-rank Adapter Representation (PoLAR), an orthogonalized parameterization paired with Riemannian optimization to enable more stable and expressive adaptation. Building on this PoLAR-BLL model, we leverage the variational (V) inference framework to put forth a scalable Bayesian fine-tuning approach which jointly seeks the PoLAR parameters and approximate posterior of the LL parameters via alternating optimization. The resulting PoLAR-VBLL is a flexible framework that nicely integrates architecture-enhanced optimization with scalable Bayesian inference to endow LLMs with well-calibrated UQ. Our empirical results verify the effectiveness of PoLAR-VBLL in terms of generalization and uncertainty estimation on both in-distribution and out-of-distribution data for various common-sense reasoning tasks.

## 1 INTRODUCTION

Large language models (LLMs) have demonstrated remarkable capabilities across diverse domains, from natural language understanding to complex reasoning tasks (Brown et al., 2020; Touvron et al., 2023). When deploying to safety-critical applications, uncertainty quantification (UQ) is of utmost importance to self-assess the reliability of the LLM-based decisions. While large-scale pre-trained models exhibit reasonable calibration during pre-training (Kadavath et al., 2022), they fail to accurately express predictive uncertainty after parameter-efficient fine-tuning (PEFT) using limited data in downstream tasks (Jiang et al., 2021). Particularly, fine-tuned LLMs often exhibit significant overconfidence, which poses serious risks in high-stakes scenarios where reliable uncertainty estimation is essential for trustworthy decision-making (Yang et al., 2024).

To endow fine-tuned LLMs with well-calibrated UQ, several attempts have been made by leveraging advances in Bayesian neural networks (BNNs). Ensemble approaches (Lakshminarayanan et al., 2017; Wang et al., 2023) require training multiple model copies, which incurs significant computational overhead (Wang et al., 2023). Post-hoc methods like Laplace approximation (LA) apply Bayesian inference after MAP estimation, but this bifurcated optimization—where posterior approximation is separated from training—leads to suboptimal estimation (Yang et al., 2024). Variational methods like BLoB (Wang et al., 2024), while enabling joint optimization of mean and covariance during training, require expensive Monte Carlo sampling with prohibitive memory overhead that scales poorly with model size, making them impractical for large-scale deployment. Going beyond these approaches, there are other methods for UQ in BNNs, including deep kernel learning (Wilson

et al., 2016) and variational Bayesian last layers (VBLL) (Harrison et al., 2024), which have not been explored for LLM fine-tuning.

On the other hand, the prohibitive computational cost of full fine-tuning has led to the widespread adoption of PEFT methods. The Low-Rank Adaptation (LoRA) (Hu et al., 2022), which parameterizes weight updates as the product of two low-rank matrices, suffers from directional diversity collapse where the stable rank often collapses to values close to 1, severely underutilizing the allocated subspace (Lion et al., 2025). Alternative approaches like DoRA (Liu et al., 2024) decompose weights into magnitude and direction but still suffer from suboptimal rank utilization, while AdaLoRA (Zhang et al., 2023) attempts adaptive rank allocation but requires expensive SVD operations. The recently proposed PoLAR (Lion et al., 2025) addresses these limitations through polar decomposition with orthogonality constraints, demonstrating superior rank utilization and providing an improved foundation for uncertainty quantification. However, existing Bayesian LLM fine-tuning approaches still adopt the vanilla LoRA representation. There is a pressing need to tailor advances in architecture-aware optimization to scale up the UQ performance of LLMs in practice.

Building on the aforementioned prior works, the contribution of this paper is summarized as follows.

- Relying on the Bayesian last layer (BLL) model, where the LLM-based *deterministic* feature extractor is followed by random LL parameters for UQ, we leverage PoLAR-based LLM adapter, an orthogonalized parameterization to alleviate the rank collapse of LoRA representation and further enable more stable and expressive adaptation (Lion et al., 2025).

- This PoLAR-BLL model is amenable to variational (V) training, where we will jointly seek the PoLAR parameters via efficient landing field methods in Riemannian optimization and the approximate posterior of the LL parameters. The resulting PoLAR-VBLL leverages both architectural improvements to the underlying adaptation mechanism and scalable Bayesian inference that alleviates the computational overheads of existing methods.

- Further, given the trained PoLAR parameters and variational posterior, we will apply an additional post-hoc LA around the posterior mean of the LL parameters to enhance uncertainty calibration. Different from existing LA-based methods that work with maximum-a-posterior (MAP) parameter estimates, the current approach also benefits from the variational training.

- Comprehensive evaluations corroborate that PoLAR-VBLL consistently outperforms existing approaches in both predictive accuracy and uncertainty calibration on both in-distribution and out-of-distribution tasks.

## 2 RELATED WORK

### 2.1 UQ FOR FINE-TUNED LLMS AND BNNS

While large-scale pre-trained models exhibit reasonable calibration during pre-training (Kadavath et al., 2022), they fail to accurately express predictive uncertainty after fine-tuning (Jiang et al., 2021), particularly when adapted to domain-specific tasks with limited data. This degradation necessitates Bayesian approaches for reliable uncertainty estimation in safety-critical applications. Recent Bayesian PEFT methods exhibit limitations. Ensemble approaches (Lakshminarayanan et al., 2017) require training multiple LoRA copies with significant computational overhead. Laplace-LoRA (Yang et al., 2024) applies post-hoc approximation after MAP estimation, but this bifurcated optimization leads to suboptimal posterior estimates. BLoB (Wang et al., 2024) performs variational inference directly on LoRA parameters during training, achieving joint mean-covariance optimization. However, BLoB requires expensive Monte Carlo sampling for evaluating the so-termed evidence lower bound, resulting in prohibitive memory overhead that scales poorly with model size, while remaining fundamentally constrained by LoRA's low stable rank. Several variants aim to reduce BLoB's high memory cost. ScalaBL (Samplawski et al., 2025) uses stochastic subspace inference to reduce the number of variational parameters; C-LoRA (Rahmati et al., 2025) replaces them with deterministic contextual MLPs; and TFB (Shi et al., 2024) applies post-hoc search for Bayesian inference of deterministically trained adapters. Despite these optimizations, these methods still rely on Monte Carlo sampling and suffer from inference latency. VBLL (Harrison et al., 2024) demonstrates superior computational efficiency compared to full-adapter Bayesian methods

by targeting only the classification layer. Unlike BLoB's expensive Monte Carlo sampling, VBLL achieves analytical solutions with memory complexity scaling with output dimensions rather than adapter parameters. Previous applications in Bayesian optimization demonstrate its effectiveness (Brunzema et al., 2025), but its adaptation to LLM fine-tuning with advanced adapter architectures, such as PoLAR, remains unexplored.

## 2.2 PARAMETER-EFFICIENT FINE-TUNING

The prohibitive computational cost of full fine-tuning for billion-parameter models has made parameter-efficient fine-tuning essential. LoRA (Hu et al., 2022) has gained widespread adoption by learning additive low-rank updates $\Delta \mathbf{W}$ on top of the frozen pre-trained weights $\mathbf{W}$. Subsequent work has aimed to improve LoRA's effectiveness further. AdaLoRA (Zhang et al., 2023) introduces adaptive rank allocation during training. DoRA (Liu et al., 2024) decomposes weights into magnitude and direction components. GaLore (Zhao et al., 2024) applies low-rank projection to optimizer states to reduce memory requirements. However, recent analysis reveals fundamental limitations: LoRA suffers from directional diversity collapse where the stable rank of $\Delta \mathbf{W}$ remains well below the allocated linear algebraic rank, limiting expressiveness. PoLAR addresses this through a re-parametrization with orthogonal constraints on direction matrices, and a tailored Riemannian optimization (Ablin & Peyré, 2022) is employed for faster training on GPUs. In spite of these advances, adaptation to the Bayesian counterparts remains a rather uncharted territory – existing Bayesian fine-tuning approaches all rely on the vanilla LoRA.

## 3 VARIATIONAL TRAINING OF LLM-BASED BAYESIAN LAST LAYER MODEL VIA ORTHOGONAL LOW-RANK ADAPTATION

Toward adapting the advances of PEFT so as to endow uncertainty-aware fine-tuning of LLM with scalability, we present a unified framework that combines Polar-decomposed Low-rank Adapter Representation (PoLAR) with Variational Bayesian Last Layers (VBLL) for parameter-efficient fine-tuning of LLMs with principled uncertainty quantification. The resulting approach addresses the fundamental limitation of standard LoRA's low stable rank while providing calibrated uncertainty estimates through scalable variational Bayesian inference.

## 3.1 BAYESIAN LAST LAYER MODEL WITH LLM-BASED FEATURE EXTRACTOR

To endow LLM-based inference with UQ, we will rely on the Bayesian last layer (BLL) model Harrison et al. (2024), where a deterministic LLM-based feature extractor is followed by random last layer weights for uncertainty representation. Specifically, let $\boldsymbol{\phi}_{\mathbf{W}}(\mathbf{x}) \in \mathbb{R}^d$ be the $d$-dimensional feature mapping with $\mathbf{W}$ collecting the weights of the LLM. Given training dataset $\mathcal{D} := \{(\mathbf{x}_n, \mathbf{y}_n)\}_{n=1}^N$ with $\mathbf{y}_n := [y_{n,1}, \ldots, y_{n,C}]^\top \in \{0,1\}^{C \times 1}$ being a one-hot encoding of a $C$-class classification task, the BLL model per sample $n$ is given by

$$p(\mathbf{y}_n|\mathbf{x}_n, \boldsymbol{\Theta}) = \frac{\exp(\mathbf{y}_n^\top \mathbf{z}_n)}{\mathbf{1}_C^\top \exp(\mathbf{z}_n)}, \ \ \mathbf{z}_n = \boldsymbol{\Theta}\boldsymbol{\phi}_{\mathbf{W}}(\mathbf{x}_n) \tag{1}$$

where $\mathbf{1}_C$ is a $C \times 1$ all-one vector, $\boldsymbol{\Theta} = [\boldsymbol{\theta}_1, \ldots, \boldsymbol{\theta}_C]^\top \in \mathbb{R}^{C \times d}$ is the classification weight matrix with $\boldsymbol{\theta}_c \in \mathbb{R}^d$ being the *random* weight vector for class $c$ with iid Gaussian prior

$$p(\boldsymbol{\Theta}) = \prod_{c=1}^C \mathcal{N}(\boldsymbol{\theta}_c; \mathbf{0}, \sigma_\theta^2 \mathbf{I}_d) \tag{2}$$

where the prior variance $\sigma_\theta^2$ is a *hyperparameter* to be tuned.

Direct optimization of the marginal likelihood $p(\mathcal{D}) = \int p(\mathcal{D}|\boldsymbol{\Theta})p(\boldsymbol{\Theta})d\boldsymbol{\Theta}$ is intractable due to the nonlinear softmax-based likelihood function in equation 1. Moreover, gradient computation would require the full marginal likelihood, making mini-batch training impossible, and the flexibility of neural network features can lead to over-concentration of the posterior. To address these issues, we rely on the variational inference framework that jointly seeks the model parameters $\mathbf{W}$ and parameter posterior $q(\boldsymbol{\Theta})$ by maximizing the evidence lower bound (ELBO)

$$\mathcal{L}^{\text{ELBO}}(q(\boldsymbol{\Theta}), \mathbf{W}; \mathcal{D}) = \mathbb{E}_{q(\boldsymbol{\Theta})}[\log p(\mathcal{D}|\boldsymbol{\Theta})] - \text{KL}(q(\boldsymbol{\Theta})\|p(\boldsymbol{\Theta})) \tag{3}$$

where

$$\log p(\mathcal{D}|\boldsymbol{\Theta}) = \log \prod_{n=1}^{N} \log p(\mathbf{y}_n|\mathbf{x}_n, \boldsymbol{\Theta}) = \sum_{n=1}^{N} \mathbf{y}_n^{\top} \mathbf{z}_n - \sum_{n=1}^{N} \log(\mathbf{1}_C^{\top} \exp(\mathbf{z}_n)) \qquad (4)$$

For the sake of tractability, the approximate posterior of $\boldsymbol{\Theta}$ will be assumed to be factorizable across classes and the per-class parameter posterior will be approximated by a Gaussian with mean $\boldsymbol{\mu}_c$ and covariance $\mathbf{S}_c$, namely,

$$q(\boldsymbol{\Theta}) = \prod_{c=1}^{C} q(\boldsymbol{\Theta}_c) = \prod_{c=1}^{C} \mathcal{N}(\boldsymbol{\theta}_c; \boldsymbol{\mu}_c, \mathbf{S}_c) \qquad (5)$$

where we assume factorization across classes (reducing computational complexity) while retaining full covariance $\mathbf{S}_c \in \mathbb{R}^{d \times d}$ within each class (capturing feature correlations).

Taking the expectation of (4) wrt $q(\boldsymbol{\Theta})$ in (5) is intractable due to the log-softmax. We will apply again Jensen's inequality to yield a lower bound as

$$\mathbb{E}_{q(\boldsymbol{\Theta})} \left[ -\log \sum_{c=1}^{C} \exp(z_{n,c}) \right] \geq -\log \mathbb{E} \left[ \sum_{c=1}^{C} \exp(z_{n,c}) \right] = -\log \sum_{c=1}^{C} \mathbb{E}\left[\exp(z_{n,c})\right] . \qquad (6)$$

Since $z_{n,c} = \boldsymbol{\theta}_c^{\top} \boldsymbol{\phi}_{\mathbf{W}}(\mathbf{x}_n)$ and $\boldsymbol{\theta}_c \sim \mathcal{N}(\boldsymbol{\mu}_c, \mathbf{S}_c)$, we have

$$\mathbb{E}[z_{n,c}] = \boldsymbol{\mu}_c^{\top} \boldsymbol{\phi}_{\mathbf{W}}(\mathbf{x}_n) \qquad (7)$$

$$\mathbb{E}[\exp(z_{n,c})] = \exp\left( \boldsymbol{\mu}_c^{\top} \boldsymbol{\phi}_{\mathbf{W}}(\mathbf{x}_n) + \frac{1}{2} \boldsymbol{\phi}_{\mathbf{W}}(\mathbf{x}_n)^{\top} \mathbf{S}_c \boldsymbol{\phi}_{\mathbf{W}}(\mathbf{x}_n) \right) \qquad (8)$$

Further, the KL divergence term is expressed explicitly as

$$\mathrm{KL}(q(\boldsymbol{\Theta})\|p(\boldsymbol{\Theta})) = \sum_{c=1}^{C} \mathrm{KL}(\mathcal{N}(\boldsymbol{\mu}_c, \mathbf{S}_c)\|\mathcal{N}(\mathbf{0}, \sigma_{\theta}^2 \mathbf{I}_d))$$

$$= \sum_{c=1}^{C} \left( \frac{1}{2\sigma_{\theta}^2} \left( \mathrm{tr}(\mathbf{S}_c) + \boldsymbol{\mu}_c^{\top} \boldsymbol{\mu}_c \right) - \frac{1}{2} \log |\mathbf{S}_c| \right) + \frac{dC}{2} \log \sigma_{\theta}^2 - \frac{dC}{2} \qquad (9)$$

Combining all terms, the ELBO objective is given by

$$\mathcal{L}^{\mathrm{ELBO}}(\boldsymbol{\Psi}, \mathbf{W}; \mathcal{D}) = \sum_{n=1}^{N} \left[ \sum_{c=1}^{C} y_{n,c} \boldsymbol{\mu}_c^{\top} \boldsymbol{\phi}_{\mathbf{W}}(\mathbf{x}_n) - \mathrm{LSE}_c \left( \boldsymbol{\mu}_c^{\top} \boldsymbol{\phi}_{\mathbf{W}}(\mathbf{x}_n) + \frac{1}{2} \boldsymbol{\phi}_{\mathbf{W}}(\mathbf{x}_n)^{\top} \mathbf{S}_c \boldsymbol{\phi}_{\mathbf{W}}(\mathbf{x}_n) \right) \right]$$

$$- \sum_{c=1}^{C} \left[ \frac{1}{2\sigma_{\theta}^2} \left( \mathrm{tr}(\mathbf{S}_c) + \boldsymbol{\mu}_c^{\top} \boldsymbol{\mu}_c \right) - \frac{1}{2} \log |\mathbf{S}_c| \right] + \mathrm{const} \qquad (10)$$

where $\boldsymbol{\Psi} := \{(\boldsymbol{\mu}_c, \mathbf{S}_c)\}_{c=1}^{C}$ collects the variational parameters and $\mathrm{LSE}_c(\cdot) = \log \sum_{c=1}^{C} \exp(\cdot)$ denotes the log-sum-exp function with sum over class $c$, which provides numerical stability when computing the logarithm of sums of exponentials.

Given a pre-trained LLM with weights $\mathbf{W}_0$, the fine-tuned weight parameterization is typically given by $\mathbf{W} := \mathbf{W}_0 + \Delta\mathbf{W}$. For PEFT, $\Delta\mathbf{W}$ is typically sought as a low-rank representation. However, standard LoRA-based approaches suffer from low *stable rank* during training, where the learned adapters collapse to suboptimal low-dimensional subspaces. To address this issue, we will adapt orthogonalized LoRA by leveraging the advances in Riemannian optimization.

## 3.2 FINE-TUNED LLM VIA ORTHOGONALIZED LORA

A powerful feature extractor is essential, not only because it provides more informative inputs to VBLL, but also because UQ is particularly valuable when the performance on the downstream task is already strong. While LoRA remains a popular approach, recent studies suggest that it cannot fully utilize the allocated rank, resulting in a significant gap relative to its full expressive potential. In

particular, standard LoRA often suffers from a low stable rank, which is a smooth proxy for matrix rank, during training, causing the learned adapters to collapse into suboptimal low-dimensional subspaces even if its rank $r$ is chosen large. This issue becomes more significant in the context of UQ for LLMs; see Fig.1(a). To this end, we advocate the use of recently developed orthogonalized low-rank adapters.

**Orthogonal Parametrization.** We leverage the PoLAR parameterization, where the additive weight for a particular layer $\Delta \mathbf{W} \in \mathbb{R}^{m \times n}$ is given by

$$\Delta \mathbf{W} = \mathbf{U} \boldsymbol{\Lambda} \mathbf{V}^\top. \tag{11}$$

Here, $\mathbf{U} \in \mathrm{St}(m, r)$, $\mathbf{V} \in \mathrm{St}(n, r)$, and $\boldsymbol{\Lambda} \in \mathbb{R}^{r \times r}$ is unconstrained for effective optimization with $\mathrm{St}(m, r) := \{\mathbf{M} \in \mathbb{R}^{m \times r} | \mathbf{M}^\top \mathbf{M} = \mathbf{I}_r\}$ denoting a Stiefel manifold, i.e., matrices with orthonormal columns. These orthogonality constraints effectively prevent rank collapse when optimized properly. Integrating PoLAR into the VBLL framework, we will adapt the ELBO objective (10) by setting $\mathbf{W} := \mathbf{W}_0 + \Delta \mathbf{W}$ with $\Delta \mathbf{W} = \mathbf{U} \boldsymbol{\Lambda} \mathbf{V}^\top$ parametrized by PoLAR. Thus, the resulting PoLAR-VBLL jointly seek the PoLAR parameters $\boldsymbol{\Psi}_{\mathrm{polar}} := \{\mathbf{U}, \boldsymbol{\Lambda}, \mathbf{V}\}$ and the variational parameters $\boldsymbol{\Psi}$ via

$$\{\hat{\boldsymbol{\Psi}}, \hat{\boldsymbol{\Psi}}_{\mathrm{polar}}\} = \underset{\boldsymbol{\Psi}, \boldsymbol{\Psi}_{\mathrm{polar}}}{\arg\max} \ \mathcal{L}^{\mathrm{ELBO}}(\boldsymbol{\Psi}, \boldsymbol{\Psi}_{\mathrm{polar}}; \mathcal{D}) \quad \text{s.to } \mathbf{U} \in \mathrm{St}(m, r), \mathbf{V} \in \mathrm{St}(n, r) \ . \tag{12}$$

**Scalable Optimization via Landing Fields.** To cope with the manifold constraints on $\mathbf{U}$ and $\mathbf{V}$, standard approaches rely on Riemannian optimization, which involves retraction operations. On Stiefel manifolds, these retractions require either SVD or QR factorization, making them impractical for large-scale models. This computational bottleneck can be alleviated using landing methods (Gao et al., 2022; Schechtman et al., 2023). For instance, optimizing $\mathbf{U}$ simply requires to replace its Euclidean gradient with the so-termed landing field:

$$\boldsymbol{\Gamma}(\mathbf{U}) = \boldsymbol{\psi}(\mathbf{U})\mathbf{U} + \lambda \nabla N(\mathbf{U}) \tag{13}$$

where $\boldsymbol{\psi}(\mathbf{U}) = \mathrm{Skew}(\nabla_{\mathbf{U}} \mathcal{L}(\mathbf{U}, \boldsymbol{\Lambda}, \mathbf{V})\mathbf{U}^\top)$ is the (generalized) Riemannian gradient component and $\nabla N(\mathbf{U}) = 4\mathbf{U}(\mathbf{U}^\top \mathbf{U} - \mathbf{I}_r)$ is the gradient of the infeasibility penalty $N(\mathbf{U}) = \|\mathbf{U}^\top \mathbf{U} - \mathbf{I}_r\|_F^2$. The parameter $\lambda > 0$ controls the strength of penalization for constraint violations. In other words, landing is an infeasible method, but with a properly chosen $\lambda$, the constraints are satisfied asymptotically at convergence. By avoiding costly SVD operations, this approach achieves a $3\times$ to $18\times$ speedup compared to retraction-based methods on GPUs, depending on the chosen rank.

The combination of orthogonal parameterization and scalable optimization yields theoretical benefits. Notably, PoLAR has been shown, under some assumptions, to converge faster as the rank $r$ increases, in stark contrast to LoRA (Lion et al., 2025). This improved scaling with $r$ enables the design of more expressive feature extractors tailored to available memory budgets, thereby justifying our adoption of PoLAR.

**Joint Optimization of PoLAR-VBLL.** To solve the optimization problem in (12), we will adopt alternating optimization, that consists of the following two steps per iteration.

- *Variational posterior update:* The gradients with respect to the variational parameters $\boldsymbol{\Psi}_c$ follow standard variational inference procedures (see Eqs. (21)-(22) in App. A.3);
- *PoLAR parameter update:* For the PoLAR parameters constrained to Stiefel manifolds, we employ landing field (cf. Eq. 13) to avoid expensive retraction operations. See App. A.3 (Eqs. (23)–(30)) for detailed derivation of the Riemannian gradients and updates.

The unified framework, together with infeasible Riemannian optimization for computational efficiency, yields a feature extractor that enhances both downstream performance and the reliability of uncertainty quantification.

### 3.3 UNCERTAINTY-AWARE PREDICTIVE INFERENCE

Having available the parameter estimates after training in PoLAR-VBLL, we are ready to predict for the label $y \in \{1, \dots, C\}$ for any given test input $\mathbf{x}$. Specifically, this predictive pdf is given by

$$p(y|\mathbf{x}, \mathcal{D}) = \int_{\boldsymbol{\Theta}} p(y|\boldsymbol{\Theta}, \mathbf{x}) q(\boldsymbol{\Theta}) d\boldsymbol{\Theta} \approx \frac{1}{K} \sum_{k=1}^{K} p(y|\mathbf{x}, \boldsymbol{\Theta}^{(k)}) \tag{14}$$

where we have employed Monte Carlo sampling to approximate the integral via $\boldsymbol{\Theta}^{(k)} \sim q(\boldsymbol{\Theta})$ and $p(y|\mathbf{x}, \boldsymbol{\Theta}^{(k)}) = \text{softmax}(\boldsymbol{\Theta}^{(k)}\boldsymbol{\phi}_{\hat{\mathbf{W}}}(\mathbf{x}))$ with $\hat{\mathbf{W}} = \mathbf{W}_0 + \hat{\mathbf{U}}\hat{\boldsymbol{\Lambda}}\hat{\mathbf{V}}^\top$.

While the PoLAR-VBLL framework provides an efficient method for end-to-end training, the ELBO objective (10), derived via Jensen's inequality, constitutes a tractable but possibly loose lower bound on the true log marginal likelihood. Maximizing this ELBO is effective for identifying a high-quality mode of the posterior—the estimated variational mean $\hat{\boldsymbol{\mu}}_c$—the resulting variational covariance $\hat{\mathbf{S}}_c$ may not perfectly capture the true posterior curvature. To alleviate this issue, we introduce an additional step via post-hoc LA to further refine the quality of the learned covariance matrix. Specifically, given the estimated PoLAR parameters $\hat{\boldsymbol{\Psi}}_{\text{polar}}$, we will evaluate the Hessian of $\log p(\mathcal{D}, \boldsymbol{\Theta}|\hat{\boldsymbol{\Psi}}_{\text{polar}})$ at the posterior mean $\{\hat{\boldsymbol{\mu}}_c\}_c$ as

$$\mathbf{H} = -\nabla_{\boldsymbol{\Theta}}^2 \left( \log p(\mathcal{D}|\boldsymbol{\Theta}, \hat{\boldsymbol{\Psi}}_{\text{polar}}) + \log p(\boldsymbol{\Theta}) \right) \Big|_{\boldsymbol{\Theta} = \{\boldsymbol{\mu}_c\}_c} \tag{15}$$

where $\log p(\mathcal{D}|\boldsymbol{\Theta}, \hat{\boldsymbol{\Psi}}_{\text{polar}})$ and $\log p(\boldsymbol{\Theta})$ are given by (4) and (2). Note that $\mathbf{H}$ is also the Bayesian Fisher information matrix of $\boldsymbol{\Theta}$, whose inverse $\boldsymbol{\Sigma} = \mathbf{H}^{-1}$, the Bayesian Cramer-Rao lower bound, can be taken as a covariance matrix for $\boldsymbol{\Theta}$. For the sake of tractability, we will still enforce a factorizable posterior over $\boldsymbol{\Theta}$, by ignoring the off-diagonal elements in $\boldsymbol{\Sigma}$. With $\boldsymbol{\Sigma}_c$ being the matrix on the diagonal of $\boldsymbol{\Sigma}$ corresponding to $\boldsymbol{\theta}_c$, the resulting corrected posterior is

$$\tilde{q}(\boldsymbol{\theta}_c|\mathcal{D}) = \mathcal{N}(\boldsymbol{\theta}_c; \hat{\boldsymbol{\mu}}_c, \boldsymbol{\Sigma}_c) \tag{16}$$

which will be used to make the prediction in (14).

**Remark.** Our strategy uses the scalable VBLL framework to first identify a high-quality mode $\hat{\boldsymbol{\mu}}_c$ along with the PoLAR parameters, and then applies LA as a 'finishing touch' to better characterize the posterior covariance around this well-chosen point. Notably, the post-hoc LA calibration does not affect the accuracy of the calibrated model (Yang et al., 2024). This hybrid approach nicely combines the strengths of variational training and post-hoc LA for enhanced uncertainty assessment. We have empirically validated the benefits of this additional step in our ablation studies, demonstrating improved performance on key UQ metrics such as calibration and out-of-distribution detection.

## 4 EXPERIMENTAL RESULTS

In this section, we compare our PoLAR-VBLL with existing methods on real-world datasets. We first introduce the experimental settings, including baselines, fine-tuning protocols, and evaluation procedures. We then evaluate PoLAR-VBLL's uncertainty estimation and generalization abilities in both in-distribution and out-of-distribution scenarios.

### 4.1 SETTINGS

**Fine-tuning and Evaluation.** We implement PoLAR-VBLL using the PEFT library (Mangrulkar et al., 2022) and fine-tune the `LlaMA2-7B` model (Touvron et al., 2023) on common-sense reasoning tasks. Additional evaluations on `LlaMA-3.1-8B` are delegated to the Appendices in the supplementary file due to space limitations; see Table 8. Following Laplace-LoRA (Yang et al., 2024) and BLOB (Wang et al., 2024), we apply PoLAR adapters (Lion et al., 2025) to the output layer as well as the queries and values of all attention layers. For the hyperparameters, we follow the default configurations outlined in the PEFT library (Mangrulkar et al., 2022) and the original PoLAR implementation (Lion et al., 2025) to guarantee the highest level of reproducibility. This encompasses aspects such as the total number of training steps, the learning rate, and the LoRA rank (see App. A.4 for further details). For fairness, the rank of the adapters in all the methods is set to $r = 8$ here.

For common-sense reasoning tasks, we cast them as a classification problem corresponding to possible answers from each dataset and fine-tune the LLM to maximize the ELBO objective in Eq. (10). For the classification head in our VBLL framework, we initialize the variational posterior means $\{\boldsymbol{\mu}_c\}$ using the pre-trained language model head weights corresponding to answer tokens (e.g., A, B, C, D for multiple-choice tasks), which significantly reduces training time by leveraging the model's pre-existing knowledge of answer formatting. For evaluation, in addition to Accuracy (ACC), we use Expected Calibration Error (ECE) (Naeini et al., 2015) and Negative Log-Likelihood (NLL) to assess the models' uncertainty estimation ability.

Table 1: Performances on ID datasets in terms of ACC, ECE, and NLL using `LlaMA2-7B`. Bold and underlined denote the best and the second-best performance, respectively.

| Metric | Method | Datasets | | | |
|--------|--------|-------|-------|-------|------|
| | | **WG-S** | **ARC-C** | **ARC-E** | **OBQA** |
| **ACC (%)** | MLE | 68.99±0.58 | 69.10±2.84 | 85.65±0.92 | 81.52±0.25 |
| | MAP | 68.62±0.71 | 67.59±0.40 | 86.55±0.55 | 81.38±0.65 |
| | MCD | 69.46±0.62 | 68.69±1.30 | 86.21±0.46 | 81.72±0.10 |
| | ENS | 69.57±0.66 | 66.20±2.01 | 84.40±0.81 | 81.38±0.91 |
| | BBB | 66.54±7.87 | 68.13±1.27 | 86.86±0.74 | 82.06±0.59 |
| | LA | 69.45±1.73 | 66.78±0.69 | 80.05±0.22 | 82.07±0.67 |
| | BLoB ($N$=10) | 69.07±0.34 | 68.81±1.09 | 86.56±0.35 | 81.52±0.74 |
| | **PoLAR-VBLL** | **71.62±0.27** | **70.92±0.24** | **88.03±0.44** | **82.53±0.12** |
| **ECE (%)** | MLE | 29.83±0.58 | 29.00±1.97 | 13.12±1.39 | 12.55±0.46 |
| | MAP | 29.76±0.87 | 29.42±0.68 | 12.07±0.55 | 13.26±0.82 |
| | MCD | 27.98±0.44 | 27.53±0.80 | 12.20±0.56 | 13.10±0.11 |
| | ENS | 28.52±0.55 | 29.16±2.37 | 12.57±0.58 | 15.34±0.27 |
| | BBB | 21.81±12.95 | 26.23±1.47 | 12.28±0.58 | 11.38±1.07 |
| | LA | 13.47±1.43 | 16.25±2.61 | 33.29±0.57 | 6.12±1.55 |
| | BLoB ($N$=10) | 9.35±1.37 | 9.59±1.88 | 3.64±0.53 | **3.77±1.47** |
| | **PoLAR-VBLL** | **7.31±0.32** | **7.41±0.78** | **2.63±0.81** | 4.63±1.43 |
| **NLL** | MLE | 3.17±0.37 | 2.85±0.27 | 1.17±0.13 | 0.73±0.03 |
| | MAP | 2.46±0.34 | 2.66±0.11 | 0.90±0.05 | 0.75±0.01 |
| | MCD | 2.79±0.53 | 2.67±0.15 | 1.00±0.14 | 0.77±0.03 |
| | ENS | 2.71±0.08 | 2.46±0.22 | 0.82±0.03 | 1.06±0.04 |
| | BBB | 1.40±0.55 | 2.23±0.04 | 0.91±0.06 | 0.66±0.05 |
| | LA | 0.67±0.01 | 1.03±0.04 | 0.88±0.00 | 0.72±0.01 |
| | BLoB ($N$=10) | 0.63±0.01 | **0.78±0.02** | **0.40±0.01** | **0.50±0.01** |
| | **PoLAR-VBLL** | **0.60±0.01** | 0.91±0.00 | 0.47±0.03 | 0.63±0.02 |

**Baselines and Implementation Details.** We compare PoLAR-VBLL with state-of-the-art approaches for UQ applied on top of LoRA fine-tuning, including Monte-Carlo Dropout (MCD) (Gal & Ghahramani, 2016), Bayes By Backprop (BBB) (Blundell et al., 2015; Xiong et al., 2023), Deep Ensemble (ENS) (Lakshminarayanan et al., 2017; Balabanov & Linander, 2024; Wang et al., 2023), and the latest Laplace-LoRA (LA) (Yang et al., 2024; Kristiadi et al., 2024). Additionally, we report the performance of two standard PEFT baseline methods: Maximum Likelihood Estimation (MLE) (Hu et al., 2022; Myung, 2003; Le Cam, 1990) and Maximum A Posteriori (MAP) (Greig et al., 1989). Comparisions with additional BLoB-based variants, including ScalaBL (Samplawski et al., 2025), C-LoRA (Rahmati et al., 2025), and TFB (Shi et al., 2024), are presented in the supplementary file; see Table 8.

We fine-tune a LlaMA2-7B model on four datasets requiring common-sense reasoning abilities, Winogrande-small (WG-S) (Sakaguchi et al., 2021), ARC-Challenge (ARC-C) (Clark et al., 2018), ARC-Easy (ARC-E) (Clark et al., 2018), OpenBookQA (OBQA) (Mihaylov et al., 2018), and additional chemistry (Chem) and physics (Phy) from the MMLU benchmark (Hendrycks et al., 2021a;b) for out-of-distribution evaluation. The datasets are split in the same manner as those in BLoB, and for each baseline, we report the better result between our reproduced numbers and those seen in BLoB. For all baseline methods, we utilize the same pre-trained LLM backbone and maintain consistent hyperparameters across all datasets.

## 4.2 RESULTS ON IN-DISTRIBUTION (ID) DATASETS

As shown in Table 1, our proposed PoLAR-VBLL demonstrates strong performance on the four ID commonsense reasoning tasks. Specifically, our approach attains the highest ACC on all evaluated datasets, while simultaneously achieving best or second-best performance in terms of ECE and NLL. This distinct our approach from BLoB, where a smaller $N$, i.e., number of samples at

Table 2: Performances on OOD datasets in terms of ACC, ECE, and NLL using `LlaMA2-7B`. Bold and underlined denote the best and the second-best performance, respectively.

| Metric | Method | Datasets | | | | |
| | | In-Dist. | Smaller Dist. Shift | | Larger Dist. Shift | |
| | | OBQA | ARC-C | ARC-E | Chem | Phy |
| --- | --- | --- | --- | --- | --- | --- |
| ACC (%) | MLE | 81.52±0.25 | 66.20±0.87 | 75.12±0.83 | 40.62±2.25 | 28.82±1.30 |
| | MAP | 81.38±0.91 | 69.59±0.23 | 75.47±0.73 | 44.79±0.00 | 28.47±1.20 |
| | MCD | 81.72±0.10 | 69.03±0.70 | 76.00±1.58 | 42.71±0.01 | 29.17±4.54 |
| | ENS | 81.38±0.65 | 67.34±0.70 | 75.18±2.03 | 43.75±1.04 | 30.56±2.62 |
| | BBB | 82.06±0.59 | 67.25±1.18 | 75.83±0.75 | 42.36±0.49 | 30.21±2.25 |
| | LA | 82.07±0.67 | 69.14±1.15 | 74.94±0.96 | 44.10±1.30 | 31.60±0.49 |
| | BLoB (N=10) | 81.52±0.74 | 67.71±1.13 | 76.37±0.80 | 44.79±1.47 | 31.60±2.73 |
| | **PoLAR-VBLL** | **82.53±0.12** | **70.07±0.48** | **81.24±0.78** | **45.67±0.58** | **33.33±0.98** |
| ECE (%) | MLE | 12.55±0.46 | 22.20±0.39 | 16.47±0.86 | 21.72±0.30 | 29.60±1.29 |
| | MAP | 15.34±0.27 | 19.31±0.46 | 15.68±0.51 | 17.55±1.95 | 30.25±2.18 |
| | MCD | 14.45±0.84 | 19.54±0.33 | 15.32±1.16 | 17.90±0.63 | 29.53±4.20 |
| | ENS | 13.26±0.82 | 7.59±1.43 | 6.44±0.83 | 12.04±4.57 | 17.52±1.28 |
| | BBB | 11.38±1.07 | 19.90±0.66 | 13.41±0.85 | 15.67±1.23 | 26.10±4.76 |
| | LA | 6.12±1.15 | 5.84±0.64 | 8.51±1.06 | 10.76±3.41 | 13.91±0.90 |
| | BLoB (N=10) | **3.77±1.47** | 9.55±0.40 | 5.48±1.27 | 9.77±1.35 | 18.29±1.35 |
| | **PoLAR-VBLL** | 4.63±1.43 | **5.12±0.90** | **5.09±0.77** | **6.49±2.07** | **6.03±2.21** |
| NLL | MLE | 0.73±0.03 | 1.16±0.00 | 0.92±0.03 | 1.56±0.06 | 1.66±0.05 |
| | MAP | 1.06±0.04 | 1.10±0.07 | 0.93±0.04 | 1.55±0.06 | 1.65±0.03 |
| | MCD | 1.06±0.08 | 1.08±0.00 | 0.88±0.03 | 1.59±0.07 | 1.67±0.05 |
| | ENS | 0.75±0.01 | 0.86±0.01 | 0.69±0.03 | **1.28±0.00** | 1.39±0.03 |
| | BBB | 0.66±0.05 | 1.06±0.01 | 0.79±0.02 | 1.49±0.05 | 1.62±0.06 |
| | LA | 0.72±0.01 | **0.81±0.00** | 0.70±0.02 | 1.35±0.03 | **1.36±0.01** |
| | BLoB (N=10) | **0.50±0.01** | 0.83±0.01 | **0.60±0.01** | 1.38±0.01 | 1.46±0.02 |
| | **PoLAR-VBLL** | 0.63±0.02 | 0.88±0.00 | 0.69±0.01 | 1.29±0.00 | **1.36±0.01** |

inference, gives a better ACC but with significantly worse ECE or NLL, and its ACC is still worse than our method in terms of ACC; see Table 3 in App. A.5 for more details. This simultaneous improvement in ACC, ECE, and NLL is particularly significant, as it addresses the prevalent over-confidence problem inherent in standard MLE fine-tuning approaches, and it validates the efficacy of our methodology in developing more reliable and well-calibrated models.

The performance gains can be attributed to the distinct yet synergistic roles of PoLAR and the VBLL inference scheme. The consistent accuracy improvements come with PoLAR, which mitigates the rank collapse in LoRA, thereby leading to a more expressive feature representation. The signifi-cant gains in calibration are primarily driven by the VBLL, which places a posterior distribution over the final layer's weights, explicitly modeling uncertainty and effectively reducing model over-confidence. The synergy between these components is critical, as the superior feature foundation provided by PoLAR enables VBLL to learn a more nuanced and reliable mapping from features to predictive distributions, ultimately resulting in a model that is simultaneously more accurate and better calibrated. The NLL and ECE metrics can be further decreased by increasing the number of training epochs.

## 4.3 RESULTS ON OUT-OF-DISTRIBUTION (OOD) DATASETS

To assess the robustness of our approach under distributional shifts, we fine-tune our model on OBQA and evaluate its performance across datasets with varying degrees of distribution mismatch. Since the in-distribution dataset OBQA consists of multiple-choice, elementary-level science ques-tions, we consider ARC-E and ARC-C to exhibit smaller distributional shifts. In comparison, the college-level chemistry and physics subsets of the MMLU benchmark represent larger distributional shifts.

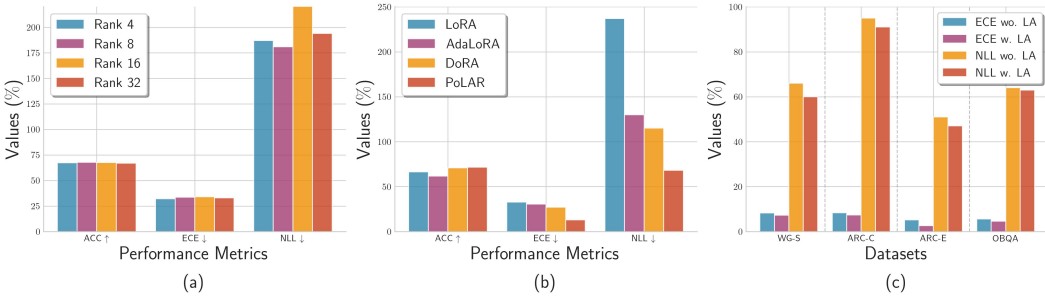

Figure 1: Ablation studies on the WG-S dataset using `LLaMA 2-7B`: (a) Performance of LoRA-VBLL using different ranks; (b) VBLL coupled with different adapters; and (c) ECE and NLL performances of PoLAR-VBLL with and without LA.

As presented in Table 2, our proposed method, PoLAR-VBLL, demonstrates exceptional OOD generalization capabilities. It consistently achieves the highest predictive accuracy across all evaluation settings, with particularly notable improvements in larger distribution shifts.

This superior performance indicates that our approach effectively captures generalizable representations that transfer well beyond the training distribution. From an uncertainty quantification perspective, PoLAR-VBLL exhibits remarkable calibration consistency across distributional shifts. The method achieves the best or second-best ECE performance across all out-of-distribution settings. Similarly, the NLL results demonstrate competitive performance, with our method maintaining reliable probabilistic predictions across varying shift magnitudes. These results underscore a critical finding: while many existing methods suffer from degraded uncertainty estimation under distribution shift, our approach maintains both high accuracy and well-calibrated uncertainty estimates. This dual robustness is essential for practical deployment, where models must not only perform well on shifted data but also provide trustworthy confidence indicators to enable appropriate decision-making in uncertain scenarios.

### 4.4 ABLATION STUDIES

We conduct ablation studies to validate each component of our PoLAR-VBLL framework. All three experiments are evaluated on the WG-S, with the same `LLaMA 2-7B` backbone. Our analysis of adapter rank in Figure 1(a), standard LoRA demonstrates minimal performance differences across various ranks, with performance remaining relatively flat regardless of rank size, confirming that rank collapse prevents effective utilization of larger rank allocations. To justify our choice of PoLAR, we benchmarked it against other PEFT methods from LoRA, adaLoRA, and DoRA. The results in Figure 1(b) are definitive: PoLAR achieves superior performance across accuracy, ECE, and NLL. Such empirical results show that PoLAR's more expressive feature representation provides a better foundation not only for the prediction task but also for subsequent uncertainty estimation within the VBLL framework. We also evaluated the post-hoc LA. As shown in Figure 1(c), applying LA after training consistently reduces both ECE and NLL across all tested datasets. This demonstrates that our framework identifies a high-quality posterior mode, allowing the LA step to effectively refine the covariance structure and further improve the final uncertainty estimates. Additionally, our VBLL approach achieves the lowest GPU memory consumption among competing UQ methods, providing computational advantages through analytical ELBO computation as shown in Table 6 in App. B.1. Lastly, an ablation study has been conducted to show the effects of VBLL and LA coupled with the same PoLAR adapter in Table 9 in App. B.3, where it is shown that PoLAR-VBLL outperforms PoLAR-LA and combining VBLL with post-hoc LA offers the maximal perforance gains.

### 5 CONCLUSIONS

This paper introduced PoLAR-VBLL, a scalable and unified framework for uncertainty-aware fine-tuning of LLMs. PoLAR addresses the rank collapse issue in conventional adapters through orthogonality constraints, yielding a more expressive feature extractor. Building on this foundation, VBLL enables efficient, sampling-free Bayesian training on the final layer for principled UQ. Extensive experiments demonstrate that PoLAR-VBLL consistently outperforms state-of-the-art baselines in both accuracy and uncertainty calibration. This work presents a principled and practical pathway towards developing more reliable and trustworthy fine-tuned LLMs for real-world applications.

## REPRODUCIBILITY STATEMENT

To ensure the reproducibility of our work, we provide all necessary implementation details. Our code is implemented using the PyTorch framework. A complete list of dependencies is provided in the Appendix A.4 and `requirements.txt` file in the project's root directory. All experiments were conducted on a server equipped with one NVIDIA A6000 ada (48GB) GPU. A single full run of our main experiment requires approximately 72 hours ( 500 epochs) to complete. We will publicly release all source code, model weights, and scripts used to generate key figures, accompanied by a detailed `README.md` file that includes step-by-step instructions for environment setup, data preprocessing, and model training/evaluation. The code will be made available in a GitHub repository upon acceptance. The datasets used in our study are publicly available, and the specific preprocessing steps are detailed in Appendix A.4 .

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

# A APPENDIX

USAGE OF LLMS

During the preparation of this paper, we utilized large language models (LLMs), such as OpenAI's ChatGPT, only for the purpose of improving typos, grammar, and clarity. For instance, we utilized ChatGPT to proofread and refine the phrasing of specific sentences, as well as to verify spelling and grammatical accuracy. However, all core research ideas, experimental designs, result analysis, and the final narrative structure of the paper were exclusively conceived and formulated by the authors. No LLM was used to generate any of the core scientific content of this work, such as algorithm design, theoretical derivations, or experimental results.

## A.1 COMPUTATIONAL COMPLEXITY ANALYSIS

We provide a detailed computational complexity analysis for each phase of our PoLAR-VBLL framework, demonstrating its efficiency compared to alternative approaches for uncertainty quantification in fine-tuned LLMs.

### A.1.1 TRAINING PHASE COMPLEXITY

The joint optimization of PoLAR adapter parameters $\mathbf{\Psi}_{\text{polar}} := \{\mathbf{U}, \mathbf{\Lambda}, \mathbf{V}\}$ and variational parameters $\mathbf{\Psi} = \{\boldsymbol{\mu}_1, \ldots, \boldsymbol{\mu}_C, \mathbf{S}_1, \ldots, \mathbf{S}_C\}$ involves the following computational costs per training iteration:

**Feature Extraction:** Computing LLM features $\phi_{\mathbf{W}}(\mathbf{x}_i)$ where $\mathbf{W} = \mathbf{W}_0 + \mathbf{U}\mathbf{\Lambda}\mathbf{V}^\top$ for a batch of size $B$ requires $\mathcal{O}(B \cdot \text{LLM}_{\text{cost}})$ operations, where $\text{LLM}_{\text{cost}}$ represents the computational cost of a single forward pass through the base language model.

**PoLAR Parameter Updates:** The landing field optimization on Stiefel manifolds incurs:

- Euclidean Gradient Calculation: $\mathcal{O}(mnr)$
- Riemannian gradient computation: $\mathcal{O}(m^2 r + n^2 r)$ for skew-symmetric operations $\boldsymbol{\psi}(\mathbf{U})$ and $\boldsymbol{\psi}(\mathbf{V})$
- Constraint gradient computation: $\mathcal{O}((m + n)r^2)$ for infeasibility penalties $\nabla N(\mathbf{U}) = 4\mathbf{U}(\mathbf{U}^\top\mathbf{U} - \mathbf{I}_r)$ and $\nabla N(\mathbf{V})$
- Parameter updates: $\mathcal{O}(r(m + n + r))$ for $\mathbf{U}, \mathbf{V}$, and $\mathbf{\Lambda}$ updates via landing field method

**VBLL Parameter Updates:** Variational inference optimization requires:

- ELBO computation: $\mathcal{O}(B \cdot C \cdot d)$ for likelihood terms and log-sum-exp operations in Eq. 12
- KL divergence computation: $\mathcal{O}(C \cdot d^2)$ for trace and determinant operations in covariance matrices $\mathbf{S}_c$
- Gradient computation: $\mathcal{O}(C \cdot d^2)$ for gradients with respect to variational means $\boldsymbol{\mu}_c$ and covariances $\mathbf{S}_c$
- Parameter updates: $\mathcal{O}(B \cdot C \cdot d^2)$ for updating $C$ class-specific posterior distributions

The total complexity per training iteration is:

$$\mathcal{O}\left(B \cdot \text{LLM}_{\text{cost}} + r(m^2 + n^2) + B \cdot C \cdot d^2 + r(m + n + r)\right) \tag{17}$$

For $T$ training iterations, the overall training complexity becomes:

$$\mathcal{O}\left(T \cdot \left(B \cdot \text{LLM}_{\text{cost}} + r(m^2 + n^2) + B \cdot C \cdot d^2 + r(m + n + r)\right)\right) \tag{18}$$

### A.1.2 PREDICTIVE INFERENCE COMPLEXITY

The uncertainty-aware prediction phase involves:

**Optional Laplace Calibration:** Computing the Hessian of the negative log-likelihood for posterior refinement requires $\mathcal{O}(C \cdot d^2)$ operations using Kronecker-factored approximation (KFAC), which is significantly more efficient than the naive $\mathcal{O}((C \cdot d)^2)$ full Hessian computation.

**Monte Carlo Sampling:** For $K$ posterior samples from $q(\boldsymbol{\Theta})$:

- Parameter sampling: $\mathcal{O}(K \cdot C \cdot d^2)$ for sampling from multivariate Gaussians $\mathcal{N}(\boldsymbol{\mu}_c, \mathbf{S}_c)$
- Forward computation: $\mathcal{O}(K \cdot C \cdot d)$ for logit computation $\mathbf{z} = \boldsymbol{\Theta}\boldsymbol{\phi}_{\mathbf{W}}(\mathbf{x}^*)$ and softmax normalization

Total inference complexity per test point:

$$\mathcal{O}\left(\text{LLM}_{\text{cost}} + C \cdot d^2 + K \cdot C \cdot d^2\right) \tag{19}$$

### A.1.3 COMPARISON WITH BASELINE METHODS

**vs. Standard LoRA:** Our PoLAR parameterization adds $\mathcal{O}(r^2)$ overhead per update compared to LoRA's $\mathcal{O}(r)$ due to orthogonality constraints, but this is negligible when $r \ll d$ while providing substantially improved stable rank utilization.

**vs. BLoB:** BLoB requires expensive Monte Carlo sampling during training with $\mathcal{O}(N_{\text{MC}} \cdot \text{LLM}_{\text{cost}})$ cost per ELBO evaluation, where $N_{\text{MC}}$ is the number of Monte Carlo samples. Our VBLL approach achieves analytical ELBO computation, eliminating this sampling overhead.

**vs. Ensemble Methods:** Maintaining $M$ separate adapter copies requires $\mathcal{O}(M \cdot r(m+n))$ storage and $\mathcal{O}(M \cdot \text{LLM}_{\text{cost}})$ inference time. Our Bayesian approach achieves comparable uncertainty quality with $\mathcal{O}(C \cdot d^2)$ additional parameters.

**vs. Laplace-LoRA:** Post-hoc Laplace approximation around suboptimal MAP estimates requires similar Hessian computation but lacks the joint optimization benefits of our integrated approach.

### A.1.4 MEMORY COMPLEXITY

The space complexity of our framework is:

$$\mathcal{O}\left(|\mathbf{W}_0| + r(m+n+r) + C \cdot d^2 + B \cdot d\right) \tag{20}$$

where $|\mathbf{W}_0|$ represents the frozen pre-trained model size, $r(m+n+r)$ accounts for PoLAR parameters, $C \cdot d^2$ stores VBLL covariance matrices, and $B \cdot d$ handles intermediate feature storage during batch processing.

## A.2 DETAILED ALGORITHM SPECIFICATIONS

## A.3 DETAILED GRADIENT DERIVATIONS

This section provides the complete derivation of gradient updates for the joint PoLAR-VBLL optimization procedure described in Section 3.2.

### A.3.1 VARIATIONAL PARAMETER UPDATES

The gradients with respect to the variational parameters $\{\boldsymbol{\mu}_c, \mathbf{S}_c\}$ follow standard variational inference procedures:

$$\frac{\partial \mathcal{L}^{\text{ELBO}}}{\partial \boldsymbol{\mu}_c} = \frac{1}{|\mathcal{D}|} \sum_{i=1}^{|\mathcal{D}|} \left[ y_{i,c} \boldsymbol{\phi}_{\mathbf{W}}(\mathbf{x}_i) - \frac{\exp(\boldsymbol{\mu}_c^\top \boldsymbol{\phi}_{\mathbf{W}}(\mathbf{x}_i) + \frac{1}{2}\boldsymbol{\phi}_{\mathbf{W}}(\mathbf{x}_i)^\top \mathbf{S}_c \boldsymbol{\phi}_{\mathbf{W}}(\mathbf{x}_i))}{\sum_{j=1}^{C} \exp(\boldsymbol{\mu}_j^\top \boldsymbol{\phi}_{\mathbf{W}}(\mathbf{x}_i) + \frac{1}{2}\boldsymbol{\phi}_{\mathbf{W}}(\mathbf{x}_i)^\top \mathbf{S}_j \boldsymbol{\phi}_{\mathbf{W}}(\mathbf{x}_i))} \boldsymbol{\phi}_{\mathbf{W}}(\mathbf{x}_i) \right]$$
$$- \frac{1}{\sigma_\theta^2} \boldsymbol{\mu}_c \tag{21}$$

$$\frac{\partial \mathcal{L}^{\text{ELBO}}}{\partial \mathbf{S}_c} = \frac{1}{2|\mathcal{D}|} \sum_{i=1}^{|\mathcal{D}|} \left[ -\frac{\exp(\boldsymbol{\mu}_c^\top \boldsymbol{\phi}_{\mathbf{W}}(\mathbf{x}_i) + \frac{1}{2}\boldsymbol{\phi}_{\mathbf{W}}(\mathbf{x}_i)^\top \mathbf{S}_c \boldsymbol{\phi}_{\mathbf{W}}(\mathbf{x}_i))}{\sum_{j=1}^{C} \exp(\boldsymbol{\mu}_j^\top \boldsymbol{\phi}_{\mathbf{W}}(\mathbf{x}_i) + \frac{1}{2}\boldsymbol{\phi}_{\mathbf{W}}(\mathbf{x}_i)^\top \mathbf{S}_j \boldsymbol{\phi}_{\mathbf{W}}(\mathbf{x}_i))} \boldsymbol{\phi}_{\mathbf{W}}(\mathbf{x}_i) \boldsymbol{\phi}_{\mathbf{W}}(\mathbf{x}_i)^\top \right]$$
$$- \frac{1}{2\sigma_\theta^2} \mathbf{I}_d + \frac{1}{2} \mathbf{S}_c^{-1} \tag{22}$$

---

**Algorithm 1** PoLAR-VBLL Training

---

**Require:** Pre-trained LLM weights $\mathbf{W}_0$, training dataset $\mathcal{D} = \{(\mathbf{x}_i, \mathbf{y_i})\}_{\mathbf{i=1}}^{|\mathcal{N}|}$, rank $r$, hyperparameters $\{\eta_{\text{polar}}, \eta_{\text{vbll}}, \lambda, \sigma_\theta^2\}$

**Ensure:** Converged PoLAR parameters $\{\hat{\mathbf{U}}, \hat{\mathbf{\Lambda}}, \hat{\mathbf{V}}\}$, variational posterior $q(\mathbf{\Theta})$
  1: Initialize $\mathbf{U}_0 \in \text{St}(m, r)$, $\mathbf{V}_0 \in \text{St}(n, r)$ via QR decomposition of random matrices
  2: Initialize $\mathbf{\Lambda}_0 \sim \mathcal{N}(\mathbf{0}, 0.01^2 \mathbf{I}_{r \times r})$
  3: Initialize VBLL parameters: $\boldsymbol{\mu}_{c,0} = \mathbf{w}_{\text{pretrain},c}$, $\mathbf{S}_{c,0} = \sigma_\theta^2 \mathbf{I}_d$ for $c = 1, \ldots, C$
  4: **for** $t = 0, 1, \ldots, T - 1$ **do**
  5:   Sample mini-batch $\mathcal{B}_t \subset \mathcal{D}$ of size $B$
  6:   Extract features: $\boldsymbol{\phi}_t(\mathbf{x}_i) = \boldsymbol{\phi}_{\mathbf{W}_0 + \mathbf{U}_t \mathbf{\Lambda}_t \mathbf{V}_t^\top}(\mathbf{x}_i)$ for all $\mathbf{x}_i \in \mathcal{B}_t$
  7:   Compute ELBO: $\mathcal{L}_t^{\text{ELBO}}(\mathbf{\Psi}_{\text{polar}}, \mathbf{\Psi}; \mathcal{B}_t)$ using Eq. 12
  8:   // PoLAR parameter updates via landing field method
  9:   Compute weight gradient: $\mathbf{G}_t = \frac{\partial \mathcal{L}_t^{\text{ELBO}}}{\partial (\mathbf{U}_t \mathbf{\Lambda}_t \mathbf{V}_t^\top)}$
 10:   Compute factor gradients:
 11:     $\nabla_{\mathbf{\Lambda}} \mathcal{L}_t^{\text{ELBO}} = \mathbf{U}_t^\top \mathbf{G}_t \mathbf{V}_t$
 12:     $\nabla_{\mathbf{U}} \mathcal{L}_t^{\text{ELBO}} = \mathbf{G}_t \mathbf{V}_t \mathbf{\Lambda}_t^\top$
 13:     $\nabla_{\mathbf{V}} \mathcal{L}_t^{\text{ELBO}} = \mathbf{G}_t^\top \mathbf{U}_t \mathbf{\Lambda}_t$
 14:   Compute Riemannian gradients:
 15:     $\boldsymbol{\psi}(\mathbf{U}_t) = \text{Skew}(\nabla_{\mathbf{U}} \mathcal{L}_t^{\text{ELBO}} \cdot \mathbf{U}_t^\top)$
 16:     $\boldsymbol{\psi}(\mathbf{V}_t) = \text{Skew}(\nabla_{\mathbf{V}} \mathcal{L}_t^{\text{ELBO}} \cdot \mathbf{V}_t^\top)$
 17:   Landing field updates:
 18:     $\mathbf{\Gamma}(\mathbf{U}_t) = \boldsymbol{\psi}(\mathbf{U}_t) \mathbf{U}_t + \lambda \cdot 4\mathbf{U}_t(\mathbf{U}_t^\top \mathbf{U}_t - \mathbf{I}_r)$
 19:     $\mathbf{\Gamma}(\mathbf{V}_t) = \boldsymbol{\psi}(\mathbf{V}_t) \mathbf{V}_t + \lambda \cdot 4\mathbf{V}_t(\mathbf{V}_t^\top \mathbf{V}_t - \mathbf{I}_r)$
 20:   Update PoLAR parameters:
 21:     $\mathbf{U}_{t+1} = \mathbf{U}_t - \eta_{\text{polar}} \mathbf{\Gamma}(\mathbf{U}_t)$
 22:     $\mathbf{V}_{t+1} = \mathbf{V}_t - \eta_{\text{polar}} \mathbf{\Gamma}(\mathbf{V}_t)$
 23:     $\mathbf{\Lambda}_{t+1} = \mathbf{\Lambda}_t - \eta_{\text{polar}} \nabla_{\mathbf{\Lambda}} \mathcal{L}_t^{\text{ELBO}}$
 24:   // VBLL parameter updates
 25:   **for** $c = 1, \ldots, C$ **do**
 26:     Compute variational gradients using Eqs. 21–22
 27:     $\boldsymbol{\mu}_{c,t+1} = \boldsymbol{\mu}_{c,t} - \eta_{\text{vbll}} \frac{\partial \mathcal{L}_t^{\text{ELBO}}}{\partial \boldsymbol{\mu}_c}$
 28:     $\mathbf{S}_{c,t+1} = \mathbf{S}_{c,t} - \eta_{\text{vbll}} \frac{\partial \mathcal{L}_t^{\text{ELBO}}}{\partial \mathbf{S}_c}$
 29:     Project $\mathbf{S}_{c,t+1}$ to positive definite cone if necessary
 30:   **end for**
 31: **end for**
 32: **Return** $\hat{\mathbf{U}} = \mathbf{U}_T$, $\hat{\mathbf{\Lambda}} = \mathbf{\Lambda}_T$, $\hat{\mathbf{V}} = \mathbf{V}_T$, $q(\mathbf{\Theta}) = \prod_{c=1}^{C} \mathcal{N}(\boldsymbol{\theta}_c; \boldsymbol{\mu}_{c,T}, \mathbf{S}_{c,T})$

---

### A.3.2 PoLAR Parameter Updates

For the PoLAR parameters, we employ the chain rule to propagate gradients through the feature extractor $\boldsymbol{\phi}_{\mathbf{W}}(\mathbf{x})$ where $\mathbf{W} = \mathbf{W}_0 + \mathbf{U}\mathbf{\Lambda}\mathbf{V}^\top$. Let $\mathbf{G} := \frac{\partial \mathcal{L}^{\text{ELBO}}}{\partial (\mathbf{U}\mathbf{\Lambda}\mathbf{V}^\top)}$ denote the gradient with respect to the weight update. Then:

$$\frac{\partial \mathcal{L}^{\text{ELBO}}}{\partial \mathbf{\Lambda}} = \mathbf{U}^\top \mathbf{G} \mathbf{V} \tag{23}$$

$$\nabla_{\mathbf{U}} \mathcal{L}^{\text{ELBO}} = \mathbf{G} \mathbf{V} \mathbf{\Lambda}^\top \tag{24}$$

$$\nabla_{\mathbf{V}} \mathcal{L}^{\text{ELBO}} = \mathbf{G}^\top \mathbf{U} \mathbf{\Lambda} \tag{25}$$

### A.3.3 Riemannian Gradient Computation

Since $\mathbf{U}$ and $\mathbf{V}$ are constrained to Stiefel manifolds, we convert the Euclidean gradients to their Riemannian counterparts. For a matrix $\mathbf{X} \in \text{St}(m, r)$, the Riemannian gradient is:

---

**Algorithm 2** PoLAR-VBLL Uncertainty-Aware Prediction

---

**Require:** Converged PoLAR parameters $\{\hat{\mathbf{U}}, \hat{\mathbf{\Lambda}}, \hat{\mathbf{V}}\}$, variational posterior $q(\mathbf{\Theta})$, test input $\mathbf{x}^*$, training data $\mathcal{D}$, number of samples $K$, Laplace refinement flag
**Ensure:** Predictive distribution $p(y^*|\mathbf{x}^*, \mathcal{D})$
 1: **if** Laplace refinement enabled **then**
 2:     // Optional posterior refinement via Laplace approximation
 3:     Compute converged means: $\boldsymbol{\mu}_c^* = \boldsymbol{\mu}_{c,T}$ for $c = 1, \ldots, C$
 4:     Compute Hessian using KFAC approximation:
 5:        $\mathbf{H}_c = -\nabla_{\boldsymbol{\theta}_c}^2 \log p(\mathcal{D}|\boldsymbol{\theta}_c = \boldsymbol{\mu}_c^*, \hat{\mathbf{U}}, \hat{\mathbf{\Lambda}}, \hat{\mathbf{V}})$
 6:     Form Laplace posterior: $q_{\text{Lap}}(\boldsymbol{\theta}_c) = \mathcal{N}(\boldsymbol{\mu}_c^*, \mathbf{H}_c^{-1})$ for $c = 1, \ldots, C$
 7: **end if**
 8: // Extract test features
 9: $\boldsymbol{\phi}^* = \boldsymbol{\phi}_{\mathbf{W}_0 + \hat{\mathbf{U}}\hat{\mathbf{\Lambda}}\hat{\mathbf{V}}^\top}(\mathbf{x}^*)$
10: // Monte Carlo sampling for predictive distribution
11: Initialize prediction accumulator: $\mathbf{p}_{\text{pred}} = \mathbf{0}_C$
12: **for** $k = 1, \ldots, K$ **do**
13:     **if** Laplace refinement enabled **then**
14:        Sample classification weights: $\boldsymbol{\theta}_c^{(k)} \sim q_{\text{Lap}}(\boldsymbol{\theta}_c)$ for $c = 1, \ldots, C$
15:     **else**
16:        Sample classification weights: $\boldsymbol{\theta}_c^{(k)} \sim q(\boldsymbol{\theta}_c) = \mathcal{N}(\boldsymbol{\mu}_{c,T}, \mathbf{S}_{c,T})$ for $c = 1, \ldots, C$
17:     **end if**
18:     Form weight matrix: $\mathbf{\Theta}^{(k)} = [\boldsymbol{\theta}_1^{(k)}, \ldots, \boldsymbol{\theta}_C^{(k)}]^\top$
19:     Compute logits: $\mathbf{z}^{(k)} = \mathbf{\Theta}^{(k)}\boldsymbol{\phi}^*$
20:     Compute sample prediction: $\mathbf{p}^{(k)} = \text{softmax}(\mathbf{z}^{(k)})$
21:     Accumulate: $\mathbf{p}_{\text{pred}} = \mathbf{p}_{\text{pred}} + \mathbf{p}^{(k)}$
22: **end for**
23: Average predictions: $p(y^*|\mathbf{x}^*, \mathcal{D}) = \frac{1}{K}\mathbf{p}_{\text{pred}}$
24: **Return** Predictive distribution $p(y^*|\mathbf{x}^*, \mathcal{D})$

---

$$\text{grad}_R f(\mathbf{X}) = \nabla f(\mathbf{X}) - \mathbf{X}\nabla f(\mathbf{X})^\top \mathbf{X} \tag{26}$$

Applying this to our PoLAR parameters:

$$\boldsymbol{\psi}(\mathbf{U}) = \text{Skew}(\nabla_{\mathbf{U}}\mathcal{L}^{\text{ELBO}} \cdot \mathbf{U}^\top) = \text{Skew}(\mathbf{G}\mathbf{V}\mathbf{\Lambda}^\top\mathbf{U}^\top) \tag{27}$$

$$\boldsymbol{\psi}(\mathbf{V}) = \text{Skew}(\nabla_{\mathbf{V}}\mathcal{L}^{\text{ELBO}} \cdot \mathbf{V}^\top) = \text{Skew}(\mathbf{G}^\top\mathbf{U}\mathbf{\Lambda}\mathbf{V}^\top) \tag{28}$$

where $\text{Skew}(\mathbf{A}) = \frac{1}{2}(\mathbf{A} - \mathbf{A}^\top)$ extracts the skew-symmetric component.

### A.3.4 LANDING FIELD UPDATES

Following the infeasible optimization approach, we replace the expensive retraction operations with landing field updates:

$$\mathbf{\Gamma}(\mathbf{U}) = \boldsymbol{\psi}(\mathbf{U})\mathbf{U} + \lambda\nabla N(\mathbf{U}) \tag{29}$$

$$\mathbf{\Gamma}(\mathbf{V}) = \boldsymbol{\psi}(\mathbf{V})\mathbf{V} + \lambda\nabla N(\mathbf{V}) \tag{30}$$

where $\nabla N(\mathbf{U}) = 4\mathbf{U}(\mathbf{U}^\top\mathbf{U} - \mathbf{I}_r)$ and $\nabla N(\mathbf{V}) = 4\mathbf{V}(\mathbf{V}^\top\mathbf{V} - \mathbf{I}_r)$ are the gradients of the infeasibility penalties $N(\mathbf{U}) = \|\mathbf{U}^\top\mathbf{U} - \mathbf{I}_r\|_F^2$ and $N(\mathbf{V}) = \|\mathbf{V}^\top\mathbf{V} - \mathbf{I}_r\|_F^2$, respectively.

The complete update procedure alternates between updating the variational parameters using standard gradient-based optimizers (e.g., Adam) on Eqs. 21–22, and updating the PoLAR parameters using the landing field approach on Eqs. 23, 29, and 30.

## A.4 Implementation Details

### A.4.1 Training Settings

**Model Architecture.** Our implementation builds upon the `LLaMA-2-7B` foundation model (Touvron et al., 2023), utilizing its pre-trained language modeling head for VBLL mean initialization.

**PoLAR Configuration.** The manifold penalty coefficient in PoLAR $\lambda = 1.0$. We parameterize the $\mathbf{S}$ matrix using the identity initialization and apply Landing Field optimization (Lion et al., 2025; Gao et al., 2022; Schechtman et al., 2023) with gradient type set to "landing". The Landing Field callback is enabled during training to maintain stability in optimization on the Grassmann manifold.

**VBLL Parameterization.** For VBLL, we adopt the dense parameterization for computational efficiency while maintaining uncertainty quantification capabilities. The Jensen bound is used for approximating the softmax function. Prior hyperparameters are set as follows: prior scale $\sigma_0^2 = 1.0$, Wishart scale $\nu_0 = 10^{-2}$, degrees of freedom $\rho = 1.0$. The regularization weight for KL divergence is computed as $\lambda_{\text{reg}} = 1/|\mathcal{D}_{\text{train}}|$ where $|\mathcal{D}_{\text{train}}|$ is the training set size. This regularization weight can be used to adjust the emphasis of model performance on ACC or Uncertainty Quantification ability. All parameter values are the default classification setting in the VBLL library (Harrison et al., 2024). For the standard training process, we employ a two-step training approach. The $\lambda_{\text{reg}}$ is first set to $1/\mathcal{D}$ for the ACC increasing. After training for a couple of epochs, increasing $\lambda_{\text{reg}}$ further will largely suppress both NLL and ECE metrics.

**Training Configuration.** For all shared parameters, we follow the setting of BLoB's official single-GPU scripts, except for LoRA Rank and LoRA Alpha, to ease BNN training and improve performance. We train all methods (PoLAR-VBLL and baselines) for 500 epochs with a batch size of 4, evaluation batch size of 8, and maximum sequence length of 300 tokens. All methods use AdamW (Loshchilov & Hutter, 2017) optimizers with learning rate $10^{-4}$ and a CosineAnnealing-WarmRestarts scheduler (Loshchilov & Hutter, 2016). Baselines are reproduced strictly following the implementations from their official repositories. For sampling-based methods (BLoB, TFB, ScalaBL, C-LoRA), we set training sampling $K_{\text{train}} = 1$ (single sample per forward pass) and inference sampling $K_{\text{eval}} = 10$. LoRA/PoLAR rank ($r = 16$), alpha ($\alpha = 32$), and dropout (0.1). All training is conducted in BF16 precision on CUDA devices. For all MC-based uncertainty quantification evaluations, we use $n_{\text{samples}} = 10$.

**LA Calibration.** For post-hoc calibration, we apply LA with a diagonal Hessian structure over all model parameters. The prior precision is set to 1.0.

### A.4.2 Computational Environment

**Hardware Specifications** All experiments are conducted on a high-performance computing system equipped with NVIDIA RTX A6000 Ada GPUs and AMD 9600 Threadripper processors with 64 cores and 128 threads. This configuration provides substantial computational resources for both GPU-accelerated training and CPU-intensive operations such as Hessian computation for Laplace approximation.

**Software Dependencies.** Our implementation leverages several key Python packages: PyTorch (Paszke, 2019) for deep learning operations, HuggingFace PEFT (Mangrulkar et al., 2022) for adapter implementations, custom Laplace approximation libraries (Yang et al., 2024; Daxberger et al., 2021; Kristiadi et al., 2024) for post-hoc uncertainty calibration, PoLAR optimization libraries (Lion et al., 2025), and VBLL (Variational Bayesian Last Layer) implementations (Harrison et al., 2024). Complete dependency specifications and version information are provided in our `requirements.txt` file, which will be made available upon acceptance.

Table 3: Performances on ID datasets in terms of ACC, ECE, and NLL using `LlaMA2-7B`. Bold and underlined denote the best and the second-best performance, respectively. Here, we include PoLAR-VBLL with and without LA.

| Metric | Method | Datasets | | | |
|--------|--------|------|-------|-------|------|
| | | WG-S | ARC-C | ARC-E | OBQA |
| ACC (%) | MLE | 68.99±0.58 | 69.10±2.84 | 85.65±0.92 | 81.52±0.25 |
| | MAP | 68.62±0.71 | 67.59±0.40 | 86.55±0.55 | 81.38±0.65 |
| | MCD | 69.46±0.62 | 68.69±1.30 | 86.21±0.46 | 81.72±0.10 |
| | ENS | 69.57±0.66 | 66.20±2.01 | 84.40±0.81 | 81.38±0.91 |
| | BBB | 66.54±7.87 | 68.13±1.27 | 86.86±0.74 | 82.06±0.59 |
| | LA | 69.45±1.73 | 66.78±0.69 | 80.05±0.22 | 82.07±0.67 |
| | BLoB (N=0) | 70.89±0.82 | 70.83±1.57 | 86.68±0.60 | 82.73±0.41 |
| | PoLAR-VBLL (wo. LA) | **71.62±0.27** | **70.92±0.24** | **88.03±0.44** | **82.53±0.12** |
| | **PoLAR-VBLL** | **71.62±0.27** | **70.92±0.24** | **88.03±0.44** | **82.53±0.12** |
| ECE (%) | MLE | 29.83±0.58 | 29.00±1.97 | 13.12±1.39 | 12.55±0.46 |
| | MAP | 29.76±0.87 | 29.42±0.68 | 12.07±0.55 | 13.26±0.82 |
| | MCD | 27.98±0.44 | 27.53±0.80 | 12.20±0.56 | 13.10±0.11 |
| | ENS | 28.52±0.55 | 29.16±2.37 | 12.57±0.58 | 15.34±0.27 |
| | BBB | 21.81±12.95 | 26.23±1.47 | 12.28±0.58 | 11.38±1.07 |
| | LA | 13.47±1.43 | 16.25±2.61 | 33.29±0.57 | 6.12±1.55 |
| | BLoB (N=0) | 20.62±0.83 | 20.61±1.16 | 9.43±0.38 | 8.36±0.38 |
| | PoLAR-VBLL (wo. LA) | 8.26±0.60 | 8.36±0.13 | 5.22±0.41 | 5.58±0.34 |
| | **PoLAR-VBLL** | **7.31±0.32** | **7.41±0.78** | **2.63±0.81** | **4.63±1.43** |
| NLL | MLE | 3.17±0.37 | 2.85±0.27 | 1.17±0.13 | 0.73±0.03 |
| | MAP | 2.46±0.34 | 2.66±0.11 | 0.90±0.05 | 0.75±0.01 |
| | MCD | 2.79±0.53 | 2.67±0.15 | 1.00±0.14 | 0.77±0.03 |
| | ENS | 2.71±0.08 | 2.46±0.22 | 0.82±0.03 | 1.06±0.04 |
| | BBB | 1.40±0.55 | 2.23±0.04 | 0.91±0.06 | 0.66±0.05 |
| | LA [116] | 0.67±0.01 | 1.03±0.04 | 0.88±0.00 | 0.72±0.01 |
| | BLoB (N=0) | 0.91±0.10 | 1.19±0.02 | 0.56±0.01 | **0.56±0.02** |
| | PoLAR-VBLL (wo. LA) | 0.66±0.03 | 0.95±0.07 | 0.51±0.03 | 0.64±0.01 |
| | **PoLAR-VBLL** | **0.60±0.01** | **0.91±0.00** | **0.47±0.03** | 0.63±0.02 |

## A.5 ADDITIONAL EXPERIMENTAL RESULTS

### A.5.1 FULL COMPARISON ON IN-DISTRIBUTION DATASETS

Table 3 provides a comprehensive comparison. BLoB exhibits a trade-off between accuracy and uncertainty quantification: at N=0 sampling, it achieves higher accuracy performance but with reduced calibration quality. Even under these optimal accuracy conditions, BLoB achieves lower accuracy than our method across all datasets and shows higher ECE and NLL values. As sampling increases, BLoB exhibits improved uncertainty metrics, albeit with a corresponding reduction in predictive accuracy.

Our PoLAR-VBLL framework achieves competitive accuracy across all datasets while maintaining strong uncertainty calibration. The framework demonstrates that it is possible to obtain both high predictive performance and well-calibrated uncertainties without requiring the typical trade-off. The additional LA refinement further enhances our method's ECE and NLL performance while maintaining accuracy, suggesting that our variational training provides a robust foundation for posterior refinement. These results suggest that the combination of PoLAR's enhanced feature representation and VBLL's principled uncertainty quantification offers a promising approach for achieving both accuracy and calibration in uncertainty-aware models.

## A.6 TRAINING PROTOCOL: EXTENDED TRAINING SCHEDULE

**Motivation for Extended Training.** A natural question arises regarding our choice of training schedule, particularly given that some baseline methods specify shorter training durations in their original implementations. We address this by examining the BLoB baseline as a representative case study on WinoGrande-Simple (WG-S).

**Convergence Analysis of BLoB Official Implementation.** We strictly followed the official BLoB repository[1] and executed their provided shell script (`blob-llama-all-single-gpu.sh`) without modification. For WG-S, the official configuration specifies:

- Batch size: 4
- Maximum gradient steps: 5,000
- Training samples: 640 (steps per epoch: $640/4 = 160$)
- Total epochs: $5000/160 = 31.25$ epochs

Table 4: Training dynamics of BLoB on WG-S using official implementation. Training accuracy and loss exhibit significant instability with no clear convergence by epoch 32.

| Epoch | Training Accuracy (%) | | | Training Loss | | |
|---|---|---|---|---|---|---|
| | Seed 1 | Seed 2 | Seed 3 | Seed 1 | Seed 2 | Seed 3 |
| 1 | 51.13 | 49.12 | 53.64 | 0.7062 | 0.7097 | 0.6923 |
| 5 | 72.75 | 70.63 | 78.89 | 0.5796 | 0.5459 | 0.5003 |
| 10 | 67.97 | 62.40 | 76.86 | 0.6869 | 0.6569 | 0.4949 |
| 15 | 51.61 | 61.27 | 68.78 | 0.7080 | 0.6729 | 0.6043 |
| 20 | 44.35 | 62.77 | 58.29 | 0.7190 | 0.6315 | 0.6758 |
| 25 | 50.23 | 60.71 | 67.43 | 0.7099 | 0.6535 | 0.5755 |
| 30 | 50.39 | 61.47 | 68.86 | 0.6959 | 0.6480 | 0.5933 |
| 31 | 52.55 | 58.83 | 64.50 | 0.6993 | 0.6502 | 0.6084 |
| 32 | 45.86 | 64.08 | 65.17 | 0.7130 | 0.6290 | 0.5758 |

Table 5: Validation results for BLoB at epoch 32 (5,000 steps) on WG-S dataset. Seed 1 produced NaN predictions and is excluded. Low validation accuracy suggests undertraining.

| Seed | Val ACC (%) | Val ECE (%) | Val NLL |
|---|---|---|---|
| Seed 2 | 60.99 | 3.28 | 0.66 |
| Seed 3 | 58.14 | 3.27 | 0.67 |

**Critical Observations.** Table 4 summarizes the training dynamics over poch 32 (5,000 steps) across three random seeds provided by the official scripts. The training curves exhibit significant instability with no apparent convergence at 32 epochs. Training accuracy fluctuates dramatically across all seeds—for instance, Seed 1 drops from 72.75% at epoch 5 to 44.35% at epoch 20, while training loss fails to monotonically decrease. Validation accuracy remains substantially low (58–61%), far below the results reported in the original BLoB paper, and Seed 1 consistently produces NaN predictions during validation across multiple independent runs. The low ECE and NLL values likely reflect undertraining rather than good calibration, as the model has not yet learned to make confident predictions on this task.

**Rationale for Longer Training Schedule.** Given the apparent lack of convergence in the official BLoB implementation and similar observations with other baselines, we adopted a unified training schedule of 500 epochs for all methods. This ensures fair comparison under identical training conditions, provides sufficient duration for all methods to reach stable performance, and offers adequate

---

[1]Cloned via git from the official source https://github.com/Wang-ML-Lab/bayesian-peft

margin based on our empirical observation that most methods converge within 200 epochs, while certain baselines on challenging datasets require 300–400+ epochs in the worst case.

This extended training protocol ensures that performance differences reflect genuine methodological advantages rather than artifacts of premature termination or convergence failures.

# B  EXTEND EXPERIMENTS ON LLaMA 3.1 8B

## B.1  MEMORY USAGE AND RUN TIME

We evaluate the computational efficiency of different uncertainty quantification methods on the WG-S dataset. All experiments are conducted with a training batch size of 4, an inference batch size of 8, LoRA rank $r = 16$, $\alpha = 32$, and a sequence length of 400. These hyperparameters are kept consistent across all methods to ensure a fair comparison. In terms of PoLAR-BLoB, we applied variational inference to the core matrix $\Lambda \in \mathbb{R}^{r \times r}$ in the PoLAR decomposition $\Delta \mathbf{W} = \mathbf{U}\Lambda\mathbf{V}^{\top}$, while keeping the orthogonal factors $\mathbf{U} \in \text{St}(m, r)$ and $\mathbf{V} \in \text{St}(n, r)$ deterministic.

Table 6: GPU memory usage and runtime comparison across different uncertainty quantification methods. Bold denotes the best performance for each metric.

| Method | Training Memory (MB) | Test Memory (MB) | Runtime per Epoch (min) |
|---|---|---|---|
| PoLAR-VBLL (Ours) | 32,272 | 16,396 | 1:13 |
| PoLAR-BLoB | 31,728 | 18,874 | 14:36 |
| LoRA-BLoB | 40,475 | 18,762 | 14:49 |
| PoLAR-LA-LL | 31,714 | 15,662 | 0:34 |
| PoLAR-LA | 31,714 | 43,678 | 0:44 |
| LoRA-LA-LL | 30,612 | 16,313 | 0:43 |
| LoRA-LA | 30,612 | 42,131 | 0:57 |
| LoRA-VBLL | 30,546 | 16,764 | 1:14 |
| TFB | 30,612 | 16,667 | 0:48 |
| ScalaBL | 27,318 | 19,552 | 10:42 |
| C-LoRA | 24,236 | 19,102 | 10:04 |

As shown in Table 6, our method demonstrates a substantial reduction in runtime compared to BLoB-based methods, achieving approximately 12× speedup (1:13 vs. ∼14:40 min/epoch). Meanwhile, PoLAR-VBLL maintains a competitive inference memory footprint that is significantly lower than full-network Laplace approximations such as PoLAR-LA (16,396 MB vs. 43,678 MB).

The computational efficiency of PoLAR-VBLL stems from its architectural design that fundamentally differs from existing approaches. While BLoB-based methods rely on expensive full-network sampling that necessitates $K$ complete forward passes through the entire LLM backbone, our framework employs head-only sampling. This design choice enables a single backbone pass while restricting the stochastic sampling to the computationally lightweight last layer, dramatically reducing both memory consumption and inference time.

Furthermore, our PoLAR-VBLL leverages an analytical ELBO solution, which allows for exact gradient computation without the sampling overhead inherent in Monte Carlo-based approaches. In contrast, BLoB incurs approximately 50% additional parameter overhead (Samplawski et al., 2025; Rahmati et al., 2025) due to maintaining both mean and variance parameters across all adapter layers.

Table 7: Variational parameters per layer for different methods.

| Method | Variational Parameters per Layer | Calculation |
|---|---|---|
| LoRA-BLoB | 131,072 | $r \times d \times 2 = 16 \times 4096 \times 2$ |
| PoLAR-BLoB | 512 | $r \times r \times 2 = 16 \times 16 \times 2$ |
| ScalaBL | 32 | $r \times 2 = 16 \times 2$ |

Table 7 presents the variational parameter counts per layer for different methods. The dramatic difference between LoRA-BLoB (131,072 parameters) and PoLAR-BLoB (512 parameters) arises from the structural distinction in where stochasticity is introduced. LoRA-BLoB performs variational inference on the full low-rank matrices of dimension $d \times r$, whereas PoLAR-BLoB restricts the variational treatment to the core matrix of dimension $r \times r$. It is worth noting that C-LoRA adopts an entirely different approach by using deterministic parameters. Specifically, C-LoRA employs a contextual MLP module to dynamically generate input-dependent perturbation matrices, with parameter count per layer given by $(r \times 64 + 64) + (64 \times r^2 \times 2 + r^2 \times 2)$. The lower memory usage of C-LoRA can be attributed to two factors: first, the absence of the reparameterization trick eliminates the need to store noise matrices and intermediate states for backpropagation; second, deterministic parameters avoid the doubling of optimizer states (momentum and variance in AdamW) that variational methods require for both mean and variance parameters.

**Memory Analysis Beyond Static Parameter Counts.** A superficial analysis might suggest that LoRA adapters contribute less than 10% additional memory overhead. However, this perspective considers only static parameter counts and overlooks the dynamic memory consumption during training. In practice, GPU memory comprises several components: the base model parameters (`Llama-2-7B` in fp16), adapter parameters, optimizer states that store momentum and variance for each trainable parameter (effectively $2\times$ the trainable parameter memory), gradient buffers, and activation memory. Among these, the activation and reparameterization overhead often constitutes the dominant factor in peak memory usage. LoRA-BLoB performs reparameterization on large projection matrices of dimension $d \times r$, requiring storage of high-dimensional noise matrices and intermediate computation states. In contrast, PoLAR restricts this operation to the compact core matrix of dimension $r \times r$. Crucially, this overhead scales linearly with both batch size and sequence length, making the architectural choice increasingly important for larger-scale training.

In fact, BLoB's high memory cost is well-documented and motivates recent methods (ScalaBL (Samplawski et al., 2025), C-LoRA (Rahmati et al., 2025)) specifically designed to reduce this overhead.

**Experimental Configuration and Fair Comparison.** Our experimental settings differ from those in the original BLoB paper: we use LoRA rank of 16, alpha of 32, sequence length of 400, and target all projection layers including `q_proj`, `k_proj`, `v_proj`, `o_proj`, `gate_proj`, `up_proj`, and `down_proj`. The original BLoB paper uses rank of 8, alpha of 16, sequence length of 300, targeting only `q_proj`, `v_proj`, and `lm_head`. These differences render absolute numbers not directly comparable with those reported in the original BLoB paper. Nevertheless, our comparison remains fair as all methods are evaluated under identical settings within our unified framework. All methods use $K = 1$ samples during training, consistent with official BLoB implementation and standard practice. All baselines are implemented from their official repositories with the same percentage of trainable parameters.

In summary, PoLAR-VBLL effectively bridges the gap between predictive performance and computational efficiency, making it a practical choice for uncertainty quantification in resource-constrained deployment scenarios.

## B.2 ADDITIONAL BACKBONE EVALUATION ON LLAMA-3.1-8B

We conducted comprehensive experiments on `Llama-3.1-8B` across all six datasets with all baseline methods using the 5-epoch fine-tuning setting suggested by TFB (Shi et al., 2024). Note that this setting is not optimal for variational Bayesian methods, as the ELBO objective balances both accuracy and KL regularization simultaneously. In contrast, methods with deterministic training and post-hoc posterior estimation optimize solely for cross-entropy, potentially allowing them to achieve higher accuracy within limited training epochs. The batch size is set to $4$, and the maximum sequence length is restricted to 300 tokens. For the LoRA configuration, we set the rank to $r = 8$ and the scaling factor to $\alpha = 16$.

These comprehensive results in Table 8 demonstrate that our framework generalizes effectively beyond `Llama-2-7B` to larger and more recent model architectures, consistently achieving superior uncertainty quantification while maintaining competitive predictive performance, even under the limited training budget that inherently favors deterministic methods.

Table 8: Performances on ID datasets in terms of ACC, ECE, and NLL using `Llama-3.1-8B`. The evaluation is done across six datasets used in Wang et al. (2024); Shi et al. (2024). **Bold** and underlined denote the best and the second-best performance, respectively. **Note on ScalaBL:** We strictly followed the official implementation and hyperparameter configurations for ScalaBL. However, despite our best efforts, our reproduction yielded a model where the low ECE comes at the cost of substantially reduced accuracy. This pattern suggests potential underfitting in this specific setting.

| Metric | Method | Datasets | | | | | |
|--------|--------|----------|--------|--------|--------|--------|--------|
| | | WG-S | ARC-C | ARC-E | WG-M | OBQA | BoolQ |
| ACC (%) | LA | 77.22±1.38 | 84.49±0.37 | 89.86±0.54 | 84.04±0.23 | 88.32±0.33 | 88.26±0.50 |
| | PoLAR-LA | 77.51±0.94 | 84.57±1.10 | **91.80±0.19** | 83.88±0.90 | 88.70±0.56 | **89.54±0.86** |
| | PoLAR-LA-LL | 77.51±0.94 | 84.57±1.10 | **91.80±0.19** | 83.88±0.90 | 88.70±0.56 | **89.54±0.86** |
| | TFB-LL | 77.31±2.01 | 82.75±0.20 | 89.95±0.59 | 83.44±1.09 | 88.66±0.30 | 89.15±1.40 |
| | TFB | 77.61±1.19 | 83.34±1.03 | 90.90±0.01 | 83.31±0.35 | 88.44±0.24 | 87.98±1.79 |
| | BLOB | 76.68±0.99 | 82.88±0.51 | 91.49±0.21 | 80.78±1.05 | 87.90±0.35 | 88.58±0.17 |
| | C-LoRA | 73.07±0.50 | 78.72±0.47 | 89.88±0.87 | 77.46±0.78 | 86.59±0.14 | 88.18±1.29 |
| | ScalaBL | 49.92±0.69 | 79.40±1.48 | 89.21±0.20 | 53.40±1.43 | 69.62±7.99 | 82.18±2.14 |
| | PoLAR-VBLL (w/o LA) | **77.91±0.47** | **85.05±0.77** | 91.60±0.03 | **84.97±0.05** | **89.07±0.23** | 89.32±1.29 |
| | **PoLAR-VBLL** | **77.91±0.47** | **85.05±0.77** | 91.60±0.03 | **84.97±0.05** | **89.07±0.23** | 89.32±1.29 |
| ECE (%) | LA | 4.15±0.52 | 6.35±0.84 | 9.94±0.59 | 6.71±1.69 | 3.54±0.13 | 2.18±0.47 |
| | PoLAR-LA | 3.58±0.18 | 4.52±1.40 | 3.27±0.86 | 4.31±0.29 | 2.83±0.49 | 3.35±0.45 |
| | PoLAR-LA-LL | 7.95±0.92 | 6.07±0.72 | **2.94±0.67** | 6.37±1.10 | 3.75±0.01 | 5.52±0.12 |
| | TFB-LL | 9.99±0.90 | 5.02±2.15 | 3.13±0.24 | 4.12±1.57 | 3.58±0.21 | 3.87±1.41 |
| | TFB | 9.05±0.39 | 6.53±1.99 | 3.17±0.21 | 3.68±1.57 | 2.76±0.16 | 3.89±1.68 |
| | BLOB | 14.83±1.21 | 9.29±0.57 | 4.12±0.26 | 8.23±1.04 | 3.33±0.29 | 3.36±0.94 |
| | C-LoRA | 18.36±0.58 | 15.56±2.44 | 5.40±0.23 | 7.96±2.84 | 6.58±1.23 | 3.94±0.47 |
| | ScalaBL | 4.86±0.50 | 18.75±7.45 | 10.07±0.66 | 2.73±0.87 | 24.67±4.34 | 14.48±1.29 |
| | PoLAR-VBLL (w/o LA) | 9.04±0.18 | 7.40±0.26 | 3.61±0.22 | 6.10±0.01 | 2.50±0.10 | 2.63±0.23 |
| | **PoLAR-VBLL** | **3.31±1.40** | **3.56±0.41** | 3.22±0.42 | **3.00±0.10** | **2.44±0.11** | **1.88±0.27** |
| NLL | LA | 0.67±0.04 | 0.63±0.04 | 0.41±0.02 | 0.52±0.01 | 0.40±0.02 | 0.36±0.01 |
| | PoLAR-LA | 0.60±0.05 | 0.52±0.01 | 0.31±0.01 | 0.46±0.02 | 0.34±0.02 | 0.32±0.01 |
| | PoLAR-LA-LL | 0.76±0.12 | 0.58±0.05 | 0.26±0.01 | 0.52±0.06 | 0.37±0.01 | 0.34±0.03 |
| | TFB-LL | 0.59±0.05 | 0.53±0.01 | 0.28±0.02 | 0.41±0.01 | **0.32±0.01** | 0.29±0.04 |
| | TFB | **0.55±0.01** | 0.51±0.02 | 0.27±0.02 | 0.41±0.01 | **0.32±0.01** | 0.30±0.02 |
| | BLOB | 0.79±0.06 | 0.62±0.07 | 0.29±0.01 | 0.47±0.03 | 0.38±0.00 | **0.28±0.00** |
| | C-LoRA | 0.85±0.03 | 0.88±0.08 | 0.35±0.01 | 0.56±0.07 | 0.45±0.05 | 0.31±0.01 |
| | ScalaBL | 0.65±0.00 | 0.69±0.13 | 0.35±0.00 | 0.64±0.00 | 0.96±0.23 | 0.46±0.08 |
| | PoLAR-VBLL (w/o LA) | 0.61±0.01 | 0.51±0.05 | 0.27±0.03 | 0.40±0.01 | **0.32±0.02** | 0.29±0.00 |
| | **PoLAR-VBLL** | **0.55±0.02** | **0.50±0.05** | **0.24±0.04** | 0.39±0.01 | **0.32±0.02** | 0.29±0.00 |

## B.3 Ablation Study on the effects of VBLL and LA

To systematically evaluate the contribution of VBLL and LA in our framework, we conduct a comprehensive ablation study across all six datasets using Llama-3.1-8B as the backbone. Following the experimental protocol suggested by TFB (Shi et al., 2024), we fine-tune all methods for five epochs. Since our method applies Laplace Approximation (LA) exclusively to the last layer, we include PoLAR-LA-LL (which applies LA only to the last layer of a deterministically trained model) to ensure a direct and fair comparison. For completeness, we also compare against PoLAR-LA, which applies LA across all adapter layers.

The results presented in Table 9 reveal the following important findings regarding the individual contributions of VBLL and LA to our framework.

**VBLL as the Dominant Factor for Uncertainty Quantification.** The complete PoLAR-VBLL framework achieves the best ECE and NLL across nearly all datasets while maintaining competitive accuracy. Notably, even without the final LA refinement step, the PoLAR-VBLL (w/o LA) variant already delivers highly competitive calibration performance, particularly on OBQA and BoolQ where it achieves the second-best ECE. This observation demonstrates that VBLL is the primary driver of UQ quality in our framework, rather than relying on LA as a remedial component.

**Limitations of Deterministic Training for Uncertainty Estimation.** A comparison between PoLAR-LA and PoLAR-LA-LL provides valuable insights into the limitations of post-hoc uncer-

Table 9: Ablation study of effects of VBLL and LA coupled with the PoLAR adapter across six datasets using `LlaMa-3.1-8B`. **Bold** and underlined denote the best and the second-best performance, respectively.

| Metric | Method | Datasets | | | | | |
|--------|--------|------|-------|-------|------|------|-------|
| | | WG-S | ARC-C | ARC-E | WG-M | OBQA | BoolQ |
| ACC (%) ↑ | PoLAR-LA | 77.51±0.94 | 84.57±1.10 | **91.80±0.19** | 83.88±0.90 | 88.70±0.56 | **89.54±0.86** |
| | PoLAR-LA-LL | 77.51±0.94 | 84.57±1.10 | **91.80±0.19** | 83.88±0.90 | 88.70±0.56 | **89.54±0.86** |
| | PoLAR-VBLL (w/o LA) | **77.91±0.47** | **85.05±0.77** | 91.60±0.03 | **84.97±0.05** | **89.07±0.23** | 89.32±1.29 |
| | **PoLAR-VBLL (Full)** | **77.91±0.47** | **85.05±0.77** | 91.60±0.03 | **84.97±0.05** | **89.07±0.23** | 89.32±1.29 |
| ECE (%) ↓ | PoLAR-LA | 3.58±0.18 | 4.52±1.40 | 3.27±0.86 | 4.31±0.29 | 2.83±0.49 | 3.35±0.45 |
| | PoLAR-LA-LL | 7.95±0.92 | 6.07±0.72 | **2.94±0.67** | 6.37±1.10 | 3.75±0.01 | 5.52±0.12 |
| | PoLAR-VBLL (w/o LA) | 9.04±0.18 | 7.40±0.26 | 3.61±0.22 | 6.10±0.01 | 2.50±0.10 | 2.63±0.23 |
| | **PoLAR-VBLL (Full)** | **3.31±1.40** | **3.56±0.41** | 3.22±0.42 | **3.00±0.10** | **2.44±0.11** | **1.88±0.27** |
| NLL ↓ | PoLAR-LA | 0.60±0.05 | 0.52±0.01 | 0.31±0.01 | 0.46±0.02 | 0.34±0.02 | 0.32±0.01 |
| | PoLAR-LA-LL | 0.76±0.12 | 0.58±0.05 | 0.26±0.01 | 0.52±0.06 | 0.37±0.01 | 0.34±0.03 |
| | PoLAR-VBLL (w/o LA) | 0.61±0.01 | 0.51±0.05 | 0.27±0.03 | 0.40±0.01 | **0.32±0.02** | **0.29±0.00** |
| | **PoLAR-VBLL (Full)** | **0.55±0.02** | **0.50±0.05** | **0.24±0.04** | **0.39±0.01** | **0.32±0.02** | **0.29±0.00** |

tainty estimation on deterministically trained models. While both methods achieve comparable NLL values, this similarity is largely attributable to high-confidence predictions rather than proper calibration. The substantially higher ECE of PoLAR-LA-LL compared to PoLAR_LA across most datasets indicates that applying LA exclusively to the last layer of a deterministically trained model is insufficient for achieving well-calibrated uncertainty estimates. This performance gap suggests that deterministic training fails to discover posterior geometries amenable to uncertainty quantification, necessitating LA compensation across all adapter layers to achieve reasonable calibration.

**VBLL Provides Superior Initialization for Laplace Refinement.** A striking observation emerges from comparing our full PoLAR-VBLL method against PoLAR-LA: despite applying LA only to the last layer, PoLAR-VBLL achieves superior calibration compared to PoLAR-LA, which applies LA across all layers. This result underscores the effectiveness of variational training in discovering high-quality posterior modes. The VBLL component actively guides the optimization process toward parameter configurations that inherently support reliable uncertainty estimation, thereby providing an ideal foundation for subsequent LA refinement. Consequently, only minimal post-hoc adjustment at the last layer is required to achieve state-of-the-art uncertainty quantification.

**Trade-off Between Accuracy and Calibration.** Under the five-epoch fine-tuning protocol suggested by TFB (Shi et al., 2024), which favors methods based on deterministic training followed by post-hoc posterior estimation, we observe that deterministic methods (PoLAR-LA and PoLAR-LA-LL) achieve marginally higher accuracy on certain datasets such as ARC-E and BoolQ. This advantage stems from their exclusive optimization of the cross-entropy loss without regularization from the KL divergence term. In contrast, VBLL jointly optimizes predictive accuracy and the variational objective, which may result in slightly lower accuracy on some benchmarks. However, this joint optimization yields substantially superior uncertainty quantification, as evidenced by the consistently better ECE and NLL achieved by PoLAR-VBLL across the majority of datasets.

**Summary.** These comprehensive results across six datasets demonstrate that VBLL constitutes the core working component of our framework. The synergy between VBLL, which performs mode discovery during training, and LA, which refines the local posterior geometry post-hoc, represents an intentional and effective design choice. The experimental evidence firmly establishes that LA serves as a complementary refinement step rather than a compensatory mechanism for a deficient VBLL component.

### B.3.1 DISENTANGLING POLAR AND VBLL CONTRIBUTIONS

Another potential concern is whether our performance gains stem primarily from the PoLAR parameterization rather than the VBLL component. To address this, we conduct an additional ablation study on WinoGrande-Simple using `Llama-2-7B`, where we apply PoLAR parameterization to multiple baseline uncertainty quantification methods.

Table 10: Extended ablation study comparing PoLAR-VBLL with PoLAR substituted baselines on WG-S using `Llama-2-7B`. Results demonstrate that variational treatment of the classification layer (VBLL) is more effective than applying VI to adapter parameters (BLoB).

| Method | ACC (%) ↑ | ECE (%) ↓ | NLL ↓ |
|---|---|---|---|
| PoLAR-LA | 70.33±0.69 | 12.16±2.58 | 0.69±0.03 |
| PoLAR-LA-LL | 70.33±0.69 | 14.63±1.14 | 0.71±0.05 |
| PoLAR-BLoB | 70.39±0.26 | 12.06±0.81 | 0.73±0.04 |
| PoLAR-VBLL (w/o LA) | **71.62±0.27** | 8.26±0.60 | 0.66±0.03 |
| **PoLAR-VBLL (Full)** | **71.62±0.27** | **7.31±0.32** | **0.60±0.01** |

The results in Table 10 clearly demonstrate that VBLL, rather than PoLAR parameterization alone, is responsible for our method's superior uncertainty quantification. All variants in this ablation employ the same PoLAR adapter structure, yet their calibration performance varies dramatically. PoLAR + LAP and PoLAR + BLoB achieve comparable calibration, while PoLAR-VBLL (w/o LA) delivers substantially better performance with approximately $1.5\times$ lower ECE. Notably, even without the LA refinement step, PoLAR-VBLL (w/o LA) already achieves strong calibration, confirming that VBLL constitutes the primary working mechanism. The final LA step provides further refinement, but the core uncertainty quantification capability comes from VBLL's ability to discover well-calibrated posterior modes during training.

### B.4 TIGHTNESS OF THE JENSEN BOUND

A potential concern regarding our VBLL formulation is whether the Jensen bound employed in Eq. 6 provides a sufficiently tight approximation to the true ELBO objective. To address this concern, we conduct an empirical comparison between our analytical Jensen-based estimator and a Monte Carlo (MC) estimator with 50 samples across the full training horizon.

Specifically, we train PoLAR-VBLL on the WG-S dataset using `Llama-3.1-8B` as the backbone and record the training loss computed by both estimators at regular intervals over 400 training steps. The results are presented in Table 11.

Table 11: Comparison of training loss trajectories between the Jensen bound and 50-sample Monte Carlo estimation on WGS dataset using `Llama-3.1-8B`.

| Training Steps | VBLL (Jensen) | VBLL (50-sample MC) | Absolute Gap |
|---|---|---|---|
| 0 | 69.50 | 61.23 | 8.27 |
| 50 | 50.71 | 50.97 | 0.26 |
| 100 | 45.57 | 45.65 | 0.08 |
| 150 | 40.95 | 40.86 | 0.09 |
| 200 | 36.96 | 36.95 | 0.01 |
| 250 | 33.57 | 33.74 | 0.17 |
| 300 | 31.08 | 31.36 | 0.28 |
| 350 | 29.55 | 29.89 | 0.34 |
| 400 | 28.49 | 28.83 | 0.34 |

The results reveal several important observations regarding the fidelity of our Jensen-based optimization. First, the initial gap between the two estimators (8.27 at step 0) undergoes rapid convergence within the first 50 training steps, decreasing to merely 0.26. This rapid alignment indicates that the Jensen bound quickly becomes an accurate proxy for the true objective as the model parameters move away from their random initialization.

Second, after this initial convergence phase, the absolute gap remains remarkably stable throughout the remainder of training, consistently staying below 0.35 from step 50 to step 400. This stability demonstrates that the Jensen bound maintains its approximation quality across the entire optimization trajectory, rather than degrading as the posterior distribution evolves during training.

Third, and most critically, the gap does not exhibit any increasing trend as training progresses. This absence of divergence confirms that optimizing the Jensen-based lower bound does not lead the model toward regions where the bound becomes loose or misleading. Instead, the Jensen estimator and the MC estimator track each other closely throughout the full training horizon.

These extended results provide strong empirical evidence that our analytical Jensen-based formulation maintains fidelity to the true ELBO objective. The tight correspondence between the two estimators validates our design choice of employing the Jensen bound, which enables efficient closed-form gradient computation without sacrificing optimization quality. This computational advantage is substantial: while the 50-sample MC estimator requires 50 forward passes through the last layer per training step, our Jensen-based approach achieves comparable optimization trajectories with a single analytical computation.

### B.5 SENSITIVITY TO PRIOR AND INITIALIZATION

We conduct a comprehensive sensitivity analysis to evaluate the robustness of our method with respect to two critical factors: (1) the choice of prior distribution, and (2) the initialization of variational parameters. We investigate the sensitivity to the prior scale parameter $\sigma_0$, which controls the width of the Gaussian prior over the last-layer weights $\mathcal{N}(\mathbf{0}, \sigma_0^2 \mathbf{I})$. To assess initialization robustness, all experiments are conducted across three different random seeds $\{1, 2, 3\}$, which affect both data shuffling and stochastic aspects of variational parameter initialization. We report the mean and standard deviation across these seeds.

Table 12: Sensitivity analysis of prior scale $\sigma_0$ on WG-S under LLaMA 2 7B. Results are averaged over three random seeds with standard deviations reported. ACC: accuracy (%), ECE: expected calibration error (%), NLL: negative log-likelihood.

| Prior Scale ($\sigma_0$) | ACC ($\uparrow$) | ECE ($\downarrow$) | NLL ($\downarrow$) |
|---|---|---|---|
| 0.1 | $70.92 \pm 0.24$ | $9.10 \pm 0.53$ | $0.68 \pm 0.04$ |
| **1.0** (Default) | $\mathbf{71.62 \pm 0.27}$ | $\mathbf{8.26 \pm 0.60}$ | $\mathbf{0.66 \pm 0.03}$ |
| 10.0 | $70.70 \pm 0.50$ | $10.59 \pm 1.17$ | $0.69 \pm 0.07$ |

Table 12 summarizes the performance under different prior scales on the WG-S dataset. We can make the following observations:

**(1) Optimal prior scale:** The prior scale $\sigma_0 = 1.0$ achieves the best overall performance across all metrics. Both overly restrictive ($\sigma_0 = 0.1$) and overly diffuse ($\sigma_0 = 10.0$) priors result in degraded performance, with decreases in accuracy and increases in both calibration error and negative log-likelihood.

**(2) Convergence dynamics:** We observe that as the prior scale increases from 0.1 to 10.0, the optimization process converges progressively more slowly during training. This suggests that excessively wide priors introduce additional optimization challenges, potentially requiring more iterations to reach comparable solution quality.

**(3) Initialization robustness:** The relatively small standard deviations across different random seeds demonstrate that our method exhibits strong robustness to initialization. This stability is consistent across all tested prior scales, indicating that the variational learning process reliably converges to high-quality solutions despite variations in random initialization. The consistency across seeds also validates the reproducibility of our approach.

### B.6 STABLE RANK ANALYSIS AND THEORETICAL JUSTIFICATION

In this section, we provide both theoretical motivation and empirical validation for combining PoLAR with VBLL. We first establish the theoretical foundation linking feature geometry to uncertainty quantification quality, and then present empirical evidence demonstrating that PoLAR preserves the geometric properties essential for reliable Bayesian inference.

### B.6.1 THEORETICAL MOTIVATION: DISTANCE-AWARE FEATURES FOR BAYESIAN LAST LAYER METHODS

Recent work on deterministic uncertainty quantification has established that last-layer Bayesian methods critically depend on the geometric properties of the feature extractor. In particular, SNGP (Liu et al., 2020) demonstrates that distance-aware features—where semantically distinct inputs remain well-separated in the feature space—are essential for reliable uncertainty estimation. We argue that VBLL shares this requirement: when the Bayesian last layer receives features from a distance-preserving extractor, it can effectively distinguish between in-distribution (ID) and out-of-distribution (OOD) samples based on their relative positions in the feature space.

A critical failure mode arises when the learned transformation exhibits low effective dimensionality, a phenomenon termed feature collapse (Postels et al., 2021). Under feature collapse, the adapter projects high-dimensional inputs onto a narrow, low-dimensional subspace, causing semantically diverse inputs—including OOD samples—to cluster together indistinguishably from ID data. This geometric compression fundamentally limits the Bayesian last layer's capacity to detect distribution shift, as the distance information necessary for uncertainty-aware inference is lost during feature extraction.

The stable rank of the learned weight update $\Delta \mathbf{W}$ provides a quantitative measure of this geometric property. Defined as

$$\text{stable-rank}(\Delta \mathbf{W}) = \frac{\|\Delta \mathbf{W}\|_F^2}{\|\Delta \mathbf{W}\|_2^2}, \tag{31}$$

the stable rank captures the effective dimensionality of the transformation by computing the ratio of the squared Frobenius norm to the squared spectral norm. A stable rank approaching 1.0 indicates a nearly rank-1 projection that severely compresses the feature space, while higher values suggest a more isotropic transformation that preserves multiple effective directions.

### B.6.2 EMPIRICAL VALIDATION: PoLAR PRESERVES FEATURE GEOMETRY

To empirically validate our theoretical motivation, we conduct a comparative stable rank analysis between LoRA and PoLAR across multiple datasets. Figure 2 presents the distribution of stable rank values for both methods.

The results reveal a striking contrast between the two adaptation strategies. Standard LoRA exhibits an average stable rank of approximately 1.53, approaching the theoretical minimum of 1.0. This low value indicates that LoRA effectively performs a nearly rank-1 projection, compressing the learned updates into a highly anisotropic subspace despite the nominally higher allocated rank. Such geometric compression aligns with previous observations of rank collapse in LoRA (Lion et al., 2025) and explains the suboptimal performance of LoRA-based uncertainty quantification methods, particularly in OOD detection scenarios where distance preservation is critical.

In contrast, PoLAR maintains a significantly higher average stable rank of approximately 2.86. By constraining the low-rank factors $\mathbf{U}$ and $\mathbf{V}$ to the Stiefel manifold through orthogonality constraints, PoLAR encourages a more isotropic transformation that preserves multiple effective directions in the feature space. This geometric preservation directly supports the requirements of VBLL: the Bayesian last layer receives features that maintain semantic distances between inputs, enabling more reliable uncertainty estimation for both ID and OOD samples.

The connection between stable rank and downstream performance is evident in our experimental results. Across all evaluation benchmarks, PoLAR-based methods consistently outperform their LoRA counterparts in both predictive accuracy and uncertainty calibration. The higher stable rank of PoLAR translates to richer feature representations that better capture task-specific patterns while preserving the geometric structure necessary for principled Bayesian inference.

### B.6.3 SUMMARY: A PRINCIPLED DESIGN

We emphasize that our framework follows a principled design methodology where each component addresses a specific, well-motivated requirement, rather than claiming an axiomatic derivation from first principles. The roles of the three components are as follows:

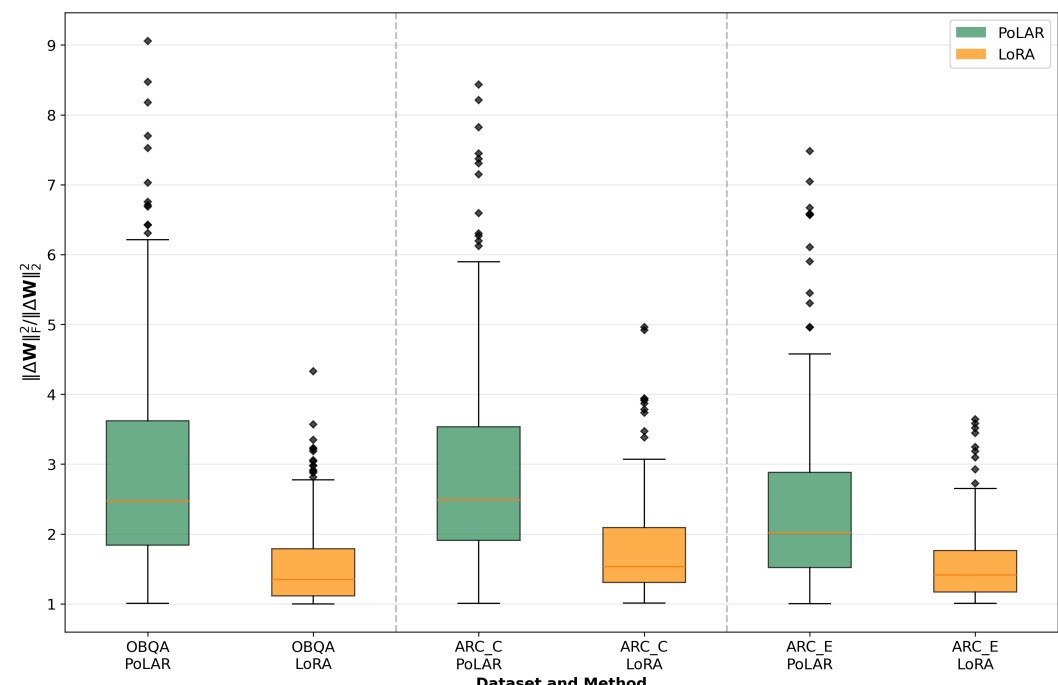

Figure 2: Stable rank comparison between PoLAR and LoRA across three datasets. PoLAR consistently achieves higher stable rank values, indicating better preservation of feature geometry and effective utilization of the allocated parameter space.

PoLAR serves as the feature extractor that preserves feature geometry through orthogonality constraints, maintaining the high stable rank necessary for distance-aware representations. This geometric preservation directly addresses the requirements identified by SNGP (Liu et al., 2020) for effective last-layer uncertainty methods.

VBLL provides tractable predictive uncertainty through variational inference on the last layer weights. Beyond uncertainty quantification, VBLL actively guides the optimization process toward high-quality posterior modes that support reliable calibration. This mode discovery during training is essential for the subsequent refinement step.

The optional LA refines the local posterior geometry around the well-calibrated mode identified by VBLL. Crucially, without VBLL to locate an appropriate posterior mode, LA alone cannot achieve proper calibration—as demonstrated by the inferior performance of PoLAR-LA-LL compared to PoLAR-VBLL in our ablation studies. VBLL provides the necessary initialization that enables LA to serve as an effective finishing touch.

This principled combination, where each component's role is both theoretically motivated and empirically validated, underlies the consistent superior performance of PoLAR-VBLL across diverse benchmarks and model architectures.

