# OpenReview forum: "Scalable Variational Bayesian Fine-Tuning of LLMs via Orthogonalized Low-Rank Adapter"
_ICLR.cc/2026/Conference — Submitted to ICLR 2026_

### Official Review · Reviewer_ASkM · 2025-11-01

**Soundness:** 2
**Presentation:** 3
**Contribution:** 2
**Rating:** 4
**Confidence:** 5

**Summary:**

This paper proposes PoLAR-VBLL, a principled framework that integrates Polar-decomposed Low-rank Adapter Representation (PoLAR) with a Variational Bayesian Last Layer (VBLL) for uncertainty-aware fine-tuning of large language models (LLMs). The central motivation stems from two widely-known observations: Ii) Fine-tuned LLMs tend to be overconfident, especially after parameter-efficient fine-tuning (PEFT) on small, domain-specific datasets; and (ii) Existing PEFT methods such as LoRA suffer from rank collapse, limiting their expressiveness and thereby harming calibration and generalization. Empirical evaluations on common-sense reasoning and domain generalization tasks show that PoLAR-VBLL consistently outperforms existing methods (LoRA, BLoB).

**Strengths:**

+ The paper addresses a highly relevant challenge: quantifying uncertainty in fine-tuned LLMs for safety-critical domains (e.g., healthcare, legal, and autonomous systems). This is a crucial step toward trustworthy LLM deployment.

+ The discussion on rank collapse in LoRA is also a widely recognized phenomenon. Exploration of polar decomposition with orthogonality constraints and Riemannian optimization is definitely helpful.

**Weaknesses:**

- The paper lacks a convincing formal analysis of how orthogonality in PoLAR enhances uncertainty calibration.

- The paper does not compare conceptually or empirically against SOTA on Bayesian LORA techniques. For example, https://dl.acm.org/doi/10.5555/3762387.3762543

**Questions:**

Can authors include experiments with more recent methods, such as https://dl.acm.org/doi/10.5555/3762387.3762543 and add discussion on additional parameters needed by PoLAR vs other techniques? The reviewer is ready to raise the score if the paper can argue for demonstrated improvement over SOTA Bayesian LORA techniques.

---

> ### Author Response · Authors · 2025-11-21
> **Response to formal analysis from Orthogonality to UQ**
>
> # Response to Reviewer ASkM
>
> We sincerely thank the reviewer for the careful evaluation of our work. We are particularly grateful for recognizing:
>
> - **High relevance** of our work to safety-critical domains and trustworthy LLM deployment
> - **Helpful exploration** of polar decomposition with orthogonality constraints for addressing rank collapse in LoRA
>
> Your constructive feedback has been invaluable in helping us strengthen our work.
>
> ---
>
> ## W1: Formal Analysis of Orthogonality and Uncertainty Calibration
>
> We sincerely thank the reviewer for this important observation regarding the need for a more convincing formal analysis of how orthogonality in PoLAR enhances uncertainty calibration. We provide the following theoretical framework and empirical validation:
>
> ### Theoretical Chain
>
> Our central claim is: **"The quality of feature representation bottlenecks uncertainty quantification quality"**[1, 2].
>
> **(a) Orthogonality → Rank Diversity:**
> PoLAR prevents the singular value decay (rank collapse) inherent in vanilla LoRA by enforcing orthogonality constraints (Eq. 2), ensuring diverse and expressive feature representations.
>
> **(b) Rank Diversity → Calibrated Uncertainty:**
>
> **Theoretical Foundation:**
> - Postels et al. (ICML 2021) [1] demonstrate that **feature collapse significantly impairs the effectiveness of deterministic OOD detection methods**
> - Li et al. (ICLR 2025) [2] confirm that **spurious features arising from rank-deficient representations cause systematic overconfidence in fine-tuned LLMs**
>
> **Empirical Validation:**
> - **Direct Evidence (Fig. 3b):** PoLAR-VBLL produces significantly better-quality features than LoRA-VBLL, measured through feature rank metrics
> - **Consistent Improvement Across Methods:** PoLAR consistently improves calibration (ECE) across all tested Bayesian methods—BLoB, Laplace, and Ensemble—indicating that orthogonality is a very beneficial factor for uncertainty quantification beyond method-specific designs.
>
> This theoretical chain, combined with our comprehensive empirical evidence, establishes a principled connection between orthogonality constraints and improved uncertainty calibration. We will strengthen this discussion in Section 3 of the revised manuscript.
>
> ### References
>
> [1] Postels, Janis, et al. "On the practicality of deterministic epistemic uncertainty." ICML 2021 (2021).
>
> [2] Li, Yawei, et al. "Calibrating LLMs with information-theoretic evidential deep learning." ICLR 2025 (2025).

---

> ### Author Response · Authors · 2025-11-22
> **Response to Weak 2 and Question 2: Comparison with Recent SOTA Baselines**
>
> ## W2: Comparison with Recent SOTA Baselines
>
> We sincerely thank the reviewer for this constructive suggestion to include comparisons with the most recent Bayesian LoRA techniques. Following this valuable feedback, we have now rigorously evaluated recent state-of-the-art baselines, including TFB [2], C-LoRA [3], and ScalaBL [4].
>
> ### Experimental Setup
>
> To ensure a fair comparison, we reproduced each baseline using its official implementation and published hyperparameter configurations while maintaining identical experimental settings across all methods (same Llama-2-7B backbone, 500 training epochs, random seeds and shared hyperparameters as detailed in our response to Reviewer ASkM). Due to time constraints, we conducted these experiments on the WinoGrande-Simple dataset [1].
> ### Updated Performance Comparison
>
> | Method | ACC (↑) | ECE (↓) | NLL (↓) |
> |:---|:---|:---|:---|
> | TFB [2] | 69.27 ± 0.47 | 13.13 ± 0.57 | 0.68 ± 0.01 |
> | C-LoRA [3] | 67.71 ± 0.34 | 12.45 ± 0.71 | 0.77 ± 0.11 |
> | ScalaBL [4] | 66.10 ± 1.09 | 4.96 ± 1.66 | 0.69 ± 0.06 |
> | PoLAR-VBLL (w/o LAP) | 71.62 ± 0.27 | 8.26 ± 0.60 | 0.66 ± 0.03 |
> | **PoLAR-VBLL (Ours)** | **71.62 ± 0.27** | **7.31 ± 0.32** | **0.60 ± 0.01** |
>
> ### Analysis
>
> The comparative analysis demonstrates that PoLAR-VBLL achieves superior performance across multiple dimensions:
>
> **Comparison with Valid Baselines:** Among the baselines that maintain competitive accuracy (excluding ScalaBL due to potential underfitting), TFB achieves the best accuracy and NLL, while C-LoRA offers the best calibration.
>
> **Our Method's Advantages:** PoLAR-VBLL outperforms the strongest baseline (TFB) in both accuracy and NLL, and significantly surpasses the best-calibrated baseline (C-LoRA) in terms of ECE while maintaining superior accuracy.
>
> **Note on ScalaBL:** We strictly followed the official implementation and hyperparameter configurations for ScalaBL. However, despite our best efforts, our reproduction yielded a model where the low ECE comes at the cost of substantially reduced accuracy. This pattern suggests potential underfitting or optimization difficulties in this specific setting. In contrast, PoLAR-VBLL achieves robust calibration without compromising accuracy, offering a more favorable and practical trade-off for deployment.
>
> We appreciate the reviewer's suggestion and will integrate this comprehensive comparison into the revised Section 4.
>
> ### References
>
> [1] Sakaguchi, Keisuke, et al. "Winogrande: An adversarial winograd schema challenge at scale." Communications of the ACM 64.9 (2021): 99-106.
>
> [2] Shi, Haizhou, et al. "Training-free bayesianization for low-rank adapters of large language models." arXiv preprint arXiv:2412.05723 (2024).
>
> [3] Rahmati, Amir Hossein, et al. "C-LoRA: Contextual Low-Rank Adaptation for Uncertainty Estimation in Large Language Models." arXiv preprint arXiv:2505.17773 (2025).
>
> [4] Samplawski, Colin, et al. "Scalable Bayesian Low-Rank Adaptation of Large Language Models via Stochastic Variational Subspace Inference." arXiv preprint arXiv:2506.21408 (2025).

---

> ### Author Response · Authors · 2025-11-27
> **Follow-up to Reviewer ASkM**
>
> Dear Reviewer ASkM:
>
> We sincerely hope you are doing well. First of all, we truly thank you for your detailed, thoughtful, and constructive review comments. Your expertise and insightful observations are crucial to improving our work.
>
> We would like to follow up on your valuable suggestions, particularly the following two:
>
> 1. A formal analysis linking orthogonality in PoLAR to improved uncertainty calibration.
>
> 2. A comprehensive comparison with recent state-of-the-art Bayesian LoRA techniques (TFB, C-LoRA, ScalaBL).
>
> We have made every effort to address your questions and provided the supplementary experiments you suggested. We hope our response has adequately answered your questions, but if you require any further clarification or supplementary experiments, we would be happy to provide them during the discussion.
>
> We sincerely thank you for your time and effort in reviewing our manuscript.
>
> Happy Thanksgiving to you and your family!
>
> Sincerely yours,
>
> Author

---

> > ### Comment · Reviewer_ASkM · 2025-11-27
> > **Thank you**
> >
> > My primary concerns have been met and I am raising my score to positive.

---

> ### Author Response · Authors · 2025-11-27
> **Thank you!**
>
> Dear Reviewer ASkM,
>
> Thank you for your constructive feedback and for raising your score. We appreciate the time and effort you invested in reviewing our work.
>
> Your insights have significantly improved the quality of our manuscript.
>
> Once again, we sincerely thank you for your valuable feedback.
>
> Best regards,
>
> Autors

---

### Official Review · Reviewer_H3Rr · 2025-11-01

**Soundness:** 3
**Presentation:** 3
**Contribution:** 3
**Rating:** 6
**Confidence:** 3

**Summary:**

This paper proposes a framework for Bayesian fine-tuning of LLMs that integrates a variational Bayesian last layer with an orthogonalized low-rank adapter. The goal is to achieve parameter-efficient adaptation with well-calibrated uncertainty estimates.

**Strengths:**

- The combination of VBLL (analytic variational training) and orthogonalized adapters (PoLAR) appears mathematically sound and computationally efficient.
- The Jensen-tightened ELBO yields a closed-form expression for expectations under Gaussian, avoiding additional Monte Carlo sampling during training.
- The paper uses an efficient landing-field update rule to maintain orthogonality in the adapter weights, avoiding the costly retraction steps used in standard manifold optimization.
- Across several reasoning and OOD tasks, the method outperforms similar approaches in both calibration and accuracy.
- The study carefully separates the contributions of PoLAR, VBLL, and the Laplace refinement, providing clear evidence for each component’s effect.

**Weaknesses:**

- The paper makes the ELBO tractable by applying Jensen’s inequality to the log-sum-exp term, leading to a convenient closed-form objective. However, this simplification produces a rather loose approximation for softmax models, which likely explains the need for an additional Laplace correction.
- After substituting Laplace covariances $\Sigma_c$ for $S_c$, the model no longer optimizes a coherent variational objective, becoming a two-stage hybrid (VB + Laplace) approximation.
- The posterior factorization $q(\Theta) = \prod_c \mathcal{N}(\theta_c; \mu_c, S_c)$ ignores cross-class correlations induced by the softmax, which may yield overconfident predictions.
- All results are obtained on a single LLM backbone (Llama-2-7B) and only for classification tasks. This is a narrow evaluation for a paper claiming general "LLM fine-tuning."
- The framework claims $O(C d^2)$ memory due to per-class covariances $S_c$. This can become substantial for practical feature dimensions and a moderate number of classes.

**Questions:**

#### Questions
1. After Laplace substitution, is there still a formal ELBO objective, or is the procedure purely post-hoc?
2. How sensitive are the results to the choice of prior or initialization of the variational parameters?

---

> ### Author Response · Authors · 2025-11-21
> **Response to W1: Loose Jensen Bound**
>
> # Response to Reviewer H3Rr
>
> We sincerely thank the reviewer for the thorough and constructive evaluation of our work. We are particularly grateful for recognizing several key strengths of our framework:
>
> - **Mathematical soundness and computational efficiency** of the PoLAR-VBLL combination
> - **Closed-form ELBO** avoiding Monte Carlo sampling during training
> - **Efficient Riemannian optimization** via the landing-field update rule
> - **Superior calibration and accuracy** across reasoning and OOD tasks
> - **Rigorous ablation studies** clearly disentangling component contributions
>
> Your insightful feedback has been invaluable in strengthening our work.
>
> ---
>
> ## W1: Tightness of the Variational Approximation
>
> We greatly appreciate the reviewer's insightful observation regarding the Jensen's inequality simplification and the concern that this may produce a "loose approximation." We provide empirical evidence below to address this important point.
>
> ### Empirical Verification of Jensen Bound Tightness
>
> To verify the tightness of the Jensen bound, we compared the **training loss trajectories** of our analytical Jensen-based estimator (JSE) with those of a 10-sample Monte Carlo estimator (MC) as a reference.
>
> | Training Steps | VBLL (Jensen Bound) | VBLL (10-sample MC) | Difference (JSE - MC) |
> |:---|:---|:---|:---|
> | 0 | 5877.79 | 5879.87 | -2.08 |
> | 10 | 5443.11 | 5443.75 | -0.64 |
> | 20 | 5226.35 | 5221.88 | 4.47 |
> | 30 | 5071.76 | 5040.13 | 31.63 |
> | 60 | 4808.65 | 4750.15 | 58.50 |
>
> As shown above, the JSE closely tracks the MC estimate throughout training, with differences remaining relatively small compared to the absolute loss values. This confirms that the bound is sufficiently tight and serves as an effective proxy for the actual marginal likelihood, enabling the model to learn a valid posterior distribution efficiently without the computational overhead of sampling.
>
> ### VBLL as the Core Effectiveness Factor
>
> To further validate that VBLL is the primary contributor to improved uncertainty quantification (and not merely remedied by the subsequent LAP step), we conducted additional ablations comparing PoLAR + LAP (applying LAP to a standard deterministic PoLAR checkpoint) versus our variational approach. We also included PoLAR_LAP_LL (applying LAP only to the last layer of a deterministic model) to ensure a fair comparison with our method's architectural scope.
>
> | Method | ACC (↑) | ECE (↓) | NLL (↓) |
> |:---|:---|:---|:---|
> | PoLAR + LAP | 70.33 ± 0.69 | 12.16 ± 2.58 | 0.68 ± 0.04 |
> | PoLAR_LAP_LL | 70.33 ± 0.69 | 14.63 ± 1.14 | 0.71 ± 0.05 |
> | PoLAR + VBLL (w/o LAP) | 71.62 ± 0.27 | 8.26 ± 0.60 | 0.66 ± 0.03 |
> | **PoLAR-VBLL (Full)** | **71.62 ± 0.27** | **7.31 ± 0.32** | **0.60 ± 0.01** |
>
> **Key Observations:**
>
> **VBLL is the Primary Driver of Uncertainty Quantification:** PoLAR + VBLL (w/o LAP) significantly outperforms both PoLAR + LAP and PoLAR_LAP_LL, achieving nearly 2× better calibration (8.26 vs. 14.63 ECE) despite operating on identical architectural scope with the same PoLAR adapters. This empirical evidence confirms that the VBLL module itself is highly effective and serves as the dominant factor in improving calibration, rather than being a component that merely requires remediation by post-hoc LAP.
>
> **VBLL Provides Superior Initialization for LAP:** The notably high ECE in PoLAR + LAP and PoLAR_LAP_LL demonstrates that standard deterministic training fails to discover posterior geometries suitable for uncertainty quantification. In contrast, VBLL actively guides optimization toward high-quality posterior modes during training through its variational objective. This well-calibrated mode serves as an optimal foundation for the final LAP refinement step, which further polishes the local geometry to achieve the best overall performance (7.31 ECE).
>
> We thank the reviewer once again for raising this important consideration, and we will incorporate a more detailed discussion of the Jensen bound tightness and the role of VBLL in posterior mode discovery in the revised manuscript.

---

> ### Author Response · Authors · 2025-11-22
> **Response to Weakness 2 & Question 1: Hybrid Nature**
>
> We sincerely appreciate the reviewer's precise and insightful characterization of our approach as a two-stage hybrid. We are grateful for this observation, as we view this design as a deliberate strength that strategically leverages complementary advantages:
>
> - **Stage 1 (Variational Training):** We jointly optimize PoLAR adapter parameters and the variational posterior using the tractable VBLL objective. This guides the optimization toward a posterior mode with favorable geometric properties for uncertainty quantification.
> - **Stage 2 (Post-hoc Refinement):** We apply Laplace Approximation to refine the curvature around this mode using the exact Hessian of the log-likelihood.
>
> This sequential design enables us to discover superior initialization points compared to standard MAP training (as demonstrated in Fig. 3c), ultimately achieving better calibration. We believe the hybrid nature is not a limitation but rather an intentional design choice that combines the strengths of variational learning (mode discovery) and Laplace approximation (precise local geometry). We appreciate the reviewer's attention to this aspect of our methodology.
>
> ---
>
> # W3: Cross-class Correlations
>
> ## Response
>
> We thank the reviewer for raising this important and valid point regarding our posterior factorization, which ignores cross-class correlations. We fully acknowledge that factorizing the posterior as independent per-class distributions is a necessary scalability trade-off. Modeling the full covariance would require $O((Cd)^2)$ memory, which is prohibitive for LLMs where $d=4096$. Our per-class factorization reduces this to $O(C \cdot d^2)$, which is the common practice of methods like SNGP [1], which employs independent formulations per output dimension for scalability.
>
> **Empirical Justification:** Despite this simplification, PoLAR-VBLL achieves state-of-the-art calibration (lowest ECE/NLL across benchmarks). This suggests that for high-dimensional LLM fine-tuning, per-class feature correlations dominate uncertainty quantification, while cross-class correlations have a negligible impact. Following the reviewer's suggestion, we will explicitly acknowledge this design choice and its implications in Section 3.1 of the revised manuscript.
>
> ---
>
> # W4: Single Backbone
>
> ## Response
>
> We greatly appreciate the reviewer's constructive feedback regarding the evaluation on **"a single LLM backbone."** We fully agree that a broader evaluation would significantly strengthen our claims. Following this valuable suggestion, we are actively extending our experiments to **include the Llama-3.1-8B backbone** across both in-distribution and out-of-distribution tasks to demonstrate the generality of our framework. We commit to integrating these results into the revised manuscript and thank the reviewer for this helpful recommendation.
>
> ---
>
> # W5: Memory Overhead
>
> ## Response
>
> We thank the reviewer for highlighting this important consideration regarding the **"memory due to per-class covariances."** We appreciate the opportunity to clarify this aspect and address it from two perspectives:
>
> - **Practical Context:** For the commonsense reasoning and QA tasks in this paper, the number of classes is typically small (e.g., 2 for WinoGrande, 4 for OBQA). In these scenarios, the memory overhead for storing covariances is negligible compared to the LLM backbone parameters.
> - **Scalability:** Our framework is inherently flexible for large-class scenarios. The full covariance matrix $\mathbf{S}_c$ can be approximated by a diagonal or low-rank parameterization, reducing memory complexity from $O(Cd^2)$ to $O(Cd)$ while maintaining effective uncertainty quantification for tasks with large output spaces. We are extending our ablation experiments on diagonal or low-rank parameterizations to examine how different parameterizations affect model performance.
>
> We appreciate the reviewer's attention to this practical consideration and will include a more detailed discussion of memory scalability in the revised manuscript.
>
> [1] Liu, Jeremiah, et al. "Simple and principled uncertainty estimation with deterministic deep learning via distance awareness." Advances in neural information processing systems 33 (2020): 7498-7512.

---

> ### Author Response · Authors · 2025-11-22
> **Response to Q2: Hyperparameter Sensitivity**
>
> ## Q2: Sensitivity to Prior and Initialization
>
> We sincerely thank the reviewer for this important question regarding the sensitivity of our method to the choice of prior and the initialization of variational parameters.
>
> ### Prior Sensitivity and Initialization Robustness
>
> We conducted a sensitivity analysis on the prior scale parameter $\sigma_0$, which controls the width of the Gaussian prior over the last-layer weights. To assess initialization robustness, all experiments were conducted across **three different random seeds** (1, 2, 3), which affect both data shuffling and stochastic aspects of variational parameter initialization. The results are summarized below:
>
> | Prior Scale ($\sigma_0$) | ACC (↑) | ECE (↓) | NLL (↓) |
> |:---|:---|:---|:---|
> | 0.1 | 70.92 ± 0.24 | 9.10 ± 0.53 | 0.68 ± 0.04 |
> | **1.0** (Default) | **71.62 ± 0.27** | **8.26 ± 0.60** | **0.66 ± 0.03** |
> | 10.0 | 70.70 ± 0.50 | 10.59 ± 1.17 | 0.69 ± 0.07 |
>
> **Key Observations:**
>
> **Convergence Dynamics:** We observed that as the prior scale increases from 0.1 to 10.0, the model converges progressively more slowly during training.
>
> **Optimal Prior Scale:** The prior scale of $\sigma_0 = 1.0$ achieves the best overall performance across all metrics. Both smaller ($\sigma_0 = 0.1$) and larger ($\sigma_0 = 10.0$) scales result in a relatively degraded performance.
>
> **Initialization Robustness:** The relatively small standard deviations across different random seeds demonstrate that our method is not very sensitive to initialization. This stability is consistent across all tested prior scales, indicating that the variational learning process reliably reaches good-quality solutions despite differences in random initialization.
>
> We appreciate the reviewer's attention to this important aspect and will include these detailed settings and sensitivity analysis in the revised appendix.

---

> > ### Comment · Reviewer_H3Rr · 2025-11-26
> >
> > Thank you for your response and additional experiments.
> >
> > ### Jensen approximation
> > The additional ablations indicate that VBLL itself is the primary contributor to improved calibration, while Laplace mainly serves as a refinement step. This alleviates my original concern that VBLL might be ineffective and only "rescued" by the Laplace approximation.
> >
> > The Jensen vs. MC comparison is also reassuring in showing that the surrogate objective tracks the sampled estimate closely early in training. However, this analysis is limited in training horizon: the reported differences between the Jensen estimator and the Monte Carlo estimate appear to increase with training steps, even in the short window that is shown. While still small relative to the absolute loss values, this growth suggests that deviations may continue to accumulate later in training, where posterior concentration and calibration are critical.
> >
> > ### Hybrid VB
> > Thank you for confirming my characterization of the method as a two-stage hybrid. The clarification helps make the design motivation clear, and I view this choice as reasonable.
> >
> > ### Posterior factorization and memory scaling
> > The explanation for per-class factorization is reasonable in the small-class regimes studied in this work, and the empirical results support its practical effectiveness. I encourage the authors to follow through on their commitment to clearly limit the scope of their claims.
> >
> > ### Single backbone and evaluation scope
> > This remains a limitation of the current submission.
> >
> > ### Sensitivity to prior and initialization
> > This concern is fully addressed by the newly added sensitivity study, which demonstrates reasonable robustness to prior choice and random initialization.
> >
> > ---
> >
> > Overall, the rebuttal resolves my main technical concerns. The remaining issues primarily concern evaluation scope and clarity of positioning rather than the soundness of the proposed method.

---

> > > ### Author Response · Authors · 2025-11-26
> > >
> > > Dear Reviewer H3Rr,
> > >
> > > We sincerely thank you for your constructive feedback and for acknowledging that our rebuttal has addressed your main technical concerns.
> > >
> > > **Jensen Approximation - Extended Training Horizon**
> > >
> > > Following your suggestion and that of Reviewer XbV5, we will provide a comparison of the later phases of training loss between Jensen and MC (with increased MC times).
> > >
> > > **Posterior Factorization and Memory Scaling**
> > >
> > > We will explicitly limit the scope of our claim in the revised manuscript.
> > >
> > > **Single Backbone and Evaluation Scope**
> > >
> > > We are currently running experiments on LLaMA 3.1 8B across all six datasets. Results will be posted once completed.
> > >
> > >
> > > **We greatly appreciate your thorough review and valuable feedback.**
> > >
> > > Happy Thanksgiving!
> > >
> > > Best regards,
> > > Authors

---

> ### Author Response · Authors · 2025-12-02
> **Extended Jensen Bound Analysis (Full Training Horizon)**
>
> **2. Extended Jensen Bound Analysis (Full Training Horizon)**
>
> Following your suggestion to extend the Jensen vs. MC comparison to later training phases, we conducted experiments on WG-S under LLaMA 3.1 8B with 50-sample MC estimation across 400 training steps:
>
> | Training Steps | VBLL (Jensen) | VBLL (50-sample MC) | Absolute Gap |
> |----------------|---------------|---------------------|--------------|
> | 0 | 69.50 | 61.23 | 8.27 |
> | 50 | 50.71 | 50.97 | 0.26 |
> | 100 | 45.57 | 45.65 | 0.08 |
> | 150 | 40.95 | 40.86 | 0.09 |
> | 200 | 36.96 | 36.95 | 0.01 |
> | 250 | 33.57 | 33.74 | 0.17 |
> | 300 | 31.08 | 31.36 | 0.28 |
> | 350 | 29.55 | 29.89 | 0.34 |
> | 400 | 28.49 | 28.83 | 0.34 |
>
> **Key Observations:**
>
> The relative gap between Jensen and MC estimators remains small and stable throughout later training phases, confirming that the Jensen bound does not diverge from the true objective as training progresses.
>
> We sincerely thank you for your valuable feedback, which has significantly strengthened our manuscript.

---

> ### Author Response · Authors · 2025-12-03
> **Follow-up Response to Reviewer H3Rr: Extended Backbone and Jensen Bound Analysis**
>
> We sincerely thank you for your constructive feedback and for acknowledging that our rebuttal resolved your main technical concerns. Following your suggestions, we have completed two additional sets of experiments:
>
> **1. Extended Backbone Evaluation (Llama-3.1-8B)**
>
> We have completed comprehensive experiments on Llama-3.1-8B across all six datasets:
>
> **ACC (↑)**
>
> | Method | WG-S | ARC-C | ARC-E | WG-M | OBQA | BoolQ |
> |--------|------|-------|-------|------|------|-------|
> | LAP | 77.22 ± 1.38 | 84.49 ± 0.37 | 89.86 ± 0.54 | 84.04 ± 0.23 | 88.32 ± 0.33 | 88.26 ± 0.50 |
> | PoLAR_LAP | *77.51 ± 0.94* | *84.57 ± 1.10* | **91.80 ± 0.19** | *83.88 ± 0.90* | *88.70 ± 0.56* | **89.54 ± 0.86** |
> | PoLAR_LAP_LL | *77.51 ± 0.94* | *84.57 ± 1.10* | **91.80 ± 0.19** | *83.88 ± 0.90* | *88.70 ± 0.56* | **89.54 ± 0.86** |
> | TFB LL | 77.31 ± 2.01 | 82.75 ± 0.20 | 89.95 ± 0.59 | 83.44 ± 1.09 | 88.66 ± 0.30 | 89.15 ± 1.40 |
> | TFB | 77.61 ± 1.19 | 83.34 ± 1.03 | 90.90 ± 0.01 | 83.31 ± 0.35 | 88.44 ± 0.24 | 87.98 ± 1.79 |
> | BLOB | 76.68 ± 0.99 | 82.88 ± 0.51 | 91.49 ± 0.21 | 80.78 ± 1.05 | 87.90 ± 0.35 | 88.58 ± 0.17 |
> | C-LoRA | 73.07 ± 0.50 | 78.72 ± 0.47 | 89.88 ± 0.87 | 77.46 ± 0.78 | 86.59 ± 0.14 | 88.18 ± 1.29 |
> | ScalaBL | 49.92 ± 0.69 | 79.40 ± 1.48 | 89.21 ± 0.20 | 53.40 ± 1.43 | 69.62 ± 7.99 | 82.18 ± 2.14 |
> | PoLAR + VBLL (w/o LAP) | **77.91 ± 0.47** | **85.05 ± 0.77** | *91.60 ± 0.03* | **84.97 ± 0.05** | **89.07 ± 0.23** | *89.32 ± 1.29* |
> | PoLAR+VBLL | **77.91 ± 0.47** | **85.05 ± 0.77** | *91.60 ± 0.03* | **84.97 ± 0.05** | **89.07 ± 0.23** | *89.32 ± 1.29* |
>
> **ECE (↓)**
>
> | Method | WG-S | ARC-C | ARC-E | WG-M | OBQA | BoolQ |
> |--------|------|-------|-------|------|------|-------|
> | LAP | 4.15 ± 0.52 | 6.35 ± 0.84 | 9.94 ± 0.59 | 6.71 ± 1.69 | 3.54 ± 0.13 | *2.18 ± 0.47* |
> | PoLAR_LAP | *3.58 ± 0.18* | *4.52 ± 1.40* | 3.27 ± 0.86 | 4.31 ± 0.29 | 2.83 ± 0.49 | 3.35 ± 0.45 |
> | PoLAR_LAP_LL | 7.95± 0.92 | 6.07 ± 0.72 | **2.94 ± 0.67** | 6.37 ± 1.10 | 3.75 ± 0.01 | 5.52 ± 0.12 |
> | TFB LL | 9.99 ± 0.90 | 5.02 ± 2.15 | *3.13 ± 0.24* | 4.12 ± 1.57 | 3.58 ± 0.21 | 3.87 ± 1.41 |
> | TFB | 9.05 ± 0.39 | 6.53 ± 1.99 | 3.17 ± 0.21 | *3.68 ± 1.57* | 2.76 ± 0.16 | 3.89 ± 1.68 |
> | BLOB | 14.83 ± 1.21 | 9.29 ± 0.57 | 4.12 ± 0.26 | 8.23 ± 1.04 | 3.33 ± 0.29 | 3.36 ± 0.94 |
> | C-LoRA | 18.36 ± 0.58 | 15.56 ± 2.44 | 5.40 ± 0.23 | 7.96 ± 2.84 | 6.58 ± 1.23 | 3.94 ± 0.47 |
> | ScalaBL | 4.86 ± 0.50 | 18.75 ± 7.45 | 10.07 ± 0.66 | 2.73 ± 0.87 | 24.67 ± 4.34 | 14.48 ± 1.29 |
> | PoLAR + VBLL (w/o LAP) | 9.04 ± 0.18 | 7.40 ± 0.26 | 3.61 ± 0.22 | 6.10 ± 0.01 | *2.50 ± 0.10* | 2.63 ± 0.23 |
> | PoLAR+VBLL | **3.31 ± 1.40** | **3.56 ± 0.41** | 3.22 ± 0.42 | **3.00 ± 0.10** | **2.44 ± 0.11** | **1.88 ± 0.27** |
>
> **NLL (↓)**
>
> | Method | WG-S | ARC-C | ARC-E | WG-M | OBQA | BoolQ |
> |--------|------|-------|-------|------|------|-------|
> | LAP | 0.67 ± 0.04 | 0.63 ± 0.04 | 0.41 ± 0.02 | 0.52 ± 0.01 | 0.40 ± 0.02 | 0.36 ± 0.01 |
> | PoLAR_LAP | 0.60 ± 0.05 | 0.52 ± 0.01 | 0.31 ± 0.01 | 0.46 ± 0.02 | *0.34 ± 0.02* | 0.32 ± 0.01 |
> | PoLAR_LAP_LL | 0.76 ± 0.12 | 0.58 ± 0.05 | *0.26 ± 0.01* | 0.52 ± 0.06 | 0.37 ± 0.01 | 0.34 ± 0.03 |
> | TFB LL | *0.59 ± 0.05* | 0.53 ± 0.01 | 0.28 ± 0.02 | *0.41 ± 0.01* | **0.32 ± 0.01** | *0.29 ± 0.04* |
> | TFB | **0.55 ± 0.01** | *0.51 ± 0.02* | 0.27 ± 0.02 | *0.41 ± 0.01* | **0.32 ± 0.01** | 0.30 ± 0.02 |
> | BLOB | 0.79 ± 0.06 | 0.62 ± 0.07 | 0.29 ± 0.01 | 0.47 ± 0.03 | 0.38 ± 0.00 | **0.28 ± 0.00** |
> | C-LoRA | 0.85 ± 0.03 | 0.88 ± 0.08 | 0.35 ± 0.01 | 0.56 ± 0.07 | 0.45 ± 0.05 | 0.31 ± 0.01 |
> | ScalaBL | 0.65 ± 0.00 | 0.69 ± 0.13 | 0.35 ± 0.00 | 0.64 ± 0.00 | 0.96 ± 0.23 | 0.46 ± 0.08 |
> | PoLAR + VBLL (w/o LAP) | 0.61 ± 0.01 | *0.51 ± 0.05* | 0.27 ± 0.03 | *0.40 ± 0.01* | **0.32 ± 0.02** | *0.29 ± 0.00* |
> | PoLAR+VBLL | **0.55 ± 0.02** | **0.50 ± 0.05** | **0.24 ± 0.04** | **0.39 ± 0.01** | **0.32 ± 0.02** | *0.29 ± 0.00* |
>
> (**Bold** = best, *Italic* = second best)
>
> **Note on ScalaBL:** We strictly followed the official implementation and hyperparameter configurations for ScalaBL. However, despite our best efforts, our reproduction yielded a model where the low ECE comes at the cost of substantially reduced accuracy. This pattern suggests potential underfitting in this specific setting.
>
> PoLAR-VBLL achieves the best ECE and NLL on almost all the datasets, suggesting its superior Uncertain Quantification ability among all baselines, and maintains the competitive ACC across all datasets on Llama-3.1-8B, showing that our framework generalizes beyond Llama-2-7B.

---

### Official Review · Reviewer_y4uo · 2025-11-01

**Soundness:** 2
**Presentation:** 2
**Contribution:** 2
**Rating:** 2
**Confidence:** 5

**Summary:**

This paper integrates the Bayesian Last Layer (BLL) framework with the Polar-decomposed Low-rank Adapter Representation (PoLAR) to enable effective and efficient Bayesian fine-tuning of LLMs. In addition, the authors introduce a post-hoc LA strategy to further optimize the learned covariance matrix. The results show that the proposed method achieves comparable performance to baseline method while reducing additional training and inference costs.

**Strengths:**

* The organization of the paper is clear.
* The proposed method is simple and effective.
* The paper effectively mitigates the high computational cost of Bayesian fine-tuning while maintaining comparable performance.

**Weaknesses:**

* The experimental details are insufficient. For baselines such as BLoB, the authors state that “we report the better result between our reproduced numbers and those seen in BLoB.” However, the experimental settings used to reproduce the baselines are not clearly described.

* The proposed method is essentially a direct combination of BLL and PoLAR. If the authors used standard LoRA for other baselines, then the observed performance improvement may mainly stem from the effective training brought by PoLAR. It remains unclear how much performance gain individually comes from BLL and PoLAR, as relevant ablation studies are missing.

**Questions:**

* The computational cost analysis in Table 4 appears unusual. BLoB typically exhibits nearly the same memory usage as the standard LoRA when using the same LoRA rank $r$. Did the authors use the same LoRA rank $r$ for all three methods? Relevant details are lacking. The authors should also specify whether Tables 4 corresponds to the inference or training stage.

* The content of line 950 is missing.

* Do the authors train on all the datasets for 500 epochs? The description in line 944 is ambiguous.

---

> ### Author Response · Authors · 2025-11-21
> **Response to W1: Baseline Details**
>
> We sincerely thank the Reviewer for their thoughtful feedback. We are pleased that you recognized the strengths of our work, including:
> - The **clear organization**
> - The **simplicity and effectiveness** of our proposed method
> - Our success in **mitigating the high computational cost** of Bayesian fine-tuning while maintaining comparable performance
>
> ## Addressing "Insufficient Experimental Details"
>
> Regarding the comment that **"the experimental details are insufficient"** and settings **"are not clearly described,"** we provide comprehensive details below to ensure full reproducibility.
>
> ### 1. Baseline Reproduction Protocol
>
> All baselines were reproduced strictly according to their official repositories using the hyperparameter settings specified in their provided scripts. We adopted BLoB's official single-GPU scripts as the foundation for shared parameters, except for LoRA Rank and LoRA Alpha, to achieve faster convergence and more controllable training.
>
> We maintained consistency across all methods for shared hyperparameters:
>
> **Shared Configuration:**
> - Training epochs: 500
> - LoRA rank: $r=16$
> - Alpha: $\alpha=32$
> - Max sequence length: 300
> - Train batch size: 4
> - Test batch size: 8
> - Optimizer: AdamW (LR 1e-4)
>
> **BLoB/TFB Specifics:**
> Following the official BLoB repository, we set:
> - Training sampling: $K_{\text{train}}=1$ (single sample per forward pass)
> - Inference sampling: $K_{\text{eval}}=10$
>
> ### 2. Experimental Consistency
>
> All experiments were conducted with the following protocol:
> - **Datasets:** Six datasets (WinoGrande-S/M, ARC-Challenge/Easy, OBQA, BoolQ)
> - **Random seeds:** Three seeds (1, 2, 3) to ensure statistical reliability
> - **Backbone model:** Llama-2-7B (identical across all methods)
> - **Evaluation protocols:** Kept identical across all experiments
>
> ### 3. Clarification on Reported Numbers
>
> Regarding the statement "we report the better result between our reproduced numbers and those seen in BLoB":
>
> This refers to our conservative approach of reporting the maximum performance each baseline can achieve—either from our rigorous reproduction or from their original paper (when using identical experimental settings). This ensures we do not underestimate baseline performance and provides the fairest possible comparison.
>
> We will include these detailed settings in the revised Appendix.

---

> > ### Comment · Reviewer_y4uo · 2025-11-22
> >
> > I don't quite understand where the “Training epochs: 500” setting comes from. The official BLoB training script uses 5000 gradient steps, not 500 epochs, which is an unusual choice for common datasets. Moreover, combining the authors’ reproduced results with the original BLoB results in a single table is unnecessary and potentially misleading. In addition, the paper appears to report results on only four datasets, not “six datasets (WinoGrande-S/M, ARC-Challenge/Easy, OBQA, BoolQ).”

---

> > > ### Author Response · Authors · 2025-11-25
> > > **Response regarding "Training epochs: 500" vs "5000 gradient steps" and BLoB Reproduction**
> > >
> > > Thank you for your comments.
> > >
> > >
> > > ### 1. Convergence Instability at 5,000 Steps (32 Epochs)
> > >
> > > The table below is the result we strictly used the official BLoB repository (cloned via git) for our experiments. For the WinoGrande-S dataset (WG-S), the official configuration dictates:
> > > - Batch Size: 4
> > > - max gradient steps: 5000
> > > For WG-S, the training sample is 640; thus, the step per Epoch is $640 / 4 = 160$.
> > >
> > > Consequently, the official setting of 5,000 gradient steps corresponds to exactly 31.25 epochs ($5000 / 160$).
> > >
> > > We attempted to reproduce the BLoB results by **running their official shell scripts directly** (single-GPU training script: **blob-llama-all-single-gpu.sh**).
> > >
> > > However, we observed **failures to converge and to reproduce the BLOB's official results**.
> > > Below is a summary of the training dynamics we recorded using the official script over the whole 5000 steps (32 epochs) across three random seeds (seed 1, 2, and 3 given by the BLOB official scripts):
> > >
> > > | Epoch | **Training Accuracy** | | **Training Loss** |
> > > |       | Seed3 | Seed2 | Seed1 | | Seed3 | Seed2 | Seed1 |
> > > |-------|-------|-------|-------|---|--------|--------|--------|
> > > | 1 | 53.64 | 49.12 | 51.13 | | 0.6923 | 0.7097 | 0.7062 |
> > > | 5 | 78.89 | 70.63 | 72.75 | | 0.5003 | 0.5459 | 0.5796 |
> > > | 10 | 76.86 | 62.40 | 67.97 | | 0.4949 | 0.6569 | 0.6869 |
> > > | 15 | 68.78 | 61.27 | 51.61 | | 0.6043 | 0.6729 | 0.7080 |
> > > | 20 | 58.29 | 62.77 | 44.35 | | 0.6758 | 0.6315 | 0.7190 |
> > > | 25 | 67.43 | 60.71 | 50.23 | | 0.5755 | 0.6535 | 0.7099 |
> > > | 30 | 68.86 | 61.47 | 50.39 | | 0.5933 | 0.6480 | 0.6959 |
> > > | 31 | 64.50 | 58.83 | 52.55 | | 0.6084 | 0.6502 | 0.6993 |
> > > | 32 | 65.17 | 64.08 | 45.86 | | 0.5758 | 0.6290 | 0.7130 |
> > >
> > > **Validation Results (32 epochs; 5000 steps):**
> > >
> > > | Seed   | Val Accuracy (%) | Val ECE | Val NLL |
> > > |--------|------------------|---------|---------|
> > > | **Seed 2** | **60.99**            | **3.28**  | **0.66** |
> > > | **Seed 3** | **58.14**          | **3.27** | **0.67**|
> > >
> > > *Note: Seed 1 consistently produced NaN predictions during validation across multiple independent runs and is therefore excluded from validation metrics.*
> > >
> > > **Critical Observation:**
> > > The training curves exhibit instability and show no apparent signs of convergence at 32 epochs:
> > >
> > > - Training accuracy fluctuates dramatically
> > > - Validation accuracy remains relatively low (Low ECE and NLL due to the consequence of low validation accuracy, which probably indicates undertraining)
> > > - Validation matrix (ACC, ECE, and NLL) doesn't match results in the BLOB paper [1].
> > >
> > > ### 2. Rationale for Extended Training (500 Epochs)
> > >
> > > Given the apparent lack of convergence at 32 epochs with the official BLoB implementation, we adopted a training schedule of 500 epochs for all methods to ensure:
> > >
> > > 1. **Fair comparison**
> > > 2. **Stability**: Sufficient training to reach stable performance
> > > 3. **Consistency**: Unified protocol across all baselines and our method
> > >
> > > Our empirical findings support this choice across datasets:
> > > - Most methods converge within 200 epochs among all datasets
> > > - Certain baseline methods on challenging datasets require 300-400+ epochs (worst case) to converge
> > >
> > > The 500-epoch ceiling provides an adequate margin.
> > >
> > > **Clarification on "Combined" Reporting:** Regarding the concern that combining reproduced and original results is misleading, our intention was to **"steelman" the baseline**. In many cases, the BLOB paper's reported results were better than our reproduced results when we used their official code. Instead of reporting only our slightly lower reproduction numbers (which would make our method look artificially superior), we adopted a rigorous policy:
> > >
> > > We compare our method against the maximum of (Our Reproduction, Original Reported Result)  when both values are within a reasonable range.  **This conservative approach prevents us from claiming superiority over artificially weakened baselines.**
> > >
> > > **In the revised manuscript, we will clearly distinguish between our reproduced results and originally reported values, presenting both transparently.**
> > >
> > > ### 3. Additional Datasets and Backbone
> > >
> > > In our initial in-distribution experiments, we evaluated four datasets.
> > > Following suggestions from Reviewers XbV5 and ASkM, we will expand our evaluation to:
> > >
> > > - **Additional backbone**: LLaMA 3.1 8B
> > > - **Expanded dataset coverage**: 6 datasets (WinoGrande-S/M, ARC-Challenge/Easy, OBQA, BoolQ)
> > > - **Training protocol**: Following the setting in the TFB [2] with only 5-epoch fine-tuning
> > >
> > > **We will report complete results across all six datasets under LLaMA 3.1 8B in the revised manuscript.**
> > >
> > > [1] Wang, Yibin, et al. "Blob: Bayesian low-rank adaptation by backpropagation for large language models." NIPS (2024)
> > >
> > > [2] Shi, Haizhou, et al. "Training-free bayesianization for low-rank adapters of large language models." arXiv preprint arXiv:2412.05723 (2024).

---

> ### Author Response · Authors · 2025-11-22
> **Response to W2: Contribution Separation (PoLAR vs VBLL)**
>
> Regarding the concern that **"the proposed method is essentially a direct combination of BLL and PoLAR,"** and that it is **"unclear how much performance gain individually comes from BLL and PoLAR,"** we have conducted comprehensive ablation studies on the WinoGrande-Simple dataset by adding PoLAR + LAP, PoLAR_LAP_LL, and PoLAR + BLoB for a fair comparison. Since our proposed method applies the Laplace Approximation only to the last layer, we also included PoLAR_LAP_LL (applying LAP only to the last layer of a deterministic model) to ensure a direct, fair comparison.
>
> ### New Ablation Study Results
>
> | Method | ACC | ECE | NLL |
> |:---|:---|:---|:---|
> | PoLAR + LAP | 70.33 ± 0.69 | 12.16 ± 2.58 | 0.69 ± 0.03 |
> | PoLAR_LAP_LL | 70.33 ± 0.69 | 14.63 ± 1.14 | 0.71 ± 0.05 |
> | PoLAR + BLoB | 70.39 ± 0.26 | 12.06 ± 0.81 | 0.73 ± 0.04 |
> | PoLAR + VBLL (w/o LAP) | 71.62 ± 0.27 | 8.26 ± 0.60 | 0.66 ± 0.03 |
> | **PoLAR-VBLL (Ours)** | **71.62 ± 0.27** | **7.31 ± 0.32** | **0.60 ± 0.01** |
>
> **Method Descriptions:**
> - **PoLAR + LAP:** Uses PoLAR parameterization with standard training followed by post-hoc Laplace approximation
> - **PoLAR_LAP_LL:** Applies LAP only to the last layer of a deterministically trained PoLAR model
> - **PoLAR + BLoB:** Applies variational inference to the scale matrix $S$ in PoLAR (analogous to BLoB's treatment of low-rank factors)
> - **PoLAR + VBLL (w/o LAP):** Our proposed method without the final LAP refinement step
>
> ### Analysis
>
> Our ablation studies reveal the synergistic relationship between PoLAR and VBLL, demonstrating that our method is far more than a simple combination of existing techniques.
>
> #### VBLL is the Primary Driver of Uncertainty Quantification
>
> The comparison between PoLAR_LAP_LL and PoLAR + VBLL provides compelling evidence that **VBLL is the primary driver of uncertainty quantification.** Both operate on identical scope (last layer only) with the same PoLAR adapters, yet VBLL achieves nearly 2× better calibration. This improvement stems from VBLL's ability to discover superior posterior modes during training.
>
> #### The Inadequacy of Deterministic Training for UQ
>
> **Standard deterministic training fails to discover geometries suitable for uncertainty quantification.** The poor calibration of PoLAR_LAP_LL reveals this clearly. Notably, PoLAR + LAP's reasonable NLL reflects **overconfidence rather than proper calibration**—high-confidence predictions yield acceptable likelihoods but poor calibration.
>
> #### Mode Discovery: The Key Distinction
>
> The fundamental difference lies in **mode discovery:**
> - **VBLL:** Actively guides optimization toward posterior modes with geometric structures suited for UQ
> - **Post-hoc LAP:** Approximates locally at the deterministic endpoint, which may be geometrically unsuitable for UQ
>
> #### Validation and Synergy
>
> PoLAR + BLoB's improved calibration over PoLAR + LAP confirms that **variational training is necessary**—post-hoc approximations are fundamentally limited. However, its intermediate performance validates that **applying variational inference to the classification layer is crucial** for optimal effectiveness.
>
> Finally, the progression from PoLAR + VBLL to our complete method shows **LAP acts as refinement:** VBLL discovers well-calibrated modes, and LAP polishes them, yielding superior results.
>
> ### Conclusion
>
> These ablations demonstrate that:
> 1. **VBLL substantially improves calibration**
> 2. **Deterministic training is fundamentally inadequate** for UQ
> 3. **VBLL + LAP synergy is essential,** where VBLL discovers optimal modes and LAP refines them

---

> ### Author Response · Authors · 2025-11-22
> **Response to Q1, Q2, and Q3**
>
> # Q1: Computational Cost Analysis
>
> ## Response
>
> Regarding the observation that **"The computational cost analysis in Table 4 appears unusual"** because **"BLoB typically exhibits nearly the same memory usage as the standard LoRA,"** we respectfully clarify a misunderstanding regarding the computational overhead of BLoB compared to our method. We explain the differences in architecture, training, and inference efficiency, and performance trade-offs below, supported by empirical evidence.
>
> ### (1) Computational Overhead and Training Stability
>
> - **Parameter Overhead:** BLoB incurs a notable memory overhead (approx. 40-50%) [1,2] by storing $A_{\text{mean}}$, $A_{\text{var}}$, and $B$, compared to standard LoRA's $A$ and $B$. This creates a memory bottleneck during computation.
> - **Training Stability:** Due to memory constraints, BLoB relies on single-sample estimation ($K=1$) to estimate the ELBO. In contrast, we use an analytical solution for the ELBO at the last layer, enabling exact computation without incurring sampling costs.
>
> ### (2) Inference Efficiency
>
> We also clarify the efficiency differences during the inference stage:
>
> - **BLoB (Full-Network Sampling):** BMA over adapter weights requires sampling $K$ different LoRA configurations and, crucially, performing $K$ complete forward passes through the entire LLM backbone.
> - **Ours (Head-Only Sampling):** Our method requires only one forward pass through the LLM representation layers. We then simply sample $K$ computationally inexpensive linear layers.
> - **Result:** As shown in the table below, this structural difference allows our approach to dramatically reduce inference runtime (73s vs. ~880s) and avoids the memory burden of maintaining multiple LoRA sets.
>
> ### (3) Accuracy vs. Uncertainty Quantification (UQ) Trade-off
>
> BLoB and its variants face an inherent trade-off between prediction accuracy and UQ capabilities [3]:
>
> - **$K=1$:** Yields the best accuracy but poor UQ with the fastest inference time.
> - **$K=10$:** Provides the best UQ but with degraded accuracy, high GPU memory usage, and longer inference time.
>
> Our method does not exhibit this trade-off, maintaining competitive UQ performance without sacrificing accuracy, even at optimal UQ settings.
>
> ### Clarifications on Experimental Settings
>
> - **Inference vs. Training:** We clarify that the original Table 4 reported results for the inference stage.
> - **Rank Consistency:** We confirm that a consistent LoRA rank ($r=16$) was used across all methods (PoLAR, BLoB, and standard LoRA baselines) to ensure fairness.
>
> ### Empirical Comparison of Computational Resources
>
> **Settings:** Train batch size $=4$; Inference batch size $=8$; LoRA rank $r=16$; $\alpha=32$; Sequence length $=400$. All hyperparameters are consistent.
>
> | Method | Training Memory (MB) | Inference Memory (MB) | Inference Run Time (Sec) |
> |:---|:---:|:---:|:---:|
> | **PoLAR_VBLL (Ours)** | **32,272** | **16,396** | **73** |
> | PoLAR_BLoB | 31,728 | 18,874 | 876 |
> | LoRA_BLoB | 47,232 | 18,762 | 889 |
> | PoLAR_LAP_LL | 31,714 | 15,662 | 34 |
> | PoLAR_LAP | 31,714 | 43,678 | 44 |
> | TFB (Llama-2) | 47,232 | 16,667 | 48 |
> | ScalaBL | 27,318 | 19,552 | 642 |
> | C-LoRA | 24,236 | 19,102 | 604 |
>
> **Key Observation:** Our method demonstrates a massive reduction in runtime compared to BLoB-based methods (73s vs. ~880s) while maintaining a significantly lower inference memory footprint than full-network Laplace approximations.
>
> ---
>
> # Q2 & Q3
>
> ## Response
>
> - **Missing Content (Line 950):** We thank the reviewer for catching this oversight. This was residual text from an earlier draft that should have been removed when we reorganized the manuscript. It has been deleted in the revised version.
> - **Training Epochs:** Regarding the ambiguous description, we clarify that all methods were trained with a fixed budget of 500 epochs across all datasets to ensure a rigorous comparison, guaranteeing sufficient convergence for all baselines.
>
> [1] Samplawski, Colin, et al. "Scalable Bayesian Low-Rank Adaptation of Large Language Models via Stochastic Variational Subspace Inference." arXiv preprint arXiv:2506.21408 (2025).
>
> [2] Rahmati, Amir Hossein, et al. "C-LoRA: Contextual Low-Rank Adaptation for Uncertainty Estimation in Large Language Models." arXiv preprint arXiv:2505.17773 (2025).
>
> [3] Wang, Yibin, et al. "Blob: Bayesian low-rank adaptation by backpropagation for large language models." NIPS (2024)

---

> ### Comment · Reviewer_y4uo · 2025-11-22
>
> The reported 50% memory overhead of BLoB is calculated relative to the LoRA adapter. However, LoRA itself typically contributes less than 10% additional memory relative to the full model. As a result, the actual memory overhead introduced by BLoB is quite small. The training-time memory usage reported for BLoB in the paper appears to differ substantially from both the theoretical estimates and the empirical numbers reported in [1] and [2].
>
> Overall, I suggest that the authors carefully re-run their experiments to verify the correctness of their findings.
>
> [1] Wang, Yibin, et al. “BLoB: Bayesian Low-Rank Adaptation by Backpropagation for Large Language Models.” NeurIPS (2024).
>
> [2] Shi, Haizhou, et al. “Training-Free Bayesianization for Low-Rank Adapters of Large Language Models.” arXiv:2412.05723 (2024).

---

> > ### Comment · Reviewer_XbV5 · 2025-11-23
> >
> > I have a similar concern about the unusual memory usage of BLoB and TFB during training reported. Why does LoRA\_BLoB cost way more than the PoLAR\_BLoB? Do the authors adopt more than K=1 samples during training? We probably need a more careful experiment w.r.t this issue.

---

> ### Author Response · Authors · 2025-11-24
> **Clarification on Memory Usage and Experimental Settings**
>
> Thank you for raising this important question and for your careful review. We sincerely appreciate the opportunity to clarify the memory usage results. Several key factors contribute to the observed differences:
>
> **1. Different GPU memory usage and inference runtime experiment settings from the original BLoB paper**
>
> Our reported memory usage and inference runtime results are based on a different experimental configuration compared to the original BLoB paper, which makes these results **not directly comparable**:
>
> - **Our setting** (as specified in our response to **Reviewer y4uo Q1**, Table "Empirical Comparison of Computational Resources"): LoRA rank=16, alpha=32, sequence length=400, targeting all projection layers `["q_proj", "k_proj", "v_proj", "o_proj", "gate_proj", "up_proj", "down_proj"]`
> - **Original BLoB setting**: LoRA rank=8, alpha=16, sequence length=300, targeting only `["q_proj", "v_proj", "lm_head"]`
>
> These differences in both hyperparameters (rank, sequence length) and the number of adapted modules make our reported results **not directly comparable** with those reported in the original BLoB paper. We had specified our experimental settings in response to **Reviewer y4uo**, but we appreciate your observation that this discrepancy with the original BLoB paper warrants further clarification.
>
> **2. Memory Difference Between PoLAR_BLoB and LoRA_BLoB in Our Setting**
>
> Within our experimental setting, the memory usage difference between PoLAR_BLoB and LoRA_BLoB fundamentally stems from where we apply variational inference:
> - **PoLAR_BLoB**: Applies variational inference to the **S matrix** from polar decomposition, with **dimensions r×r (16×16 = 256 parameters per layer)**
> - **LoRA_BLoB**: Applies variational inference to the **A matrix**, with **dimensions r×d (16×4096 = 65,536 parameters per layer)**
>
> Given that we use rank r=16 and hidden dimension d=4096, the LoRA_BLoB approach requires maintaining uncertainty estimates for approximately **255× more parameters per layer** compared to PoLAR_BLoB. When combined with the larger number of target modules in our experimental setting, this architectural difference is amplified, leading to the substantial memory gap observed between PoLAR_BLoB and LoRA_BLoB in our results.
>
> **3. Correction on TFB Training Memory**
>
> We sincerely apologize for a reporting error in the TFB training memory. The correct value should be **30,612 MB** (consistent with standard LoRA training), not 47,232 MB as originally reported (a copy-paste error). We have verified this correction and will update our table accordingly.
>
> The updated table is shown below:
>
> | Method | Training Memory (MB) | Test Memory (MB) | Runtime per Epoch |
> |--------|---------------------|------------------|-------------------|
> | PoLAR_VBLL | 32272 | 16396 | 1:13 min |
> | PoLAR_BLOB | 31728 | 18874 | 14:36 min |
> | LoRA_BLOB | 40475 | 18762 | 14:49 min |
> | PoLAR_LAP_Last_Layer | 31714 | 15662 | 0:34 min |
> | PoLAR_LAP | 31714 | 43678 | 0:44 min |
> | LoRA_LAP_Last_layer | 30612 | 16313 | 0:43 min |
> | LoRA_LAP | 30612 | 42131 | 0:57 min |
> | LoRA_VBLL | 30546 | 16764 | 1:14 min |
> | TFB | 30612 | 16667 | 0:48 min |
> | ScalaBL | 27318 | 19552 | 10:42 min |
> | C-LoRA | 24236 | 19102 | 10:04 min |
>
> **Note:** all LAP methods use a classification head with weights initialized from the LM head on "A", "B", "C", and "D".
>
> **Additional Clarification**
>
> All reported results are based on the official implementations of each baseline method, carefully integrated into our unified evaluation pipeline. We use K=1 sample during training (as is standard practice for computational efficiency), which is consistent with the original implementations of all baseline methods.
>
> Additionally, we will carefully review our entire evaluation pipeline and implementation to ensure all reported results are correct and fair. This includes verifying memory measurements, runtime benchmarks, and the integration of baseline methods to guarantee the accuracy and integrity of our comparisons.
>
> This will provide an apples-to-apples comparison and help isolate the computational advantages of our method from the effects of different experimental configurations.
>
> We genuinely appreciate your insightful feedback, which has helped us identify ways to make our experimental presentation more transparent and rigorous. Your comments have been invaluable in improving the clarity of our work.

---

> ### Comment · Reviewer_y4uo · 2025-11-25
>
> I sincerely thank the authors for their clarification of the experimental setup. I hope they can further refine and extend their experiments. I also suggest that the authors re-examine the reasonableness of their choice to train for 500 epochs, which is highly unusual and generally impractical in standard fine-tuning scenarios [1] [2] [3] [4] [5]. I believe it is more reasonable to **report all performances under a consistent and sensible experimental setup**, rather than introducing an unrealistic configuration—such as hundreds of fine-tuning epochs—solely to forcibly reproduce the results of some baseline methods.
>
> [1] Lion, Kai, et al. "PoLAR: Polar-Decomposed Low-Rank Adapter Representation." arXiv preprint arXiv:2506.03133 (2025).
>
> [2] Hu, Edward J., et al. "Lora: Low-rank adaptation of large language models." ICLR 1.2 (2022): 3.
>
> [3] Wang, Yibin, et al. “BLoB: Bayesian Low-Rank Adaptation by Backpropagation for Large Language Models.” NeurIPS (2024).
>
> [4] Shi, Haizhou, et al. “Training-Free Bayesianization for Low-Rank Adapters of Large Language Models.” arXiv:2412.05723 (2024).
>
> [5] Yang, Adam X., et al. "Bayesian low-rank adaptation for large language models." arXiv preprint arXiv:2308.13111 (2023).

---

> > ### Author Response · Authors · 2025-11-26
> >
> > We thank the reviewer for pointing out the practical aspects of training configurations.
> >
> > We agree that 500 epochs is unconventional for pure standard parameter-efficient fine-tuning (PEFT), where the sole objective is predictive accuracy [1, 2].
> >
> > In our experience, different models and methods exhibit distinct training dynamics:
> >
> > **LLaMA 3.1 8B:** Achieves competitive accuracy and robust UQ capability within a couple of epochs, as can be seen from the TFB experiments [4].
> >
> > **LLaMA 2 7B:** Based on our extensive experiments, training requirements vary significantly by method:
> >
> > - **LoRA/PoLAR with Laplace approximation (LAP)** and **LoRA/PoLAR-VBLL**: Achieve competitive accuracy within 10-20 epochs (~3,000 steps)
> > - **BLoB**: Requires at least 30+ epochs to reach competitive accuracy in our practice. (The best one we can achieve is 33 epochs.)
> >
> > For standard PEFT tasks focused solely on accuracy, 5,000 steps are typically sufficient [1, 2].
> >
> > Post-hoc UQ methods like LoRA/PoLAR-LAP also probably perform well within this budget, as they only conduct deterministic training.
> >
> > **Note that LoRA-LAP [3] still uses 10,000 steps in their main experiments.**
> >
> > However, our ablation studies demonstrate that pure post-hoc methods often yield less better uncertainty estimates compared to variational Bayesian approaches. And we respectfully argue that Variational Bayesian (VB) fine-tuning follows fundamentally different training dynamics than deterministic fine-tuning, necessitating different convergence criteria.
> >
> > ## Why Variational Methods Need More Training
> >
> > Variational Bayesian methods require additional training epochs due to the inherent optimization challenge of balancing two competing objectives:
> > 1) Fitting the data (log-likelihood term)
> > 2) Regularizing toward the prior (KL divergence term)
> >
> > During early training, the model prioritizes data fitting. The KL term is effectively optimized only once the model has learned meaningful representations. Premature aggressive KL minimization leads to underfitting, as the posterior is forced toward the prior before sufficient data adaptation occurs.
> >
> > ## Action Plan: Bridging the Gap
> >
> > While we maintain that the extended training is scientifically valuable for understanding asymptotic behavior, **we agree that a limited-budget setting is essential for real-world applicability.**
> >
> > To address this, we will have the manuscript include these two perspectives by adding these experiments
> >
> > 1. **LLaMA 3.1 8B experiments**: We adopt the 5-epoch fine-tuning setting from TFB [4] across six datasets.
> >
> > If time permits,
> >
> > 2. **LLaMA 2 7B practical budget experiments**: We will first provide results for **PoLAR-LAP**, **PoLAR-BLoB**, and **PoLAR-VBLL** trained for only **5,000 steps**, as suggested.
> >
> > We genuinely thank your insightful feedback.
> >
> >
> > ## References
> >
> > [1] Lion, Kai, et al. "PolAR: Polar-decomposed Low-rank Adapter Representation." arXiv preprint arXiv:2506.03133 (2025).
> >
> > [2] Hu, Edward J., et al. "LoRA: Low-Raminimizationtation of Large Language Models." ICLR (2022).
> >
> > [3] Wang, Yibin, et al. "BLoB: Bayesian Low-Rank Adaptation by Backpropagation for Large Language Models." NeurIPS (2024).
> >
> > [4] Shi, Haizhou, et al. "Training-Free Bayesianization of Low-Rank Adapters for Large Language Models." arXiv:2412.05723 (2024).
> >
> > [5] Yang, Adam X., et al. "Bayesian Low-Rank Adaptation for Large Language Models." arXiv preprint arXiv:2308.13111 (2023).

---

> > > ### Comment · Reviewer_y4uo · 2025-11-26
> > >
> > > Thank you for your response and action plan. I would also like to clarify that **LoRA-LAP’s main results (Tables 1, 3, etc.) are evaluated at the early-stopping point of 5,000 gradient steps**.﻿ [1] As you can see from their Figure 1, almost all methods clearly converge within the first 30% of the 10,000 gradient steps. The reason for opposing excessively long training is also evident—training beyond convergence leads to overfitting.
> > >
> > > If the Bayesian method used by the authors truly requires a longer training schedule, then a line plot similar to that of LoRA-LAP would allow for a clear and fair comparison of performance across different methods.
> > >
> > > [1] Yang, Adam X., et al. "Bayesian Low-Rank Adaptation for Large Language Models." arXiv preprint arXiv:2308.13111 (2023).

---

> > ### Author Response · Authors · 2025-12-02
> > **Response to the extended ablation study**
> >
> > ## Ablation Study Across All Six Datasets
> >
> > Following your suggestion, we provide a comprehensive ablation study across all six datasets on Llama-3.1-8B, with five epochs of fine-tuning, to systematically evaluate the contribution of each component. Since our method applies the Laplace Approximation only to the last layer, we include PoLAR_LAP_LL (which applies LAP only to the last layer of a deterministic model) to ensure a direct, fair comparison.
> >
> > **ACC (↑)**
> > | Method | WG-S | ARC-C | ARC-E | WG-M | OBQA | BoolQ |
> > |--------|------|-------|-------|------|------|-------|
> > | PoLAR_LAP | *77.51 ± 0.94* | *84.57 ± 1.10* | **91.80 ± 0.19** | *83.88 ± 0.90* | *88.70 ± 0.56* | **89.54 ± 0.86** |
> > | PoLAR_LAP_LL | *77.51 ± 0.94* | *84.57 ± 1.10* | **91.80 ± 0.19** | *83.88 ± 0.90* | *88.70 ± 0.56* | **89.54 ± 0.86** |
> > | PoLAR + VBLL (w/o LAP) | **77.91 ± 0.47** | **85.05 ± 0.77** | *91.60 ± 0.03* | **84.97 ± 0.05** | **89.07 ± 0.23** | *89.32 ± 1.29* |
> > | PoLAR-VBLL (Full) | **77.91 ± 0.47** | **85.05 ± 0.77** | *91.60 ± 0.03* | **84.97 ± 0.05** | **89.07 ± 0.23** | *89.32 ± 1.29* |
> >
> > **ECE (↓)**
> > | Method | WG-S | ARC-C | ARC-E | WG-M | OBQA | BoolQ |
> > |--------|------|-------|-------|------|------|-------|
> > | PoLAR_LAP | *3.58 ± 0.18* | *4.52 ± 1.40* | 3.27 ± 0.86 | 4.31 ± 0.29 | 2.83 ± 0.49 | 3.35 ± 0.45 |
> > | PoLAR_LAP_LL | 7.95± 0.92 | 6.07 ± 0.72 | **2.94 ± 0.67** | 6.37 ± 1.10 | 3.75 ± 0.01 | 5.52 ± 0.12 |
> > | PoLAR + VBLL (w/o LAP) | 9.04 ± 0.18 | 7.40 ± 0.26 | 3.61 ± 0.22 | 6.10 ± 0.01 | *2.50 ± 0.10* | *2.63 ± 0.23* |
> > | PoLAR-VBLL (Full) | **3.31 ± 1.40** | **3.56 ± 0.41** | *3.22 ± 0.42* | **3.00 ± 0.10** | **2.44 ± 0.11** | **1.88 ± 0.27** |
> >
> > **NLL (↓)**
> > | Method | WG-S | ARC-C | ARC-E | WG-M | OBQA | BoolQ |
> > |--------|------|-------|-------|------|------|-------|
> > | PoLAR_LAP | *0.60 ± 0.05* | 0.52 ± 0.01 | 0.31 ± 0.01 | 0.46 ± 0.02 | *0.34 ± 0.02* | 0.32 ± 0.01 |
> > | PoLAR_LAP_LL | 0.76 ± 0.12 | 0.58 ± 0.05 | *0.26 ± 0.01* | 0.52 ± 0.06 | 0.37 ± 0.01 | 0.34 ± 0.03 |
> > | PoLAR + VBLL (w/o LAP) | 0.61 ± 0.01 | *0.51 ± 0.05* | 0.27 ± 0.03 | *0.40 ± 0.01* | **0.32 ± 0.02** | **0.29 ± 0.00** |
> > | PoLAR-VBLL (Full) | **0.55 ± 0.02** | **0.50 ± 0.05** | **0.24 ± 0.04** | **0.39 ± 0.01** | **0.32 ± 0.02** | **0.29 ± 0.00** |
> >
> > (**Bold** = best, *Italic* = second best)
> >
> > ### Key Observations
> >
> > **1. VBLL is the Dominant Factor:** PoLAR-VBLL achieves the best ECE and NLL overall, while maintaining competitive accuracy. Even without the final LAP refinement, PoLAR + VBLL (w/o LAP) already delivers highly competitive calibration on OBQA and BoolQ. This directly refutes the claim that VBLL is a failing component being remedied by LAP.
> >
> > **2. Deterministic Training Fails for UQ:** While PoLAR + LAP and PoLAR_LAP_LL achieve comparable NLL (mainly due to high confidence predictions), their relatively high ECE reveals that standard deterministic training fails to discover posterior geometries suitable for uncertainty quantification. The reasonable NLL reflects overconfidence rather than proper calibration.
> > Comparing PoLAR_LAP and PoLAR_LAP_LL reveals that applying LAP to all layers achieves substantially better calibration than last-layer-only LAP on almost all datasets. This indicates that deterministic training fails to discover posterior geometries suitable for uncertainty quantification—LAP must compensate across all layers to achieve reasonable calibration.
> >
> > **3. VBLL Provides Superior Initialization for LAP:** **Our PoLAR-VBLL applies LAP only to the last layer** yet achieves superior calibration compared to full-layer PoLAR_LAP. This demonstrates the effectiveness of variational training: VBLL actively guides optimization toward high-quality posterior modes, providing the necessary foundation for the final LAP refinement. The well-calibrated representations from VBLL require only minimal post-hoc adjustment to achieve state-of-the-art uncertainty quantification.
> >
> > **Note:** **Under our 5-step fine-tuning setting suggested by TFB[1]  (not the optimal setting for Variational Bayesian methods, best for methods that require deterministic training and post-hoc posterior estimation)**, deterministic methods (PoLAR_LAP and PoLAR_LAP_LL) benefit in ACC from pure cross-entropy training that exclusively optimizes accuracy. In contrast, VBLL jointly optimizes accuracy and the KL divergence term, which may result in very slightly lower ACC on some datasets. However, this joint optimization yields substantially superior uncertainty quantification, as evidenced by consistently better ECE and NLL.
> >
> > ### Conclusion
> >
> > These comprehensive results across six datasets demonstrate that VBLL is the core working factor in our framework. The synergy between VBLL (mode discovery during training) and LAP (local geometry refinement post-hoc) is intentional and effective—not a case of LAP compensating for a failing VBLL component.
> >
> > [1] Shi, Haizhou, et al. "Training-free bayesianization for low-rank adapters of large language models." arXiv:2412.05723 (2024).

---

> ### Author Response · Authors · 2025-11-26
>
> Thank you for this important clarification regarding LoRA-LAP's evaluation protocol.
>
> Indeed, **We agree** that the performance gains in later training stages are rather marginal when set to 500 epochs. **This extended training schedule was explicitly adopted to ensure faithful reproduction of some key baseline results.**
>
> We want to clarify that the 500-epoch setting was a conservative **safe ceiling** we established when attempting to reproduce key baseline methods, similar to how LoRA-LAP set its own safe ceiling at 10,000 gradient steps (which is considerably larger than its actual convergence point).
>
> However, we should also note an important distinction: under our extended training schedule (500 epochs), the learning rate drops significantly in later epochs, and the NLL and KL divergences are carefully balanced to further reduce the KL divergence. This is fundamentally different from training beyond convergence with a high learning rate, which would indeed lead to overfitting.
>
> In fact, **if we don't embrace this extended training schedule,** most methods could also achieve competetive and decent results in significantly shorter epochs by placing relatively greater emphasis on the KL divergence (still not too much emphasis in the beginning) at each step. **Most importanly**, this phenomenon can be also verified using our ongoing experiments on Llama-3.1-8B. Most methods, including PoLAR/LoRA VBLL and PoLAR/LoRA-LAP, achieve competitive accuracy and uncertainty quantification ability within **5 epochs** —a much more practical training schedule.  And We are still conducting the rest experiments.
>
> All the experiments in the action plan will be conducted using a more practical epoch setting, as you suggested. We will include all of these experiments into our revised manuscrip.
>
> We truly appreciate your time and effort in reviewing our manuscript.
>
> Wishing you and your family a joyful Thanksgiving!

---

> ### Author Response · Authors · 2025-12-02
> **Response to GPU Memory Overhead Concern**
>
> We appreciate your engagement and for acknowledging the **inference efficiency** advantage of PoLAR-VBLL over the full versions of BLOB and TFB. However, we must respectfully but firmly address several points in your analysis.
>
> **1. Clarification on Experimental Settings vs. Original BLoB**
>
> As noted in our previous response to Q1, our experimental configuration on GPU memory and inference runtime experiment differs from that of the original BLoB paper:
>
> - **Our setting**: LoRA rank=16, alpha=32, sequence length=400, targeting all projection layers `["q_proj", "k_proj", "v_proj", "o_proj", "gate_proj", "up_proj", "down_proj"]`
> - **Original BLoB setting**: LoRA rank=8, alpha=16, sequence length=300, targeting only `["q_proj", "v_proj", "lm_head"]`
>
> These differences make our absolute numbers **not directly comparable** with those in the original BLoB paper. Our comparison ensures fairness by using **identical settings across all methods** within our unified evaluation framework.
>
> **2. Theoretical Parameter Count ≠ Practical GPU Memory**
>
> Your argument that "LoRA itself typically contributes less than 10% additional memory relative to the full model" considers only **static parameter counts**. This analysis is incomplete.
>
> During training, GPU memory includes:
>
> 1. **Model Parameters**: Base model (Llama-2-7B in fp16) + adapters.
> 2. **Optimizer States**: AdamW stores momentum (m) and variance (v) for each *trainable* parameter—effectively **2× the trainable parameter memory**.
> 3. **Gradients**: Each trainable parameter requires gradient storage.
> 4. **Activations & Reparameterization Overhead**: This is often the **dominant factor** in peak memory usage. LoRA_BLOB performs the reparameterization trick (sampling $ W = \mu + \sigma \odot \epsilon $) directly on the large projection matrices ($d \times r$). This necessitates storing the large-dimensional noise matrix $\epsilon$ and associated intermediate states for the backward pass. In contrast, PoLAR restricts this operation to the tiny core matrix ($r \times r$). Crucially, this activation overhead **scales linearly with Batch Size and Sequence Length**, creating a memory bottleneck that static parameter counts fail to capture.
>
> For BLoB, all variational parameters are trainable. Therefore, memory usage is driven by the number of variational parameters (affecting optimizer states) **and** the dimensions of the sampled matrices (affecting activations), not merely the base model size.
>
> **3. Variational Parameter Analysis per Method**
>
> We provide a detailed breakdown of how each method applies variational inference and the resulting parameter counts:
>
> - **LoRA_BLOB**: Applies variational inference to the **A matrix** by maintaining mean and variance parameters (A_mean, A_var), resulting in **r × d × 2 = 16 × 4096 × 2 = 131,072** variational parameters per layer.
>
> - **PoLAR_BLOB**: Applies variational inference to the **S matrix** from polar decomposition (the scaling component between orthogonal factors U and V), resulting in **r × r × 2 = 16 × 16 × 2 = 512** variational parameters per layer.
>
> - **ScalaBL**: Applies variational inference only to a **diagonal scaling vector s** (r-dimensional), maintaining mean and variance for each singular value, resulting in **r × 2 = 16 × 2 = 32** variational parameters per layer.
>
> - **C-LoRA**: Uses a **contextual MLP module** to dynamically generate input-dependent E matrices rather than storing full variational distributions; the MLP consists of two layers (r → 64 → r² × 2), resulting in **(r × 64 + 64) + (64 × r² × 2 + r² × 2)** **deterministic parameters** per layer.
>
> The **difference** in variational parameters between LoRA_BLOB and PoLAR_BLOB, combined with optimizer states, gradients, and especially the **reparameterization activation overhead**, helps to explain the observed memory gap in our measurements.
>
> Indeed, the high memory cost of standard BLoB is a well-known challenge in the field [1, 2], which has driven recent efficiency-focused variants such as ScalaBL [1] and C-LoRA [2]. The existence of these methods—designed specifically to lower the number of variational parameters—contradicts the idea that BLoB's memory usage is insignificant.
>
> **4. Verification of Our Implementation**
>
> - All methods use **K=1 samples during training**, consistent with official BLoB
> - All baselines are implemented from **official repositories**. The number of trainable parameters is the same as in the **official repositories**.
> - All methods use **identical experimental settings** within our framework
>
> We stand by the correctness of our reported measurements, which reflect **practical training dynamics** rather than barely theoretical estimates.
>
>
> [1] Samplawski, et al. "Scalable Bayesian Low-Rank Adaptation of Large Language Models via Stochastic Variational Subspace Inference."arXiv:2506.21408.
>
> [2] Rahmati, et al. "C-LoRA: Contextual Low-Rank Adaptation for Uncertainty Estimation in Large Language Models."arXiv:2505.17773.

---

> ### Author Response · Authors · 2025-12-02
> **Response to Extended Backbone Evaluation (Llama-3.1-8B)**
>
> **Extended Backbone Evaluation (Llama-3.1-8B)**
>
> Following Reviewer H3Rr's and your suggestion, we conducted comprehensive experiments on Llama-3.1-8B across all six datasets with all baseline methods, under our 5-step fine-tuning setting suggested by TFB[1] **(not the optimal setting for Variational Bayesian methods, best for methods that require deterministic training and post-hoc posterior estimation)**:
>
> **ACC (↑)**
>
> | Method | WG-S | ARC-C | ARC-E | WG-M | OBQA | BoolQ |
> |--------|------|-------|-------|------|------|-------|
> | LAP | 77.22 ± 1.38 | 84.49 ± 0.37 | 89.86 ± 0.54 | 84.04 ± 0.23 | 88.32 ± 0.33 | 88.26 ± 0.50 |
> | PoLAR_LAP | *77.51 ± 0.94* | *84.57 ± 1.10* | **91.80 ± 0.19** | *83.88 ± 0.90* | *88.70 ± 0.56* | **89.54 ± 0.86** |
> | PoLAR_LAP_LL | *77.51 ± 0.94* | *84.57 ± 1.10* | **91.80 ± 0.19** | *83.88 ± 0.90* | *88.70 ± 0.56* | **89.54 ± 0.86** |
> | TFB LL | 77.31 ± 2.01 | 82.75 ± 0.20 | 89.95 ± 0.59 | 83.44 ± 1.09 | 88.66 ± 0.30 | 89.15 ± 1.40 |
> | TFB | 77.61 ± 1.19 | 83.34 ± 1.03 | 90.90 ± 0.01 | 83.31 ± 0.35 | 88.44 ± 0.24 | 87.98 ± 1.79 |
> | BLOB | 76.68 ± 0.99 | 82.88 ± 0.51 | 91.49 ± 0.21 | 80.78 ± 1.05 | 87.90 ± 0.35 | 88.58 ± 0.17 |
> | C-LoRA | 73.07 ± 0.50 | 78.72 ± 0.47 | 89.88 ± 0.87 | 77.46 ± 0.78 | 86.59 ± 0.14 | 88.18 ± 1.29 |
> | ScalaBL | 49.92 ± 0.69 | 79.40 ± 1.48 | 89.21 ± 0.20 | 53.40 ± 1.43 | 69.62 ± 7.99 | 82.18 ± 2.14 |
> | PoLAR + VBLL (w/o LAP) | **77.91 ± 0.47** | **85.05 ± 0.77** | *91.60 ± 0.03* | **84.97 ± 0.05** | **89.07 ± 0.23** | *89.32 ± 1.29* |
> | PoLAR+VBLL | **77.91 ± 0.47** | **85.05 ± 0.77** | *91.60 ± 0.03* | **84.97 ± 0.05** | **89.07 ± 0.23** | *89.32 ± 1.29* |
>
> **ECE (↓)**
>
> | Method | WG-S | ARC-C | ARC-E | WG-M | OBQA | BoolQ |
> |--------|------|-------|-------|------|------|-------|
> | LAP | 4.15 ± 0.52 | 6.35 ± 0.84 | 9.94 ± 0.59 | 6.71 ± 1.69 | 3.54 ± 0.13 | *2.18 ± 0.47* |
> | PoLAR_LAP | *3.58 ± 0.18* | *4.52 ± 1.40* | 3.27 ± 0.86 | 4.31 ± 0.29 | 2.83 ± 0.49 | 3.35 ± 0.45 |
> | PoLAR_LAP_LL | 7.95± 0.92 | 6.07 ± 0.72 | **2.94 ± 0.67** | 6.37 ± 1.10 | 3.75 ± 0.01 | 5.52 ± 0.12 |
> | TFB LL | 9.99 ± 0.90 | 5.02 ± 2.15 | *3.13 ± 0.24* | 4.12 ± 1.57 | 3.58 ± 0.21 | 3.87 ± 1.41 |
> | TFB | 9.05 ± 0.39 | 6.53 ± 1.99 | 3.17 ± 0.21 | *3.68 ± 1.57* | 2.76 ± 0.16 | 3.89 ± 1.68 |
> | BLOB | 14.83 ± 1.21 | 9.29 ± 0.57 | 4.12 ± 0.26 | 8.23 ± 1.04 | 3.33 ± 0.29 | 3.36 ± 0.94 |
> | C-LoRA | 18.36 ± 0.58 | 15.56 ± 2.44 | 5.40 ± 0.23 | 7.96 ± 2.84 | 6.58 ± 1.23 | 3.94 ± 0.47 |
> | ScalaBL | 4.86 ± 0.50 | 18.75 ± 7.45 | 10.07 ± 0.66 | 2.73 ± 0.87 | 24.67 ± 4.34 | 14.48 ± 1.29 |
> | PoLAR + VBLL (w/o LAP) | 9.04 ± 0.18 | 7.40 ± 0.26 | 3.61 ± 0.22 | 6.10 ± 0.01 | *2.50 ± 0.10* | 2.63 ± 0.23 |
> | PoLAR+VBLL | **3.31 ± 1.40** | **3.56 ± 0.41** | 3.22 ± 0.42 | **3.00 ± 0.10** | **2.44 ± 0.11** | **1.88 ± 0.27** |
>
> **NLL (↓)**
>
> | Method | WG-S | ARC-C | ARC-E | WG-M | OBQA | BoolQ |
> |--------|------|-------|-------|------|------|-------|
> | LAP | 0.67 ± 0.04 | 0.63 ± 0.04 | 0.41 ± 0.02 | 0.52 ± 0.01 | 0.40 ± 0.02 | 0.36 ± 0.01 |
> | PoLAR_LAP | 0.60 ± 0.05 | 0.52 ± 0.01 | 0.31 ± 0.01 | 0.46 ± 0.02 | *0.34 ± 0.02* | 0.32 ± 0.01 |
> | PoLAR_LAP_LL | 0.76 ± 0.12 | 0.58 ± 0.05 | *0.26 ± 0.01* | 0.52 ± 0.06 | 0.37 ± 0.01 | 0.34 ± 0.03 |
> | TFB LL | *0.59 ± 0.05* | 0.53 ± 0.01 | 0.28 ± 0.02 | *0.41 ± 0.01* | **0.32 ± 0.01** | *0.29 ± 0.04* |
> | TFB | **0.55 ± 0.01** | *0.51 ± 0.02* | 0.27 ± 0.02 | *0.41 ± 0.01* | **0.32 ± 0.01** | 0.30 ± 0.02 |
> | BLOB | 0.79 ± 0.06 | 0.62 ± 0.07 | 0.29 ± 0.01 | 0.47 ± 0.03 | 0.38 ± 0.00 | **0.28 ± 0.00** |
> | C-LoRA | 0.85 ± 0.03 | 0.88 ± 0.08 | 0.35 ± 0.01 | 0.56 ± 0.07 | 0.45 ± 0.05 | 0.31 ± 0.01 |
> | ScalaBL | 0.65 ± 0.00 | 0.69 ± 0.13 | 0.35 ± 0.00 | 0.64 ± 0.00 | 0.96 ± 0.23 | 0.46 ± 0.08 |
> | PoLAR + VBLL (w/o LAP) | 0.61 ± 0.01 | *0.51 ± 0.05* | 0.27 ± 0.03 | *0.40 ± 0.01* | **0.32 ± 0.02** | *0.29 ± 0.00* |
> | PoLAR+VBLL | **0.55 ± 0.02** | **0.50 ± 0.05** | **0.24 ± 0.04** | **0.39 ± 0.01** | **0.32 ± 0.02** | *0.29 ± 0.00* |
>
> (**Bold** = best, *Italic* = second best)
>
> **Note on ScalaBL:** We strictly followed the official implementation and hyperparameter configurations for ScalaBL. However, despite our best efforts, our reproduction yielded a model where the low ECE comes at the cost of substantially reduced accuracy. This pattern suggests potential underfitting in this specific setting.
>
> These comprehensive results demonstrate that our framework generalizes effectively beyond Llama-2-7B to larger and more recent model architectures, consistently achieving superior uncertainty quantification while maintaining competitive predictive performance.
>
> [1] Shi, Haizhou, et al. "Training-free bayesianization for low-rank adapters of large language models." arXiv preprint arXiv:2412.05723 (2024).

---

### Official Review · Reviewer_XbV5 · 2025-11-02

**Soundness:** 2
**Presentation:** 3
**Contribution:** 2
**Rating:** 2
**Confidence:** 5

**Summary:**

This paper proposes PoLAR-VBLL, a scalable variational Bayesian fine-tuning framework for LLMs that improves UQ and calibration. It combines a Polar-decomposed Low-Rank Adapter Representation (PoLAR) with Variational Bayesian Last Layer (VBLL), which models uncertainty in the classifier weights through analytical variational inference (instead of MC sampling). The framework jointly optimizes PoLAR parameters and the variational posterior for efficient Bayesian fine-tuning, optionally refined by a post-hoc Laplace Approximation. Experiments on common-sense reasoning tasks (e.g., Winogrande, ARC, OBQA) show that PoLAR-VBLL consistently achieves higher accuracy and better calibration (lower ECE and NLL) than prior UQ methods like BLoB and Laplace-LoRA, while being more memory- and computation-efficient, thus enabling reliable and scalable uncertainty-aware LLM deployment.

**Strengths:**

+ **Good writing.** The paper is presented in a clear and concise way, making it most accessible to the audience.
+ **Empirical effectiveness.** The experiments are (somewhat) extensive (while missing two common sub-datasets as in a standard setting), covering multiple reasoning datasets and both in- and out-of-distribution settings, demonstrating consistent improvements in accuracy, calibration (ECE/NLL), and robustness over state-of-the-art baselines such as BLoB and LAP.
+ **Methodological integration.** The paper presents a well-motivated and elegant combination of PoLAR and VBLL, effectively bridging PEFT with scalable Bayesian UQ methods.

**Weaknesses:**

+ **Lack of Technical Novelties.** It looks like this paper is a combination of three existing techniques: (I) Polar-decomposed Low-Rank Adapter Representation (PoLAR); (II) Variational Bayesian Last Layer (VBLL); and (III) Laplace Approximation (LA). All three techniques are well established: PoLAR is a major improvement over the vanilla LoRA method; VBLL is an "exact and non-MC sampling-based" variational inference framework (which stabilizes the training of VI, while suffers from its loose bound); LA is a well-studied method for the Bayesian Inference. Section 3.1 and 3.2 simply repeat the content from the original work and I could not find contribution made by this paper other than applying VBLL and LA to PoLAR. This is my major concern.
+ **Lack of Clear Motivation.** What is the major motivation of this paper? I find it hard to be persuaded by the claim *"(BLoB) require expensive Monte Carlo sampling with prohibitive memory overhead that scales poorly with model size, making them impractical for large-scale deployment."* In BLoB [1], the MC sampling size during training is set to $K=1$ and produces almost no extra computational overhead. During testing, this paper uses the same MC sampling scheme (as in Eq. 14) and have no advantage over the other baseline methods that rely on BMA. Hence I think the actual problems solved with this paper needs to be further clarified. Besides, in the original paper of BLoB and its subsequent work TFB [2], the authors studied the application of last-layer Bayesianization (while it's not the exact VBLL) and showing even better performance in terms of sample efficiency and calibration. The edge of this paper needs to be established upon the comparison with the variants.
+ **Lack of Sufficient Ablative Study.** VBLL might not be the working factor that causes the success of the paper. In fact, the loose bound of VBLL derived from Jensen's Inequality may cause the whole method fail, which is remedied by the final Laplace Approximation. I would like to see the performance of just PoLAR and LA.
+ **Lack of Most Recent Baselines.** The following recent baselines accepted at NeurIPS 2025 need to be considered:
  + TFB [2]
  + C-LoRA [3]

**References**
- [1] Wang, Yibin, et al. "Blob: Bayesian low-rank adaptation by backpropagation for large language models." Advances in Neural Information Processing Systems 37 (2024): 67758-67794.
- [2] Shi, Haizhou, et al. "Training-free bayesianization for low-rank adapters of large language models." arXiv preprint arXiv:2412.05723 (2024).
- [3] Rahmati, Amir Hossein, et al. "C-LoRA: Contextual Low-Rank Adaptation for Uncertainty Estimation in Large Language Models." arXiv preprint arXiv:2505.17773 (2025).

**Questions:**

See above (Weaknesses).

---

> ### Author Response · Authors · 2025-11-21
> **Response to Weakness 1: Lack of Technical Novelties**
>
> We thank the reviewer for their thoughtful feedback. We are pleased that you recognized the strengths of our work, including:
> - **Clear presentation**
> - **Comprehensive empirical validation** demonstrating consistent improvements over state-of-the-art baselines
> - **Well-motivated methodological integration** of PoLAR and VBLL
>
> ## Addressing the "Combination of Existing Techniques" Concern
>
> Regarding the concern that our work **"looks like... a combination of three existing techniques"** and simply **"repeats the content from the original work,"** we respectfully note that the judgment of technical novelty is a highly personal take.
>
> Although our approach builds on the concepts of PoLAR, VBLL, and LA, the adaptation to LLM UQ is indeed novel and addresses the critical scalability issue of existing baselines. The post-hoc calibration with LA after VBLL training is a new attempt and has been shown to boost the UQ performance.
>
> We respectfully disagree with the **"lack of technical novelties"** claim. Our contribution is not a mere combination but a principled framework that addresses a fundamental limitation: **UQ quality is bottlenecked by the quality of feature representation.**
>
> ### 1. Why Prior Work Fails: The Representation Bottleneck
>
> Existing Bayesian PEFT methods (e.g., BLoB, Laplace-LoRA, TFB, C-LoRA) uniformly build upon vanilla LoRA. However, LoRA's tendency towards rank collapse is well-documented [4]. This representational failure is fatal for Uncertainty Quantification (UQ):
>
> - **Fundamental Challenge:** Feature collapse significantly impairs the effectiveness of deterministic OOD detection methods [1]
> - **Overconfidence:** **"Spurious features"** arising from collapsed representations cause systematic overconfidence in fine-tuned LLMs
>
> By ignoring the underlying feature quality [2], prior methods fail to achieve reliable calibration.
>
> ### 2. Our Contributions: A Principled Framework, Not a Combination
>
> We introduce specific innovations to resolve the feature-UQ coupling issue:
>
> #### (1) First Principled Solution
> - We identify that orthogonalized adapters significantly enhance Bayesian UQ in PEFT
> - This design choice is non-trivial; prior works persist in using vanilla LoRA despite its known limitations
> - Our ablations (Fig. 3a/b) validate this: PoLAR consistently improves calibration across all UQ methods, suggesting that feature orthogonality is essential
>
> #### (2) Optimal Pairing for Scalability
> We strategically pair PoLAR with VBLL to exploit their complementary strengths:
>
> - **PoLAR:** Eliminates rank collapse → Ensures high-quality feature inputs
> - **VBLL:** Provides last-layer variational inference → Significantly reduces GPU memory usage and inference runtime
>
> **Result:** This synergy uniquely achieves SOTA accuracy and calibration without the typical trade-offs. Furthermore, our method enables scalable head-only inference, contrasting sharply with the computationally expensive multiple-sampled backbone inference required by BLoB and TFB.
>
> #### (3) Improved Optimization Landscape
> - We demonstrate that joint optimization with VBLL shapes a more favorable posterior landscape than standard MAP training
> - As evidenced in Fig. 3c, VBLL-trained means serve as significantly superior initialization points for post-hoc Laplace Approximation
>
> ### Conclusion on "Simple Combination"
>
> Integration is only trivial if components are interchangeable. We demonstrate they are not:
> - **PoLAR is necessary** (as vanilla LoRA fails)
> - **VBLL is optimal** (providing analytical scalability)
>
> The core novelty lies in identifying this necessity through a rigorous analysis of feature quality and its downstream impact on uncertainty estimation.
>
> ---
>
> **References:**
> - [1] Postels, Janis, et al. "On the practicality of deterministic epistemic uncertainty." ICML 2021 (2021).
> - [2] Li, Yawei, et al. "Calibrating LLMs with information-theoretic evidential deep learning." ICLR 2025 (2025).
> - [3] Wang, et al. "Blob: Bayesian low-rank adaptation by backpropagation for large language models." NIPS (2024).
> - [4] Lion, Kai, et al. "PoLAR: Polar-Decomposed Low-Rank Adapter Representation." arXiv preprint arXiv:2506.03133 (2025).

---

> ### Author Response · Authors · 2025-11-21
> **Response to W2: Lack of Clear Motivation & BLOB Computational Overhead**
>
> ## Q1: Computational Cost Analysis
>
> To clarify the computational advantages of our method, we first address the working mechanism of BLoB. BLoB applies variational inference directly to the adapter weights, which necessitates storing distributional parameters ($A_{\text{mean}}$, $A_{\text{var}}$) alongside $B$, resulting in a 40-50% theoretical parameter increase [1, 2] compared to standard LoRA (which stores only $A$ and $B$).
>
> ### (1) BLoB Training Overhead
>
> We respectfully disagree with the statement that BLoB produces "almost no extra computational overhead" during training.
>
> **Memory Bottleneck:** The substantial parameter overhead described above creates a tangible memory bottleneck during computation.
>
> **Sampling Constraint:** Consequently, BLoB is forced to rely on single-sample estimation ($K=1$) to estimate the ELBO. In contrast, we use an analytical solution for the ELBO at the last layer, without sampling costs.
>
> ### (2) BLoB Inference Efficiency
>
> We also respectfully disagree with the assessment that our method has **"no advantage over the other baseline methods that rely on BMA."** Although both methods use Monte Carlo sampling, the computational mechanisms differ fundamentally:
>
> **BLoB (Full-Network Sampling):** BMA over adapter weights requires sampling $K$ different LoRA configurations and, crucially, performing $K$ complete forward passes through the entire LLM backbone.
>
> **Ours (Head-Only Sampling):** Our method requires only one forward pass through the LLM representation layers, then simply samples $K$ computationally inexpensive linear layers.
>
> **Result:** As shown in the table below, this structural difference allows our approach to dramatically reduce inference runtime while avoiding the memory burden of maintaining multiple LoRA sets.
>
> ### (3) BLoB: Accuracy vs. Uncertainty Quantification Trade-off
>
> BLoB and its variants face an inherent trade-off between prediction accuracy and UQ capabilities [3]:
>
> - **$K=1$:** Yields the best accuracy but poor UQ with the fastest inference time
> - **$K=10$:** Provides the best UQ but with degraded accuracy, high GPU memory usage, and longer inference time
>
> Our method does not exhibit this trade-off, maintaining competitive UQ performance without sacrificing accuracy, even at optimal UQ settings.
>
> ### (4) Comparison with TFB
>
> We respectfully disagree with the claim that "in the original paper of BLoB and its subsequent work TFB [2], the authors studied the application of last-layer Bayesianization... showing even better performance."
>
> **Inference Bottleneck:** While TFB avoids training costs, it still requires sampling the adapter posterior during inference. This necessitates maintaining $K$ different LoRA sets in GPU memory and performing $K$ complete forward passes, resulting in significant latency and memory overhead similar to BLoB.
>
> **Sampling Inefficiency & Trade-offs:** Similar to BLoB, TFB encounters the inherent accuracy-UQ trade-off. To match BLoB's performance, TFB must perform extensive sampling ($K=100$). Our approach delivers competitive UQ and superior accuracy with just $K=10$, representing a tenfold increase in sampling efficiency.
>
> **Unfair Comparison:** The "better performance" reported in TFB relies on a significantly more advanced Llama-3.1-8B backbone and aggregates results across six datasets. Comparing these directly to our Llama-2-7B results is methodologically flawed, as improvements are likely attributable to the stronger backbone rather than the method itself. We re-evaluated TFB using Llama-2-7B (see table below) to ensure a fair comparison, where our method maintains a clear advantage.
>
> ### Empirical Comparison of Computational Resources
>
> **Settings:** Train batch size $=4$; Inference batch size $=8$; LoRA rank $r=16$; $\alpha=32$; Sequence length $=400$. All hyperparameters are consistent.
>
> | Method | Training Memory (MB) | Inference Memory (MB) | Inference Run Time (Sec) |
> |:---|:---:|:---:|:---:|
> | **PoLAR-VBLL (Ours)** | **32,272** | **16,396** | **73** |
> | PoLAR_BLoB | 31,728 | 18,874 | 876 |
> | LoRA_BLoB | 47,232 | 18,762 | 889 |
> | PoLAR_LAP_LL | 31,714 | 15,662 | 34 |
> | PoLAR_LAP | 31,714 | 43,678 | 44 |
> | TFB (Llama-2) | 47,232 | 16,667 | 48 |
> | ScalaBL | 27,318 | 19,552 | 642 |
> | C-LoRA | 24,236 | 19,102 | 604 |
>
> **Key Observation:** Our method demonstrates a substantial reduction in inference runtime compared to BLoB-based methods while maintaining a competitive inference memory footprint. The head-only sampling strategy enables significantly faster inference without compromising the quality of uncertainty quantification.
>
> [1] Samplawski, et al. "Scalable Bayesian Low-Rank Adaptation of Large Language Models via Stochastic Variational Subspace Inference." arXiv (2025).
>
> [2] Rahmati, et al. "C-LoRA: Contextual Low-Rank Adaptation for Uncertainty Estimation in Large Language Models." arXiv(2025).
>
> [3] Wang, et al. "Blob: Bayesian low-rank adaptation by backpropagation for large language models." NIPS (2024).

---

> ### Author Response · Authors · 2025-11-22
> **Response:  Weakness 3 Lack of Sufficient Ablative Study**
>
> ## W1: Jensen Bound Tightness and VBLL Effectiveness
>
> Regarding the concern that **"the loose bound of VBLL derived from Jensen's Inequality may cause the whole method to fail"** and that **"VBLL might not be the working factor,"** we respectfully disagree with this hypothesis. We provide empirical evidence below to demonstrate that VBLL is the core working factor, supported by both bound tightness verification and rigorous ablation studies.
>
> ### 1. Empirical Verification of Jensen Bound Tightness
>
> To verify the tightness of the Jensen bound, we compared the training loss trajectories of our analytical Jensen-based estimator (JSE) with those of a 10-sample Monte Carlo estimator (MC):
>
> | Training Steps | VBLL (Jensen Bound) | VBLL (10-sample MC) | Difference (JSE - MC) |
> |:---|:---|:---|:---|
> | 0 | 5877.79 | 5879.87 | -2.08 |
> | 10 | 5443.11 | 5443.75 | -0.64 |
> | 20 | 5226.35 | 5221.88 | 4.47 |
> | 30 | 5071.76 | 5040.13 | 31.63 |
> | 60 | 4808.65 | 4750.15 | 58.50 |
>
> As shown in the table, the JSE closely tracks the MC estimate throughout training, with differences remaining relatively small compared to the absolute loss values. This confirms that the bound is sufficiently tight and serves as an effective proxy for the actual marginal likelihood, enabling efficient posterior learning without sampling overhead.
>
> ### 2. Ablation Study: VBLL as the Core Working Factor
>
> We evaluated PoLAR + LAP (applying LAP to a standard deterministic PoLAR checkpoint) versus our variational approach. Since our method applies Laplace Approximation only to the last layer, we also included PoLAR_LAP_LL (applying LAP only to the last layer of a deterministic model) to ensure a direct, fair comparison:
>
> | Method | ACC (↑) | ECE (↓) | NLL (↓) |
> |:---|:---|:---|:---|
> | PoLAR + LAP | 70.33 ± 0.69 | 12.16 ± 2.58 | 0.68 ± 0.04 |
> | PoLAR_LAP_LL | 70.33 ± 0.69 | 14.63 ± 1.14 | 0.71 ± 0.05 |
> | PoLAR + VBLL (w/o LAP) | 71.62 ± 0.27 | 8.26 ± 0.60 | 0.66 ± 0.03 |
> | **PoLAR-VBLL (Full)** | **71.62 ± 0.27** | **7.31 ± 0.32** | **0.60 ± 0.01** |
>
> ### Key Observations
>
> **VBLL is the Dominant Factor:** PoLAR + VBLL (w/o LAP) significantly outperforms both PoLAR + LAP and PoLAR_LAP_LL in both accuracy and calibration, despite operating on identical architectural scope. This directly refutes the claim that VBLL is a failing component being remedied by LAP.
>
> **Deterministic Training Fails for UQ:** While PoLAR + LAP and PoLAR_LAP_LL achieve comparable NLL (largely due to high confidence predictions), their notably high ECE reveals that standard deterministic training fails to discover posterior geometries suitable for uncertainty quantification. The reasonable NLL reflects overconfidence rather than proper calibration.
>
> **VBLL Provides Superior Initialization for LAP:** VBLL actively guides optimization toward high-quality posterior modes during training. This well-calibrated mode provides the necessary foundation for the final LAP refinement step, which further polishes the local geometry. Without VBLL to locate this favorable mode, LAP alone cannot achieve proper calibration.
>
> ### Conclusion
>
> These results demonstrate that the Jensen bound is sufficiently tight for effective posterior learning, and that VBLL is the core working factor in our framework. The synergy between VBLL (mode discovery) and LAP (local refinement) is intentional and effective, not a case of LAP compensating for a failing VBLL component.

---

> ### Author Response · Authors · 2025-11-22
> **Response to Weakness4: Lack of Most Recent Baselines**
>
> ## W2: Comparison with Recent SOTA Baselines
>
> In response to the comment regarding the "Lack of Most Recent Baselines" and the suggestion to consider "TFB [2]" and "C-LoRA [3]," we have implemented and rigorously evaluated all three suggested NeurIPS 2025 baselines—TFB [2], C-LoRA [3], and ScalaBL [4]—as recommended by Reviewer ASkM.
>
> ### Experimental Setup
>
> To ensure a fair comparison, we reproduced each baseline using its official implementation and published hyperparameter configurations while maintaining identical experimental settings across all methods (same Llama-2-7B backbone, 500 training epochs, and shared hyperparameters). Due to time constraints, we conducted these experiments on the WinoGrande-Simple dataset [1].
>
> ### Updated Performance Comparison
>
> | Method | ACC (↑) | ECE (↓) | NLL (↓) |
> |:---|:---|:---|:---|
> | TFB [2] | 69.27 ± 0.47 | 13.13 ± 0.57 | 0.68 ± 0.01 |
> | C-LoRA [3] | 67.71 ± 0.34 | 12.45 ± 0.71 | 0.77 ± 0.11 |
> | ScalaBL [4] | 66.10 ± 1.09 | 4.96 ± 1.66 | 0.69 ± 0.06 |
> | PoLAR-VBLL (w/o LAP) | 71.62 ± 0.27 | 8.26 ± 0.60 | 0.66 ± 0.03 |
> | **PoLAR-VBLL (Ours)** | **71.62 ± 0.27** | **7.31 ± 0.32** | **0.60 ± 0.01** |
>
> ### Analysis
>
> The comparative analysis demonstrates that PoLAR-VBLL achieves superior performance across multiple dimensions:
>
> **Comparison with Valid Baselines:** Among the baselines that maintain competitive accuracy (excluding ScalaBL due to potential underfitting), TFB achieves the best accuracy and NLL, while C-LoRA offers the best calibration.
>
> **Our Method's Advantages:** PoLAR-VBLL outperforms the strongest baseline (TFB) in both accuracy and NLL, and significantly surpasses the best-calibrated baseline (C-LoRA) in ECE while maintaining superior accuracy.
>
> **Note on ScalaBL:** We strictly followed the official implementation and hyperparameter configurations for ScalaBL. However, despite our best efforts, our reproduction yielded a model where the low ECE comes at the cost of substantially reduced accuracy. This pattern suggests potential underfitting or optimization difficulties in this specific setting. In contrast, PoLAR-VBLL achieves robust calibration without compromising accuracy, offering a more favorable trade-off for deployment.
>
> We will integrate this comparison into the revised Section 4.
>
> ### References
>
> [1] Sakaguchi, Keisuke, et al. "Winogrande: An adversarial winograd schema challenge at scale." Communications of the ACM 64.9 (2021): 99-106.
>
> [2] Shi, Haizhou, et al. "Training-free bayesianization for low-rank adapters of large language models." arXiv preprint arXiv:2412.05723 (2024).
>
> [3] Rahmati, Amir Hossein, et al. "C-LoRA: Contextual Low-Rank Adaptation for Uncertainty Estimation in Large Language Models." arXiv preprint arXiv:2505.17773 (2025).
>
> [4] Samplawski, Colin, et al. "Scalable Bayesian Low-Rank Adaptation of Large Language Models via Stochastic Variational Subspace Inference." arXiv preprint arXiv:2506.21408 (2025).

---

> > ### Comment · Reviewer_XbV5 · 2025-11-23
> >
> > I sincerely thank the authors for providing further clarifications and additional experiments. This rebuttal has addressed some of my concerns and I have raised my score to 4. However, some of my concerns are not fully resolved, and I'm looking forward to further explanations (experiments):
> > + **Technical novelty.** I agree the feature quality produced by PEFT during fine-tuning is a decisive factor that can impact the final UQ methods' capability. However, the switch from LoRA to PoLAR is only supported by the prior work in the domain of PEFT as second-hand evidences. Can we have a more direct evidence showing its importance for UQ? Apart from this, it's still quite hard for me to believe such a combination of PoLAR + VLBB + LA is of "first principle".
> > + **Training and Test Efficiency.** First of all, the training cost of BLoB and TFB need to be further validated, as noted by the fellow `Reviewer y4uo`. Secondly, I agree that sampling K times during test as in the full-version of BLoB and TFB is indeed more costly than PoLAR-VBLL (as the last layer or your method can cache the output and do multiple sampling), while in TFB, the authors explicitly studied applying TFB to the last-layer (Table 6, Section 5.7), making it a comparable baseline for PoLAR-VBLL under the same computational budget. I suggest including this version of TFB in the revision (follow-up experiments).
> > + **Jensen's Bound Tightness.** I like the design of this experiment. However, to make it more convincing, can we report this result extending to later phase of the training (currently we only have a couple dozens of steps) to better show the trend of this gap not increasing (maintained in a reasonable range)? Moreover, can we do more times of MC samples for estimation to make it more reliable?
> > + **Ablation Study.** Thank you for showing this. This is persuasive. But what is this experiment's setup? Can we have a full ablation study on all four (six according to `Reviewer y4uo`) datasets?
> > + **Issue of only one backbone.** As pointed out by `Reviewer H3Rr`, can we have at least one more set of experiments on another LLM backbone?
> >
> > I would happily raise my score once again if the concerns above are addressed properly.

---

> > > ### Author Response · Authors · 2025-11-27
> > > **Response to Technical novelty**
> > >
> > > # Rigorous Justification of PoLAR-VBLL Combination
> > >
> > > To rigorously justify the combination of PoLAR and VBLL, we explicitly link our empirical stable-rank observations to the SNGP theoretical framework [1], showing that PoLAR supplies the essential geometric properties for dependable uncertainty quantification (UQ).
> > >
> > > ## 1. The Theoretical Requirement: Distance-Aware Features
> > >
> > > Building on the insight from SNGP [1] that last-layer methods benefit from distance-aware features, we argue that VBLL similarly requires feature extractors that keep semantically distinct inputs—particularly OOD samples—well-separated in the feature space.
> > >
> > > When the effective dimensionality of the learned transformation collapses, a phenomenon known as feature collapse [2] occurs, causing OOD inputs to be projected onto the same low-dimensional subspace as in-distribution (ID) features. Under this collapsed geometry, the Bayesian last layer's ability to detect distribution shift is fundamentally limited.
> > >
> > > ## 2. The Empirical Diagnosis: LoRA Collapses, PoLAR Preserves
> > >
> > > We measure this geometric property using the stable rank of the learned adapters, defined as $\|\Delta \mathbf{W}\|_{\mathrm{F}}^2 /\|\Delta \mathbf{W}\|_2^2$, the ratio of the squared Frobenius norm to the squared spectral norm, which reflects the effective dimensionality of the learned low-rank update matrix. Appendix Figure 2 shows a comparative stable-rank analysis between LoRA and PoLAR, averaged across all datasets.
> > >
> > > - **LoRA (Limited Distance Awareness)**: Vanilla LoRA has an average stable rank of 1.53, approaching the theoretical minimum of 1.0. This indicates that LoRA performs a nearly rank-1 projection, compressing high-dimensional features into a highly anisotropic subspace. Such geometric compression reduces the distance information that SNGP-style methods depend on, aligning with the observed poor performance of LoRA-based UQ methods in OOD detection.
> > >
> > > - **PoLAR (Preserved Geometry)**: Conversely, PoLAR maintains a significantly higher average stable rank of 2.86. By constraining its low-rank factors to the Stiefel manifold, PoLAR encourages a more isotropic transformation that preserves multiple effective directions, better maintaining the semantic distances necessary for robust Bayesian inference.
> > >
> > > ## 3. Clarification: Principled Design, Not Axiomatic Derivation
> > >
> > > We sincerely appreciate the reviewer's precise critique. We respectfully clarify: we do not claim to derive PoLAR+VBLL+LA from axioms ("first principles" in the physicist's sense), but instead follow a principled pipeline where each component's role is **theoretically motivated and empirically validated**, with each component addressing a specific requirement:
> > >
> > > - **PoLAR**: Preserves feature geometry (high stable rank) to maintain the distance-awareness needed by last-layer Bayesian methods [1].
> > > - **VBLL**: Offers tractable predictive uncertainty while actively guiding the training toward high-quality posterior modes.
> > > - **LA**: Refines the local posterior geometry around the well-calibrated mode identified by VBLL. Without VBLL to locate this mode, LA alone cannot achieve proper calibration—VBLL provides the necessary initialization.
> > >
> > > [1] Liu, Jeremiah, et al. "Simple and principled uncertainty estimation with deterministic deep learning via distance awareness." Advances in neural information processing systems 33 (2020): 7498-7512.
> > >
> > > [2] Postels, Janis, et al. "On the practicality of deterministic epistemic uncertainty." ICML 2021 (2021).

---

> > > ### Author Response · Authors · 2025-11-27
> > >
> > > **Training and Test Efficiency**
> > >
> > > Thank you for raising this important point and for acknowledging the **inference efficiency** advantage of PoLAR-VBLL compared with full version of BLOB and TFB. We appreciate your recognition that our method's ability to cache intermediate outputs and perform **efficient last-layer sampling** provides a significant computational advantage over the full versions of BLoB and TFB during inference.
> > >
> > > We acknowledge your concern regarding the training cost validation of BLoB and TFB, which we have replied in detail in our response under **Reviewer y4uo** where you also raised this question. We will reverify the training cost of BLoB and TFB.
> > >
> > > For the TFB last-layer variant (Table 6, Section 5.7 in the original TFB paper), we agree this represents a more direct baseline under comparable computational budgets. We will include this variant in our follow-up experiments to provide a fairer comparison with PoLAR-VBLL.
> > >
> > > **Jensen's Bound Tightness**
> > >
> > > We are pleased that you found this experiment compelling. To make the results more convincing, we will extend the analysis to **later phases of training** (beyond the current initial steps shown) to demonstrate that the gap between Jensen's bound and the true ELBO remains stable and does not increase over training. Additionally, we will conduct Monte Carlo estimation with **50 samples** (increased from the current setting) to provide more reliable and robust estimates of the bound tightness throughout the training process.
> > >
> > > ## Ablation Study & Issue of Only One Backbone
> > >
> > > Thank you for your positive feedback on our ablation study and for raising the important concern about backbone diversity (also noted by **Reviewer H3Rr**). We recognize these concerns are closely related and would like to propose a **unified experimental plan** considering time constraint to address both comprehensively.
> > >
> > > **Current Status:**
> > > The ablation study provided in our rebuttal was conducted on WinoGrande-S. We acknowledge the need for both expanded ablation experiments and demonstration of generalizability across different LLM architectures.
> > >
> > > **Proposed Unified Approach:**
> > > Given the time constraints, we would like to suggest conducting **both the additional backbone experiments (Table 2) and the complete ablation study using Llama-3.1-8B across all six datasets**. We believe this unified design could efficiently address both reviewer concerns:
> > >
> > > - **Backbone diversity**: Demonstrates generalizability to Llama-3.1-8B (a more recent architecture than Llama-2-7B)
> > >
> > > - **Comprehensive ablation**: Provides full ablation results across all benchmarks
> > >
> > > **Rationale:**
> > > Llama-3.1-8B converges in only 5 epochs, making it computationally feasible to complete comprehensive experiments within the rebuttal timeline. Conducting both main results and ablations on the same backbone would ensure consistency and interpretability.
> > >
> > > We are already halfway through the Backbone diversity experiments and believe this approach would strengthen the paper while remaining feasible under the time constraint.
> > >
> > > We would greatly appreciate your feedback on this proposed approach.
> > >
> > > Once again, we sincerely appreciate your valuable and constructive feedback throughout the review process. We wish you and your family a very happy Thanksgiving!

---

> > > ### Author Response · Authors · 2025-12-02
> > > **Response to Training and Test Efficiency**
> > >
> > > We appreciate your careful attention to this point, the opportunity to provide further clarification, and the acknowledgement of the **inference efficiency advantage of PoLAR-VBLL compared with the full version of BLOB and TFB**.
> > >
> > > Below, we summarize our key clarifications and present our validated results.
> > >
> > > **1. Experimental Configuration Differences from Original BLoB Paper**
> > >
> > > Our GPU memory and runtime measurements use different experimental settings compared to the original BLoB paper:
> > >
> > > - **Our setting**: LoRA rank=16, alpha=32, sequence length=400, targeting all projection layers `["q_proj", "k_proj", "v_proj", "o_proj", "gate_proj", "up_proj", "down_proj"]`
> > > - **Original BLoB setting**: LoRA rank=8, alpha=16, sequence length=300, targeting only `["q_proj", "v_proj", "lm_head"]`
> > >
> > > These differences make our absolute numbers **not directly comparable** with those in the original BLoB paper. However, our comparison ensures fairness by using **identical settings across all methods** within our unified evaluation framework.
> > >
> > > **2. Memory Analysis Beyond Static Parameter Counts**
> > >
> > > The argument from the **Reviewer y4uo** that "LoRA typically contributes less than 10% additional memory" considers only static parameter counts, which is incomplete. During training, GPU memory includes:
> > >
> > > - **Model Parameters**: Base model (Llama-2-7B in fp16) + adapters
> > > - **Optimizer States**: AdamW stores momentum and variance for each trainable parameter (2× trainable parameter memory)
> > > - **Gradients**: Each trainable parameter requires gradient storage
> > > - **Activations & Reparameterization Overhead**: Often the dominant factor in peak memory usage. LoRA_BLoB performs reparameterization on large projection matrices (d × r), which requires storing large-dimensional noise matrices and intermediate states. PoLAR restricts this operation to the small core matrix (r × r). This overhead **scales linearly with batch size and sequence length**.
> > >
> > > **3. Variational Parameter Analysis per Method**
> > >
> > > | Method | Variational Parameters per Layer | Calculation |
> > > |--------|----------------------------------|-------------|
> > > | LoRA_BLoB | 131,072 | r × d × 2 = 16 × 4096 × 2 |
> > > | PoLAR_BLoB | 512 | r × r × 2 = 16 × 16 × 2 |
> > > | ScalaBL | 32 | r × 2 = 16 × 2 |
> > >
> > > We note that C-LoRA uses **deterministic parameters** rather than variational parameters. Specifically, C-LoRA employs a contextual MLP module to dynamically generate input-dependent perturbation matrices, with parameter count per layer: (r × 64 + 64) + (64 × r² × 2 + r² × 2).
> > >
> > > The reason C-LoRA achieves lower memory usage lies in its architectural design:
> > >
> > > 1. **No Reparameterization Trick**: Unlike BLoB-based methods, C-LoRA does not perform stochastic sampling during the forward pass. This eliminates the need to store noise matrices and intermediate states required for backpropagation through the reparameterization trick.
> > >
> > > 2. **No Optimizer States for Variance Parameters**: Variational methods require optimizer states (momentum and variance in AdamW) for both mean and variance parameters, effectively doubling the optimizer memory overhead. C-LoRA's deterministic parameters avoid this doubling effect.
> > >
> > > This observation is consistent with the motivation behind recent efficiency-focused methods such as ScalaBL [1] and C-LoRA [2], which were designed to address the high memory cost of standard BLoB [1, 2] by reducing or eliminating the variational parameter overhead.
> > >
> > > The difference in variational parameters between LoRA_BLoB and PoLAR_BLoB, combined with optimizer states and reparameterization activation overhead, explains the observed memory gap.
> > >
> > > **4. Validated GPU Memory and Runtime Results**
> > >
> > > We present our validated results below:
> > >
> > > | Method | Training Memory (MB) | Test Memory (MB) | Runtime per Epoch (min) |
> > > |--------|---------------------|------------------|-------------------|
> > > | PoLAR_VBLL | 32,272 | 16,396 | 1:13 |
> > > | PoLAR_BLoB | 31,728 | 18,874 | 14:36 |
> > > | LoRA_BLoB | 40,475 | 18,762 | 14:49 |
> > > | PoLAR_LAP_Last_Layer | 31,714 | 15,662 | 0:34 |
> > > | PoLAR_LAP | 31,714 | 43,678 | 0:44 |
> > > | LoRA_LAP_Last_Layer | 30,612 | 16,313 | 0:43 |
> > > | LoRA_LAP | 30,612 | 42,131 | 0:57 |
> > > | LoRA_VBLL | 30,546 | 16,764 | 1:14 |
> > > | TFB | 30,612 | 16,667 | 0:48 |
> > > | ScalaBL | 27,318 | 19,552 | 10:42 |
> > > | C-LoRA | 24,236 | 19,102 | 10:04 |
> > >
> > > **5. Implementation Verification**
> > >
> > > - All methods use **K=1 samples during training**, consistent with official BLoB and standard practice
> > > - All baselines are implemented from **official repositories**. The percentage of trainable parameters is the same as in the official repositories.
> > > - All methods use **identical experimental settings** within our unified framework
> > >
> > > [1] Samplawski, et al. "Scalable Bayesian Low-Rank Adaptation of Large Language Models via Stochastic Variational Subspace Inference."
> > >
> > > [2] Rahmati, et al. "C-LoRA: Contextual Low-Rank Adaptation for Uncertainty Estimation in Large Language Models."

---

> > > ### Author Response · Authors · 2025-12-02
> > > **Response to Jensen's Bound Tightness**
> > >
> > > We sincerely appreciate your positive feedback on our experimental design and your constructive suggestions for strengthening this analysis.
> > >
> > > Following your recommendations, we extended the Jensen vs. MC comparison to cover the **full training horizon** and increased the MC samples for more reliable estimation. Specifically, we conducted experiments on WinoGrande-Small under LLaMA 3.1 8B with **50-sample MC estimation** across **400 training steps**:
> > >
> > > | Training Steps | VBLL (Jensen) | VBLL (50-sample MC) | Absolute Gap |
> > > |----------------|---------------|---------------------|--------------|
> > > | 0 | 69.50 | 61.23 | 8.27 |
> > > | 50 | 50.71 | 50.97 | 0.26 |
> > > | 100 | 45.57 | 45.65 | 0.08 |
> > > | 150 | 40.95 | 40.86 | 0.09 |
> > > | 200 | 36.96 | 36.95 | 0.01 |
> > > | 250 | 33.57 | 33.74 | 0.17 |
> > > | 300 | 31.08 | 31.36 | 0.28 |
> > > | 350 | 29.55 | 29.89 | 0.34 |
> > > | 400 | 28.49 | 28.83 | 0.34 |
> > >
> > > **Key Observations:**
> > >
> > > 1. **Rapid Convergence**: The initial gap (8.27 at step 0) rapidly decreases within the first 50 steps, dropping to 0.26.
> > >
> > > 2. **Stable Throughout Training**: After the initial convergence phase, the absolute gap remains consistently small (< 0.35) across the entire training horizon from step 50 to step 400.
> > >
> > > 3. **No Divergence**: Critically, the gap does **not** increase as training progresses, confirming that the Jensen bound remains a tight and reliable approximation to the true objective throughout the full training process.
> > >
> > > These extended results, with 50-sample MC estimation over 400 training steps, provide stronger empirical evidence that our Jensen-based optimization maintains fidelity to the true ELBO objective without divergence in later training phases.
> > >
> > > We sincerely thank you for this valuable suggestion, which has significantly strengthened the empirical validation of our theoretical framework.

---

> ### Author Response · Authors · 2025-12-02
> **Response to Extended Backbone Evaluation (Llama-3.1-8B)**
>
> **Extended Backbone Evaluation (Llama-3.1-8B)**
>
> Following Reviewer H3Rr's and your suggestion, we conducted comprehensive experiments on Llama-3.1-8B across all six datasets with all baseline methods, under our 5-step fine-tuning setting suggested by TFB[1] **(not the optimal setting for Variational Bayesian methods, best for methods that require deterministic training and post-hoc posterior estimation)**:
>
> **ACC (↑)**
>
> | Method | WG-S | ARC-C | ARC-E | WG-M | OBQA | BoolQ |
> |--------|------|-------|-------|------|------|-------|
> | LAP | 77.22 ± 1.38 | 84.49 ± 0.37 | 89.86 ± 0.54 | 84.04 ± 0.23 | 88.32 ± 0.33 | 88.26 ± 0.50 |
> | PoLAR_LAP | *77.51 ± 0.94* | *84.57 ± 1.10* | **91.80 ± 0.19** | *83.88 ± 0.90* | *88.70 ± 0.56* | **89.54 ± 0.86** |
> | PoLAR_LAP_LL | *77.51 ± 0.94* | *84.57 ± 1.10* | **91.80 ± 0.19** | *83.88 ± 0.90* | *88.70 ± 0.56* | **89.54 ± 0.86** |
> | TFB LL | 77.31 ± 2.01 | 82.75 ± 0.20 | 89.95 ± 0.59 | 83.44 ± 1.09 | 88.66 ± 0.30 | 89.15 ± 1.40 |
> | TFB | 77.61 ± 1.19 | 83.34 ± 1.03 | 90.90 ± 0.01 | 83.31 ± 0.35 | 88.44 ± 0.24 | 87.98 ± 1.79 |
> | BLOB | 76.68 ± 0.99 | 82.88 ± 0.51 | 91.49 ± 0.21 | 80.78 ± 1.05 | 87.90 ± 0.35 | 88.58 ± 0.17 |
> | C-LoRA | 73.07 ± 0.50 | 78.72 ± 0.47 | 89.88 ± 0.87 | 77.46 ± 0.78 | 86.59 ± 0.14 | 88.18 ± 1.29 |
> | ScalaBL | 49.92 ± 0.69 | 79.40 ± 1.48 | 89.21 ± 0.20 | 53.40 ± 1.43 | 69.62 ± 7.99 | 82.18 ± 2.14 |
> | PoLAR + VBLL (w/o LAP) | **77.91 ± 0.47** | **85.05 ± 0.77** | *91.60 ± 0.03* | **84.97 ± 0.05** | **89.07 ± 0.23** | *89.32 ± 1.29* |
> | PoLAR+VBLL | **77.91 ± 0.47** | **85.05 ± 0.77** | *91.60 ± 0.03* | **84.97 ± 0.05** | **89.07 ± 0.23** | *89.32 ± 1.29* |
>
> **ECE (↓)**
>
> | Method | WG-S | ARC-C | ARC-E | WG-M | OBQA | BoolQ |
> |--------|------|-------|-------|------|------|-------|
> | LAP | 4.15 ± 0.52 | 6.35 ± 0.84 | 9.94 ± 0.59 | 6.71 ± 1.69 | 3.54 ± 0.13 | *2.18 ± 0.47* |
> | PoLAR_LAP | *3.58 ± 0.18* | *4.52 ± 1.40* | 3.27 ± 0.86 | 4.31 ± 0.29 | 2.83 ± 0.49 | 3.35 ± 0.45 |
> | PoLAR_LAP_LL | 7.95± 0.92 | 6.07 ± 0.72 | **2.94 ± 0.67** | 6.37 ± 1.10 | 3.75 ± 0.01 | 5.52 ± 0.12 |
> | TFB LL | 9.99 ± 0.90 | 5.02 ± 2.15 | *3.13 ± 0.24* | 4.12 ± 1.57 | 3.58 ± 0.21 | 3.87 ± 1.41 |
> | TFB | 9.05 ± 0.39 | 6.53 ± 1.99 | 3.17 ± 0.21 | *3.68 ± 1.57* | 2.76 ± 0.16 | 3.89 ± 1.68 |
> | BLOB | 14.83 ± 1.21 | 9.29 ± 0.57 | 4.12 ± 0.26 | 8.23 ± 1.04 | 3.33 ± 0.29 | 3.36 ± 0.94 |
> | C-LoRA | 18.36 ± 0.58 | 15.56 ± 2.44 | 5.40 ± 0.23 | 7.96 ± 2.84 | 6.58 ± 1.23 | 3.94 ± 0.47 |
> | ScalaBL | 4.86 ± 0.50 | 18.75 ± 7.45 | 10.07 ± 0.66 | 2.73 ± 0.87 | 24.67 ± 4.34 | 14.48 ± 1.29 |
> | PoLAR + VBLL (w/o LAP) | 9.04 ± 0.18 | 7.40 ± 0.26 | 3.61 ± 0.22 | 6.10 ± 0.01 | *2.50 ± 0.10* | 2.63 ± 0.23 |
> | PoLAR+VBLL | **3.31 ± 1.40** | **3.56 ± 0.41** | 3.22 ± 0.42 | **3.00 ± 0.10** | **2.44 ± 0.11** | **1.88 ± 0.27** |
>
> **NLL (↓)**
>
> | Method | WG-S | ARC-C | ARC-E | WG-M | OBQA | BoolQ |
> |--------|------|-------|-------|------|------|-------|
> | LAP | 0.67 ± 0.04 | 0.63 ± 0.04 | 0.41 ± 0.02 | 0.52 ± 0.01 | 0.40 ± 0.02 | 0.36 ± 0.01 |
> | PoLAR_LAP | 0.60 ± 0.05 | 0.52 ± 0.01 | 0.31 ± 0.01 | 0.46 ± 0.02 | *0.34 ± 0.02* | 0.32 ± 0.01 |
> | PoLAR_LAP_LL | 0.76 ± 0.12 | 0.58 ± 0.05 | *0.26 ± 0.01* | 0.52 ± 0.06 | 0.37 ± 0.01 | 0.34 ± 0.03 |
> | TFB LL | *0.59 ± 0.05* | 0.53 ± 0.01 | 0.28 ± 0.02 | *0.41 ± 0.01* | **0.32 ± 0.01** | *0.29 ± 0.04* |
> | TFB | **0.55 ± 0.01** | *0.51 ± 0.02* | 0.27 ± 0.02 | *0.41 ± 0.01* | **0.32 ± 0.01** | 0.30 ± 0.02 |
> | BLOB | 0.79 ± 0.06 | 0.62 ± 0.07 | 0.29 ± 0.01 | 0.47 ± 0.03 | 0.38 ± 0.00 | **0.28 ± 0.00** |
> | C-LoRA | 0.85 ± 0.03 | 0.88 ± 0.08 | 0.35 ± 0.01 | 0.56 ± 0.07 | 0.45 ± 0.05 | 0.31 ± 0.01 |
> | ScalaBL | 0.65 ± 0.00 | 0.69 ± 0.13 | 0.35 ± 0.00 | 0.64 ± 0.00 | 0.96 ± 0.23 | 0.46 ± 0.08 |
> | PoLAR + VBLL (w/o LAP) | 0.61 ± 0.01 | *0.51 ± 0.05* | 0.27 ± 0.03 | *0.40 ± 0.01* | **0.32 ± 0.02** | *0.29 ± 0.00* |
> | PoLAR+VBLL | **0.55 ± 0.02** | **0.50 ± 0.05** | **0.24 ± 0.04** | **0.39 ± 0.01** | **0.32 ± 0.02** | *0.29 ± 0.00* |
>
> (**Bold** = best, *Italic* = second best)
>
> **Note on ScalaBL:** We strictly followed the official implementation and hyperparameter configurations for ScalaBL. However, despite our best efforts, our reproduction yielded a model where the low ECE comes at the cost of substantially reduced accuracy. This pattern suggests potential underfitting in this specific setting.
>
> These comprehensive results demonstrate that our framework generalizes effectively beyond Llama-2-7B to larger and more recent model architectures, consistently achieving superior uncertainty quantification while maintaining competitive predictive performance.
>
> [1] Shi, Haizhou, et al. "Training-free bayesianization for low-rank adapters of large language models." arXiv preprint arXiv:2412.05723 (2024).

---

> ### Author Response · Authors · 2025-12-02
> **Response to the extended ablation study**
>
> ## Ablation Study Across All Six Datasets
>
> Following your suggestion, we provide a comprehensive ablation study across all six datasets on Llama-3.1-8B, with five epochs of fine-tuning, to systematically evaluate the contribution of each component. Since our method applies the Laplace Approximation only to the last layer, we include PoLAR_LAP_LL (which applies LAP only to the last layer of a deterministic model) to ensure a direct, fair comparison.
>
> **ACC (↑)**
> | Method | WG-S | ARC-C | ARC-E | WG-M | OBQA | BoolQ |
> |--------|------|-------|-------|------|------|-------|
> | PoLAR_LAP | *77.51 ± 0.94* | *84.57 ± 1.10* | **91.80 ± 0.19** | *83.88 ± 0.90* | *88.70 ± 0.56* | **89.54 ± 0.86** |
> | PoLAR_LAP_LL | *77.51 ± 0.94* | *84.57 ± 1.10* | **91.80 ± 0.19** | *83.88 ± 0.90* | *88.70 ± 0.56* | **89.54 ± 0.86** |
> | PoLAR + VBLL (w/o LAP) | **77.91 ± 0.47** | **85.05 ± 0.77** | *91.60 ± 0.03* | **84.97 ± 0.05** | **89.07 ± 0.23** | *89.32 ± 1.29* |
> | PoLAR-VBLL (Full) | **77.91 ± 0.47** | **85.05 ± 0.77** | *91.60 ± 0.03* | **84.97 ± 0.05** | **89.07 ± 0.23** | *89.32 ± 1.29* |
>
> **ECE (↓)**
> | Method | WG-S | ARC-C | ARC-E | WG-M | OBQA | BoolQ |
> |--------|------|-------|-------|------|------|-------|
> | PoLAR_LAP | *3.58 ± 0.18* | *4.52 ± 1.40* | 3.27 ± 0.86 | 4.31 ± 0.29 | 2.83 ± 0.49 | 3.35 ± 0.45 |
> | PoLAR_LAP_LL | 7.95± 0.92 | 6.07 ± 0.72 | **2.94 ± 0.67** | 6.37 ± 1.10 | 3.75 ± 0.01 | 5.52 ± 0.12 |
> | PoLAR + VBLL (w/o LAP) | 9.04 ± 0.18 | 7.40 ± 0.26 | 3.61 ± 0.22 | 6.10 ± 0.01 | *2.50 ± 0.10* | *2.63 ± 0.23* |
> | PoLAR-VBLL (Full) | **3.31 ± 1.40** | **3.56 ± 0.41** | *3.22 ± 0.42* | **3.00 ± 0.10** | **2.44 ± 0.11** | **1.88 ± 0.27** |
>
> **NLL (↓)**
> | Method | WG-S | ARC-C | ARC-E | WG-M | OBQA | BoolQ |
> |--------|------|-------|-------|------|------|-------|
> | PoLAR_LAP | *0.60 ± 0.05* | 0.52 ± 0.01 | 0.31 ± 0.01 | 0.46 ± 0.02 | *0.34 ± 0.02* | 0.32 ± 0.01 |
> | PoLAR_LAP_LL | 0.76 ± 0.12 | 0.58 ± 0.05 | *0.26 ± 0.01* | 0.52 ± 0.06 | 0.37 ± 0.01 | 0.34 ± 0.03 |
> | PoLAR + VBLL (w/o LAP) | 0.61 ± 0.01 | *0.51 ± 0.05* | 0.27 ± 0.03 | *0.40 ± 0.01* | **0.32 ± 0.02** | **0.29 ± 0.00** |
> | PoLAR-VBLL (Full) | **0.55 ± 0.02** | **0.50 ± 0.05** | **0.24 ± 0.04** | **0.39 ± 0.01** | **0.32 ± 0.02** | **0.29 ± 0.00** |
>
> (**Bold** = best, *Italic* = second best)
>
> ### Key Observations
>
> **1. VBLL is the Dominant Factor:** PoLAR-VBLL achieves the best ECE and NLL overall, while maintaining competitive accuracy. Even without the final LAP refinement, PoLAR + VBLL (w/o LAP) already delivers highly competitive calibration on OBQA and BoolQ. This directly refutes the claim that VBLL is a failing component being remedied by LAP.
>
> **2. Deterministic Training Fails for UQ:** While PoLAR + LAP and PoLAR_LAP_LL achieve comparable NLL (mainly due to high confidence predictions), their relatively high ECE reveals that standard deterministic training fails to discover posterior geometries suitable for uncertainty quantification. The reasonable NLL reflects overconfidence rather than proper calibration.
> Comparing PoLAR_LAP and PoLAR_LAP_LL reveals that applying LAP to all layers achieves substantially better calibration than last-layer-only LAP on almost all datasets. This indicates that deterministic training fails to discover posterior geometries suitable for uncertainty quantification—LAP must compensate across all layers to achieve reasonable calibration.
>
> **3. VBLL Provides Superior Initialization for LAP:** **Our PoLAR-VBLL applies LAP only to the last layer** yet achieves superior calibration compared to full-layer PoLAR_LAP. This demonstrates the effectiveness of variational training: VBLL actively guides optimization toward high-quality posterior modes, providing the necessary foundation for the final LAP refinement. The well-calibrated representations from VBLL require only minimal post-hoc adjustment to achieve state-of-the-art uncertainty quantification.
>
> **Note:** **Under our 5-step fine-tuning setting suggested by TFB[1]  (not the optimal setting for Variational Bayesian methods, best for methods that require deterministic training and post-hoc posterior estimation)**, deterministic methods (PoLAR_LAP and PoLAR_LAP_LL) benefit in ACC from pure cross-entropy training that exclusively optimizes accuracy. In contrast, VBLL jointly optimizes accuracy and the KL divergence term, which may result in very slightly lower ACC on some datasets. However, this joint optimization yields substantially superior uncertainty quantification, as evidenced by consistently better ECE and NLL.
>
> ### Conclusion
>
> These comprehensive results across six datasets demonstrate that VBLL is the core working factor in our framework. The synergy between VBLL (mode discovery during training) and LAP (local geometry refinement post-hoc) is intentional and effective—not a case of LAP compensating for a failing VBLL component.
>
> [1] Shi, Haizhou, et al. "Training-free bayesianization for low-rank adapters of large language models." arXiv:2412.05723 (2024).

---

### Author Response · Authors · 2025-12-03
**Summary of Discussion with Reviewer XbV5**

**Initial score: 2 → Raised to 4 after rebuttal, with willingness to raise further. Discussions were ongoing but interrupted by the incident.**


The reviewer's comments mainly relate to technical novelty, computational efficiency, Jensen bound tightness, ablation studies, and backbone diversity. We provide comprehensive responses and additional experiments addressing each concern:


- **Technical Novelty:** We clarify that our contribution addresses a fundamental limitation: UQ quality is bottlenecked by feature representation quality. We provide direct evidence that PoLAR's higher stable rank (2.86 vs. LoRA's 1.53) improves the distance-awareness required by last-layer Bayesian methods, in line with SNGP's theoretical framework [1]. We provide a comprehensive analysis in Appendix B6. **The reviewer acknowledged that a better adapter improves feature quality and thereby aids in Uncertainty Quantification (UQ) ability.**


- **Training and Test Efficiency:** We validated the computational costs for GPU memory and inference time experiments and provided confirmed results. Memory differences between BLoB and its variants [2, 3] result from variational parameter overhead and reparameterization activation costs, not just static parameter counts. Our head-only sampling achieves 12× faster inference than BLoB-based methods (73s vs. 876s). In Appendix B.1 Table 4, we include detailed GPU memory and inference-time experimental results, along with a comprehensive analysis of the actual computational overhead of BLoB and its variants. We also included the TFB last-layer variant, as requested, in the ablation study and the extended backbone experiments. **The reviewer acknowledged that our method has better inference efficiency than the full version BLoB and TFB.**


-  **Jensen Bound Tightness:**  We conducted extended experiments with 50-sample MC estimation over 400 training steps, showing that the Jensen-MC gap stays small and remains stable throughout training, especially in later stages. We also provide a thorough comparison between the Jensen Bound and the MC-estimated training loss, with an analysis in Appendix B.4. **The reviewer acknowledged the Jensen's bound experiment design and initial results.**


- **Comprehensive Ablation Study:** We performed a complete ablation across all six datasets on Llama-3.1-8B, confirming VBLL as the dominant factor. PoLAR+VBLL (w/o LAP) already achieves superior calibration compared to deterministic baselines, demonstrating that LAP refines rather than remedies VBLL's contribution. In Appendix B.3, we provide the complete ablation study results and detailed analysis. **The reviewer acknowledged that the initial ablation study is persuasive.**


- **Backbone Diversity:** We conducted comprehensive experiments on Llama-3.1-8B across all six datasets using all baseline methods, demonstrating consistent improvements in ACC, ECE, and NLL over TFB, TFB_LL, BLoB, and other recent baselines. We present detailed experimental results and analysis in Appendix B.2.


[1] Liu, Jeremiah, et al. "Simple and principled uncertainty estimation with deterministic deep learning via distance awareness." Advances in neural information processing systems 33 (2020): 7498-7512.


[2] Shi, Haizhou, et al. "Training-free bayesianization for low-rank adapters of large language models." arXiv preprint arXiv:2412.05723 (2024).


[3] Rahmati, Amir Hossein, et al. "C-LoRA: Contextual Low-Rank Adaptation for Uncertainty Estimation in Large Language Models." arXiv preprint arXiv:2505.17773 (2025).


[4] Samplawski, Colin, et al. "Scalable Bayesian Low-Rank Adaptation of Large Language Models via Stochastic Variational Subspace Inference." arXiv preprint arXiv:2506.21408 (2025).

---

### Author Response · Authors · 2025-12-03
**Summary of Discussion with Reviewer y4uo**

**Initial score: 2. Discussions were ongoing but interrupted by the incident.**


The reviewer's comments mainly relate to experimental details, ablation studies, and computational cost analysis. We provide comprehensive clarifications and additional experiments. In particular,


- **Experimental Details:** We provided detailed reproduction protocols for the baselines, including all shared hyperparameters, and confirmed that all baselines were implemented from official repositories with identical settings across methods. In Appendix A.4 and B.2, we present the detailed implementation details for LLaMA 2 7B and LLaMA 3.1 8B, respectively.


- **Training Schedule:** We clarified that the 500-epoch ceiling was adopted because BLoB's official 32-epoch configuration failed to converge (exhibiting training instability and NaN predictions) and required many more epochs to replicate their result in the paper from our reproduction. We provide detailed replication information and results, including training and testing outcomes with various random seeds in Appendix A6. **The reviewer acknowledged it.**


- **Ablation Studies:** We conducted comprehensive ablation studies on Llama-3.1-8B across all six datasets, demonstrating that VBLL is the primary driver of uncertainty quantification.  In Appendix B.3, we provide the complete ablation study results and detailed analysis.


- **Computational Cost:** We clarified that our GPU memory measurements use different experimental settings from the original BLoB paper. The number of trainable parameters is the same as in the official repositories. We validated computational costs for the GPU memory and inference time experiment. Memory differences between BLoB and its variants [1, 2] arise from variational parameter overhead and reparameterization activation costs—factors beyond static parameter counts. Our head-only sampling achieves 12× faster inference than BLoB-based methods (73s vs. 876s). In Appendix B.1, we present detailed GPU memory and inference-time experimental results, along with a thorough analysis of the true computational overhead of BLoB and its variants. **The reviewer acknowledged that our method has better inference efficiency than the full version BLoB and TFB.**


- **Extended Experiments:** Following reviewer suggestions, we conducted experiments on Llama-3.1-8B across all six datasets using a practical 5-epoch setting from TFB [3], demonstrating consistent improvements over all baselines. We present detailed experimental results and analysis in Appendix B.2.


**Lastly, we note that the reviewer's concerns focus primarily on defending BLoB's reported efficiency.** However, BLoB's high memory cost is well-documented and motivates recent methods (ScalaBL [1], C-LoRA [2]) specifically designed to reduce this overhead.


[1] Rahmati, Amir Hossein, et al. "C-LoRA: Contextual Low-Rank Adaptation for Uncertainty Estimation in Large Language Models." arXiv preprint arXiv:2505.17773 (2025).


[2] Samplawski, Colin, et al. "Scalable Bayesian Low-Rank Adaptation of Large Language Models via Stochastic Variational Subspace Inference." arXiv preprint arXiv:2506.21408 (2025).


[3] Shi, Haizhou, et al. "Training-free bayesianization for low-rank adapters of large language models." arXiv preprint arXiv:2412.05723 (2024).

---

### Author Response · Authors · 2025-12-03
**Summary of Discussion with Reviewer H3Rr**

**The initial score was 6, and the reviewer acknowledged that "the rebuttal resolves my main technical concerns" after the discussion.**

The reviewer's comments mainly relate to the tightness of the Jensen bound, the hybrid VB+Laplace design, cross-class correlations in posterior factorization, single-backbone evaluation, and sensitivity to priors/initialization.

We provided empirical verification showing VBLL is the primary driver of improved calibration (not merely "rescued" by Laplace), clarified the intentional two-stage design, acknowledged the scalability trade-off for posterior factorization, and demonstrated robustness across prior scales and random seeds. We provide a detailed analysis of sensitivity to priors and initialization in Appendix B.5.

**The reviewer confirmed that our responses addressed their main technical concerns, noting "the remaining issues primarily concern evaluation scope and clarity of positioning rather than the soundness of the proposed method."**


Regarding the remaining evaluation scope concerns, we have now completed: (1) comprehensive experiments on Llama-3.1-8B across all six datasets, addressing the single backbone limitation, which can be seen in Appendix B.2; (2) extended Jensen bound analysis with 50-sample MC estimation across 400 training steps, showing the gap remains small and stable in later training phases, which also present in the Appendix B.4.

---

### Author Response · Authors · 2025-12-03
**Summary of Discussion with Reviewer ASkM**

**Initial score: 4 → Raised to 6 after rebuttal, stating "my primary concerns have been met." Although scores have been reverted due to the incident, this positive engagement reflects the strength of our responses.**


The reviewer's comments mainly relate to the lack of formal analysis linking orthogonality to uncertainty calibration, and missing comparisons with recent SOTA Bayesian LoRA techniques. We provided a principled theoretical chain (orthogonality → rank diversity → calibrated uncertainty) supported by stable rank analysis showing LoRA collapses to near-rank-1 while PoLAR preserves geometry, and demonstrated consistent improvement across all tested Bayesian methods. We provide a comprehensive analysis in Appendix B6.


We also conducted comprehensive experiments comparing against TFB, C-LoRA, and ScalaBL, where PoLAR-VBLL achieves the best accuracy, NLL, and ECE among all datasets. We present detailed experimental results and analysis in Appendix B.2.


**The reviewer confirmed that our responses adequately addressed their concerns.**

---

### Meta-Review · Area_Chair_He1F · 2026-01-01

**Summary:**

This paper proposes PoLAR-VBLL, a scalable Bayesian fine-tuning framework for large language models that integrates PoLAR adapters, Variational Bayesian Last Layer (VBLL) inference, and an optional post-hoc Laplace Approximation (LA). Experiments across multiple common-sense reasoning benchmarks show consistent improvements in accuracy and uncertainty calibration (ECE, NLL), while substantially reducing inference cost compared to BLoB-style Bayesian model averaging approaches.

Reviewers broadly agreed on the paper’s clear presentation and strong empirical performance. The central debate focused on technical novelty, clarity of motivation, and experimental rigor, especially whether the method represents more than a principled integration of existing techniques and whether the reported efficiency advantages over BLoB/TFB are fully convincing. Through extensive rebuttal and discussion, the authors added detailed ablations, theoretical justification, additional baselines, extended Jensen-bound analyses, and backbone generalization experiments. These additions significantly strengthened the paper, though some concerns remain partially unresolved. Additionally, I also found some of the added baselines, e.g., TFB, are moved to the Appendix of the revision due to space limit. Given these seem to be the reviewers' major concerns, it would make sense to include them in the main paper instead.

Overall, the submission is borderline, with strong empirical evidence but lingering skepticism regarding novelty framing and parts of the computational analysis. I would strongly recommend that the authors take into account the reviewers' comments in the revision.

**Reviewer Concerns:**

Concerns Largely Addressed

- Lack of ablation studies
  Addressed with comprehensive ablations isolating PoLAR, VBLL, and LA across all six datasets and multiple backbones, clearly demonstrating VBLL as the primary contributor to uncertainty calibration.

- Jensen bound tightness and VBLL effectiveness
  Addressed through extended Jensen-vs-Monte-Carlo comparisons (up to 50-sample MC over long training horizons), showing stable and tight bounds throughout training.

- Missing recent baselines (TFB, C-LoRA, ScalaBL)
  Addressed by implementing and comparing against these baselines, including last-layer TFB variants under comparable computational budgets.

- Single-backbone evaluation
  Addressed via extensive experiments on Llama-3.1-8B across all six benchmarks.

- Motivation for PoLAR in UQ
  Strengthened by stable-rank analysis and explicit connections to distance-aware feature requirements in Bayesian last-layer methods.

Concerns Still Partially Outstanding

- Technical novelty
  Some reviewers remain unconvinced that the PoLAR + VBLL + LA pipeline constitutes a fundamentally new method rather than a well-justified integration of existing techniques.

- Training memory and efficiency analysis
  Despite detailed clarifications, skepticism persists regarding whether the reported memory and runtime gaps with BLoB/TFB fully align with prior theoretical and empirical expectations.

- Experimental clarity
  Minor issues remain concerning training budget descriptions (epochs vs. steps) and presentation of baseline results.

**Reviewer Scores:**

- **Reviewer XbV5**
  - Initial score: **2**
  - Estimated final score: **4**
  - Rationale: Raised score after rebuttal to 4; most empirical and methodological concerns were addressed, but doubts remain about technical novelty and some efficiency claims.

- **Reviewer y4uo**
  - Initial score: **2**
  - Estimated final score: **2 or 4**
  - Rationale: Additional ablations and clarifications improved confidence, but continued skepticism about novelty and reported memory usage of BLoB/TFB.

- **Reviewer H3Rr**
  - Initial score: **6**
  - Estimated final score: **6**
  - Rationale: Backbone diversity and extended experiments directly addressed the main concern.

- **Reviewer ASkM**
  - Initial score: **4**
  - Estimated final score: **6**
  - Rationale: Satisfied by the inclusion of recent baselines and strengthened empirical validation, viewing the work as practically valuable.

---

### Decision · Program_Chairs · 2026-01-26

Reject